# RIEMANNIAN VARIATIONAL FLOW MATCHING FOR MATERIAL AND PROTEIN DESIGN

**Olga Zaghen**[1]  **Floor Eijkelboom**[1,2]\*  **Alison Pouplin**[4]\*  **Cong Liu**[1,3]
**Max Welling**[1,2]  **Jan-Willem van de Meent**[1,2]  **Erik J. Bekkers**[1]
[1] AMLab, [2] Bosch-Delta Lab, [3] AI4Science Lab, University of Amsterdam
[4] Aalto University
o.zaghen@uva.nl

## ABSTRACT

We present Riemannian Gaussian Variational Flow Matching (RG-VFM), a geometric extension of Variational Flow Matching (VFM) for generative modeling on manifolds. Motivated by the benefits of VFM, we derive a variational flow matching objective for manifolds with closed-form geodesics based on Riemannian Gaussian distributions. Crucially, in Euclidean space, predicting endpoints (VFM), velocities (FM), or noise (diffusion) is largely equivalent due to affine interpolations. However, on curved manifolds this equivalence breaks down. We formally analyze the relationship between our model and Riemannian Flow Matching (RFM), revealing that the RFM objective lacks a curvature-dependent penalty – encoded via Jacobi fields – that is naturally present in RG-VFM. Based on this relationship, we hypothesize that endpoint prediction provides a stronger learning signal by directly minimizing geodesic distances. Experiments on synthetic spherical and hyperbolic benchmarks, as well as real-world tasks in material and protein generation, demonstrate that RG-VFM more effectively captures manifold structure and improves downstream performance over Euclidean and velocity-based baselines.

## 1 INTRODUCTION

Generative models play a central role in machine learning, as they provide a way to synthesize data and learn complex distributions. Diffusion models (Ho et al., 2020; Song et al., 2020) achieve state-of-the-art performance, but rely on a fixed Gaussian noising process with predetermined variance schedules. As a result, the reverse process is tied to this prescribed family of Gaussian marginals, and sampling requires numerical integration with diffusion-specific samplers. In contrast, Continuous normalizing flows (CNFs) (Chen et al., 2018) directly learn the vector field of an ODE that transports a base distribution into the data distribution. In principle, this allows the transport path to be fully learned, but both training and sampling are computationally demanding since likelihood evaluation involves solving a high-dimensional ODE (Ben-Hamu et al., 2022; Rozen et al., 2021; Grathwohl et al., 2019). Flow Matching (FM) (Lipman et al., 2023) offers a simulation-free alternative, as it defines per-sample interpolants between the source and the target samples, and regresses the vector field to known conditional velocities.

Recent developments have extended flow matching in two key directions. Variational Flow Matching (VFM) (Eijkelboom et al., 2024) reframes the problem as posterior inference over trajectories, providing a probabilistic perspective with flexible modeling choices. In parallel, Riemannian Flow Matching (RFM) (Chen & Lipman, 2024) has shown how incorporating non-Euclidean geometry can improve modeling of distributions supported on manifolds.

VFM has demonstrated advantages over standard FM in discrete domains (e.g., *CatFlow*) and has been extended to mixed data modalities (Guzmán-Cordero et al., 2025) as well as molecular generation tasks (Eijkelboom et al., 2025; Sakalyan et al.). A key strength of the variational formulation is its flexibility: problem-specific constraints can be incorporated directly into the objective. For

---

\*These authors contributed equally.

instance, censored flow matching for sea-ice forecasting enforces physical bounds such as non-negative ice thickness through the variational loss (Finn et al., 2025). Similarly, when the target distributions live on a Riemannian manifold, the objective can be formulated to respect the geometry of the data support, which is the setting we study in this paper.

The geometric extension is particularly relevant for biological and chemical domains where intrinsic geometric structure governs the data. Recently, generative models have been extensively applied to material discovery (Jiao et al., 2023; Fu et al., 2023; Kim et al., 2024) and to the generation of large biomolecules such as protein backbones (Guo et al., 2025; Yue et al., 2025; Yim et al., 2023a;b). These applications highlight that data often live on heterogeneous manifolds: Euclidean space for atomic coordinates, rotation groups $SO(3)$ for orientations, and other structured domains. Early works in crystal generation, such as Jiao et al. (2023), focus purely on Euclidean parameters without explicitly modeling rotational degrees of freedom. In contrast, recent methods for metal-organic frameworks (MOFs) and proteins (Yim et al., 2023a; Yue et al., 2025; Kim et al., 2024; Guo et al., 2025) adopt a mixed approach where Euclidean parameters (e.g., positions) are modeled with standard FM while non-Euclidean parameters (e.g., rotations) are modeled with RFM. These methods lack a fully variational treatment across both parameter types, and we address this gap by demonstrating the benefits of our geometric variational approach on these applications.

When extending from Euclidean space to general Riemannian manifolds, it becomes unclear how different generative objectives relate. In Euclidean space, training a generative model by predicting an endpoint (VFM), a velocity (FM), or noise (diffusion) is largely equivalent up to affine transformations, since these quantities – noise, score, velocity field, and endpoints – are interchangeable parameterizations of the same training signal (Vuong et al., 2025; Lipman et al., 2023; Eijkelboom et al., 2024). On curved manifolds, however, this near-equivalence breaks down: tangent spaces vary across points and curvature introduces higher-order deviations, preventing any explicit closed-form relation between the velocity-based and endpoint-based perspectives. This raises the questions: if the endpoint and velocity-based perspectives are no longer equivalent on curved manifolds, how do they differ, and which one is preferable?

In this paper, we develop Riemannian Gaussian Variational Flow Matching (RG-VFM), which extends VFM to Riemannian manifolds with closed-form metrics, thereby bridging the variational and geometric extensions of flow matching. Our contributions are threefold:

- We define a variational flow matching objective for general geometries, extending endpoint-based training to manifolds.
- We formally analyze its properties, establishing how RG-VFM relates to RFM and showing that the gap between them encodes curvature through Jacobi fields.
- We demonstrate that *variationalizing* existing geometric generative models in material and protein design consistently improves performance, highlighting the practical advantages of endpoint-based training.

## 2 BACKGROUND

**Flow Matching.** Modern generative modeling interprets sampling from a target distribution $p_1$ as transporting a base distribution $p_0$ by learning dynamics. Typically, $p_0$ is a standard Gaussian, and the transformation follows a time-dependent mapping $\varphi_t \colon [0,1] \times \mathbb{R}^D \to \mathbb{R}^D$ where $\varphi_0$ is the identity and $\varphi_1$ pushes $p_0$ onto $p_1$. E.g., normalizing flows (Chen et al., 2018) use an ODE governed by some time-dependent velocity field $u_t$. Though likelihood training is possible through the change of variables formula, solving an ODE during training is expensive.

Flow Matching (FM) (Lipman et al., 2023; Liu et al., 2023; Albergo et al., 2023) bypasses this by defining an interpolation between noise and data, and directly learning the associated velocity field in a self-supervised manner. Though the goal is to learn the intractable objective

$$\mathcal{L}_{\text{FM}}(\theta) = \mathbb{E}_{t,x}\big[\|u_t(x) - v_t^\theta(x)\|^2\big], \tag{1}$$

this can be made computationally feasible by reformulating $u_t$ with a conditional velocity field (i.e. assumed dynamics towards a given $x_1$, or time derivative of the interpolation), giving rise to Conditional Flow Matching (CFM):

$$\mathcal{L}_{\text{CFM}}(\theta) = \mathbb{E}_{t,x_1,x}\left[\|u_t(x \mid x_1) - v_t^\theta(x)\|^2\right]. \tag{2}$$

Minimizing eq. (2) provides an unbiased estimate of $\nabla_\theta \mathcal{L}_{\text{FM}}$, allowing efficient per-sample training. As FM can be seen as regressing directly onto the derivative of an interpolant between source and target in a self-supervised manner, it provides a unifying framework: by choosing different interpolations, dynamics, or conditioning structures, it can be adapted to various data types and constraints.

**Riemannian Flow Matching.** Riemannian Flow Matching (RFM) (Chen & Lipman, 2024) extends FM to Riemannian manifolds. Given a smooth Riemannian manifold $\mathcal{M}$ with closed-form geodesics and metric $\mathbf{g}$, RFM learns a vector field $v_t$:

$$\mathcal{L}_{\text{RFM}}(\theta) = \mathbb{E}_{t,x_1,x}\left[\left\|v_t^\theta(x) - \log_x(x_1)/(1-t)\right\|_{\mathbf{g}}^2\right],\qquad(3)$$

with $\log_x(x_1)$ denoting the Riemannian log map, which returns the initial velocity vector of the geodesic connecting $x$ to $x_1$ (more details on Riemannian manifolds are in section C.1).

Unlike Euclidean Flow Matching, RFM respects the curvature and geodesics of the underlying space $\mathcal{M}$. Through geodesic or spectral distances, it enables simulation-free training when manifold operations are available, and can utilize approximate distances when closed-form geodesics are intractable, maintaining theoretical guarantees while enabling efficient generative modeling.

**Variational Flow Matching.** Variational Flow Matching (VFM) (Eijkelboom et al., 2024) reformulates FM by introducing a variational distribution $q_t^\theta(x_1 \mid x)$ to approximate the unknown posterior $p_t(x_1 \mid x)$, where the learned velocity $v_t^\theta$ is expressed as the expectation of the conditional velocity under this variational approximation over trajectories. Then, the VFM objective is to minimize the KL divergence between joint distributions, i.e.:

$$\mathcal{L}_{\text{VFM}}(\theta) = \mathbb{E}_t\left[\text{KL}\left(p_t(x_1,x) \,\|\, q_t^\theta(x_1,x)\right)\right] = -\mathbb{E}_{t,x_1,x}\left[\log q_t^\theta(x_1 \mid x)\right] + \text{const.}\qquad(4)$$

When $u_t(x \mid x_1)$ is linear in $x_1$ – e.g. a straight-line interpolation – the expectation depends only on marginal distributions, implying this objective reduces to a series of $D$ univariate tasks:

$$\mathcal{L}_{\text{VFM}}(\theta) = -\mathbb{E}_{t,x_1,x}\left[\sum_{d=1}^{D} \log q_t^\theta(x_1^d \mid x)\right], \text{ e.g. } \mathcal{L}_{\text{VFM}}(\theta) = \mathbb{E}_{t,x_1,x}\left[\|\mu_t^\theta(x) - x_1\|^2\right]\qquad(5)$$

in case $q_t^\theta$ is Gaussian, relating VFM directly back to FM (see Eijkelboom et al. (2024) for details). For sampling with the standard flow matching case of linear interpolation, the vector field reduces to the first moment of the variational approximation:

$$v_t^\theta(x) = \mathbb{E}_{q_t^\theta(x_1|x)}\left[\frac{x_1 - x}{1 - t}\right] = \frac{\mathbb{E}_{q_t^\theta(x_1|x)}[x_1] - x}{1 - t} = \frac{\mu_t^\theta(x) - x}{1 - t}.\qquad(6)$$

A key feature of VFM is its flexibility in choosing $q_t^\theta$, as different choices allow adaptation to various geometries and data types, improving efficiency and expressiveness.

## 3 RIEMANNIAN GAUSSIAN VARIATIONAL FLOW MATCHING

The geometric generalization of the VFM framework stems from the observation that *the posterior probability $p_t(x_1 \mid x)$ implicitly encodes the geometry of the distribution's support*. For example, in CatFlow (Eijkelboom et al., 2024), defining $q_t^\theta(x_1 \mid x)$ as a categorical distribution ensures that the velocities point towards the probability simplex. This raises the question of whether other geometric information about the support of $p_1$ can be similarly encoded in $q_t^\theta(x_1 \mid x)$.

To investigate this, we consider the case where $p_t(x_1 \mid x)$ is defined as a Gaussian distribution with its support on a general manifold $\mathcal{M} := \text{supp}(p_1)$ rather than being restricted to Euclidean space. In this setting, the Riemannian Gaussian distribution naturally arises as a generalization of the Gaussian to a Riemannian manifold. We refer to velocity-inferring methods (CFM and RFM) as *vanilla* models and endpoint-inferring methods (VFM and RG-VFM) as *variational* models.

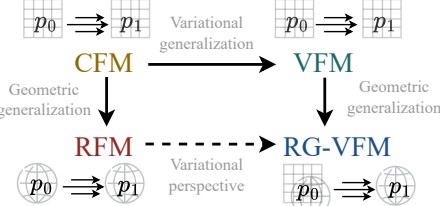

Figure 1: Overview of the models relevant for our framework. The square represents Euclidean space, while the sphere represents a general $\mathcal{M}$.

The advantages of the variational perspective in a geometric setting are twofold:

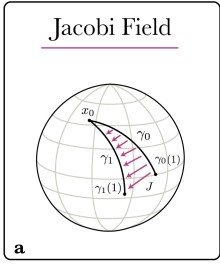 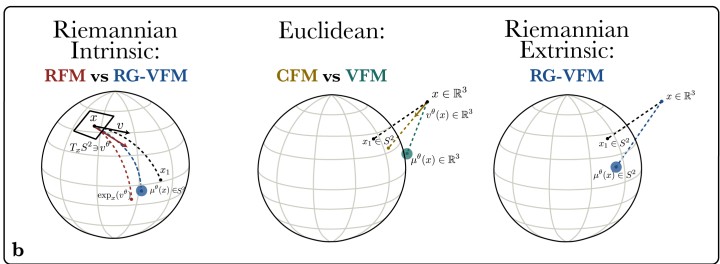

Figure 2: **a**: Representation of a shooting family of geodesics on $\mathbb{S}^2$ with corresponding Jacobi field. **b**: Visualization of predictions (color-coded to match the name colors) of each model relevant to our framework, for a target distribution $p_1$ supported on the sphere $\mathbb{S}^2$.

- *Flexibility on the support of the distribution*: the prior $p_0$ can be defined either on $\mathcal{M}$ (*intrinsic*) or in the ambient Euclidean space (*extrinsic*), while vanilla RFM only supports the intrinsic viewpoint. The extrinsic framework maintains the simplicity and efficiency of a linear flow in Euclidean space, avoiding the need for the manifold's exponential and logarithmic maps, while encoding more geometric information than purely Euclidean methods (section 3.1). Note that, while *intrinsic* and *extrinsic* traditionally refer to the manifold's internal geometry versus its embedding, we use these terms to distinguish whether points lie on the manifold or in the ambient space, rather than coordinate choices. For example, our intrinsic framework can be expressed using ambient coordinates.

- *Supervision on the endpoints*, rather than on the velocities, by minimizing their geodesic distance on the manifold, which in practice leads to more effective learning of the signal. We show this in section 4, by reformulating the objective through Jacobi fields.

## 3.1 THE RIEMANNIAN GAUSSIAN VFM OBJECTIVE

To extend VFM to the geometric case, one first needs to define a relevant variational posterior with support over the manifold. In contrast to Euclidean settings, we need to take particular care to ensure the distribution is properly defined on the manifold. Let $\mathcal{M}$ be a Riemannian manifold with metric $\mathbf{g}$: the Riemannian Gaussian (RG) distribution (Pennec, 2006) is defined as the maximum–entropy distribution specified by its mean value and covariance, formally

$$\mathcal{N}_{\text{Riem}}(z \mid \sigma, \mu) = \frac{1}{C} \exp\left(-\frac{\text{dist}_{\mathbf{g}}(z, \mu)^2}{2\sigma^2}\right), \tag{7}$$

where $z, \mu \in \mathcal{M}$ (with $\mu$ as the mean), $\sigma > 0$ is a scale parameter, and $\text{dist}_{\mathbf{g}}(z, \mu)$ denotes the geodesic distance determined by $\mathbf{g}$. The constant $C$ depends on both $z$ and $\mu$, and it normalizes the distribution over $\mathcal{M}$. A more detailed geometric explanation can be found in section C.2.

We define the Riemannian Gaussian VFM objective by using the Riemannian Gaussian as our variational approximation, i.e.

$$\mathcal{L}_{\text{RG-VFM}}(\theta) = \mathbb{E}_{t,x_1,x} \left[-\log \mathcal{N}_{\text{Riem}}(x_1 \mid \mu_t^\theta(x), \sigma_t(x))\right]. \tag{8}$$

In the Euclidean Gaussian VFM case, this setting reduces to a straightforward mean squared error optimization, so it is natural to wonder whether a similar simplification holds here. In fact, such a simplification exists under two assumptions: (1) the manifold is homogeneous – that is, any point can be transformed into any other by a distance-preserving symmetry (a formal definition is provided in section C.1); and (2) we have access to a closed-form expression for its geodesics. Notably, these requirements are not too restrictive, as most manifolds used in deep learning satisfy them, including $S^n$, $\mathbb{H}^n$, $\mathbb{T}^n$, and $SO(n)$. Formally, the following holds (see section D.1 for details):

**Proposition 3.1.** *Let $\mathcal{M}$ be a homogeneous manifold with closed-form geodesics. Then, the RG-VFM objective reduces to*

$$\mathcal{L}_{\text{RG-VFM}}(\theta) = \mathbb{E}_{t,x_1,x} \left[||\log_{x_1}(\mu_t^\theta(x))||_{\mathbf{g}}^2\right] = \mathbb{E}_{t,x_1,x} \left[\text{dist}_{\mathbf{g}}(x_1, \mu_t^\theta(x))^2\right], \tag{9}$$

*where* $\log$ *denotes the **logarithmic map** on the manifold and* $\text{dist}_{\mathbf{g}}$ *is the geodesic distance.*

---

**Algorithm 1** RG-VFM intrinsic

**Require:** base $p \in \mathcal{M}$, target $q \in \mathcal{M}$.
  Initialize parameters $\theta$ of $\mu_t$
  # **Training Phase**
  **while** not converged **do**
    sample $t \sim \mathcal{U}(0,1)$, $x_0 \sim p$, $x_1 \sim q$
    compute geodesic interpolation:
      $x_t = \exp_{x_0}(t \cdot \log_{x_0}(x_1))$
    $\ell(\theta) = \mathbb{E}_{t,x_1,x}\left[\text{dist}_g^2(x_1, \mu_t(x_t;\theta))\right]$
    $\theta = \texttt{optimizer\_step}(\ell(\theta))$
  **end while**

  # **Generation Phase**
  sample noise $x_0 \sim p$
  $x_1 = \texttt{solve\_ODE}\left([0,1], x_0, \frac{\log_{x_t}(\mu_t(x_t;\theta))}{1-t}\right)$

---

**Algorithm 2** RG-VFM extrinsic

**Require:** base $p \in \mathbb{R}^d$, target $q \in \mathcal{M}$.
  Initialize parameters $\theta$ of $\mu_t$
  # **Training Phase**
  **while** not converged **do**
    sample $t \sim \mathcal{U}(0,1)$, $x_0 \sim p$, $x_1 \sim q$
    compute linear interpolation:
      $x_t = t \cdot x_1 + (1-t) \cdot x_0$
    $\ell(\theta) = \mathbb{E}_{t,x_1,x}\left[\text{dist}_g^2(x_1, \mu_t(x_t;\theta))\right]$
    $\theta = \texttt{optimizer\_step}(\ell(\theta))$
  **end while**

  # **Generation Phase**
  sample noise $x_0 \sim p$
  $x_1 = \texttt{solve\_ODE}\left([0,1], x_0, \frac{\mu_t(x_t;\theta)-x_t}{1-t}\right)$

---

Minimizing this loss is equivalent to computing the Fréchet mean of the distribution, that is: $\mu^\star = \arg\min_{\mu_\theta \in \mathcal{M}} \mathbb{E}_{x_1}\left[\text{dist}_g(x_1, \mu_\theta)^2\right]$, averaged over the generative steps $t$ and samples $x$. In other words, the point $\mu_\theta$ minimizes the expected squared geodesic distance to the target (Fréchet, 1948). Intuitively, this can be viewed as a generalization of the mean squared error from the Euclidean setting to a Riemannian framework. We obtain this result by assuming that $\sigma_t(x)$ is constant. Nonetheless, this term could for example be set to $\sigma_t(x) = 1 - t$ to achieve time normalization, as done in our material and protein generation experiments (sections 5.2 and 5.3).

The RG-VFM objective (eq. (9)) minimizes the geodesic distance on $\mathcal{M}$ between predicted and target endpoints, so it only needs to capture the local geometry around $p_1$. This allows for a flexible choice of $p_0$'s support, leading to two plausible model variants when $\mathcal{M}$ is embedded in $\mathbb{R}^n$:

1. **RG-VFM-$\mathbb{R}^n$:** the prior $p_0$ is Euclidean with $\mathcal{M} \subsetneq \text{supp}(p_0) = \mathbb{R}^n$ and conditional velocities use linear interpolation in the ambient Euclidean space $\mathbb{R}^n$;

2. **RG-VFM-$\mathcal{M}$:** the prior is intrinsic ($\text{supp}(p_0) \subseteq \mathcal{M}$) with conditional velocities defined via geodesic interpolation on tangent spaces. Here, no embedding of $\mathcal{M}$ in $\mathbb{R}^n$ is required.

The extrinsic variant RG-VFM-$\mathbb{R}^n$ thus learns a simple linear flow while retaining a geometry-aware loss, whereas the intrinsic variant RG-VFM-$\mathcal{M}$ mirrors the RFM setup but differs in its loss definition (algorithms 1 and 2). Indeed, Vanilla RFM also requires $\text{supp}(p_0) \subseteq \mathcal{M}$ because its vector fields depend on the manifold's intrinsic geometry. Because of these different frameworks, direct comparison is only meaningful between RG-VFM-$\mathcal{M}$ and RFM, a comparison we present in the next section. The choice between intrinsic and extrinsic versions represents a trade-off: the extrinsic version can only be used in an ambient space $\mathbb{R}^d$ of sufficiently large dimension to embed the manifold without degeneracy. In such cases, linear interpolation simplifies implementation and reduces computational costs by requiring only the geodesic distance, rather than logarithm and exponential maps at every step.

## 4 RG-VFM VS RFM: A COMPARISON BASED ON JACOBI FIELDS

In this section, we refer to RG-VFM-$\mathcal{M}$ simply as RG-VFM for brevity. Given a sampled intermediate point at timestep $t$, our variational loss $\mathcal{L}_{\text{RG-VFM}}$ measures the geodesic distance between target and predicted endpoints on the manifold. In contrast, the vanilla loss $\mathcal{L}_{\text{RFM}}$ in Riemannian flow matching compares target and predicted velocities in the tangent space at that point.

In Euclidean space, these two formulations coincide since the difference between the endpoints is directly proportional to the difference between their initial velocities. But in curved space, this equivalence breaks down: curvature influences how geodesics separate from one another.

In this section, we examine how small changes in initial velocities affect geodesic endpoints by constructing families of related geodesics. In differential geometry, those variations are described by Jacobi fields, which characterize how geodesics spread apart on a Riemannian manifold. We

use this framework to establish the connection between the vanilla and variational loss functions. Specifically, we (1) define a Jacobi field formulation of the RFM and RG-VFM losses in section 4.1, (2) derive the relation between these field-specific instances (proposition 4.2), and (3) eventually establish the corresponding loss relationships in proposition 4.3.

## 4.1 Jacobi Field formulation of the Flow Matching Objectives

We consider a smooth family of geodesics $\{\gamma_s\}$ all starting from the same point $\gamma_s(0) := x_0 \in \mathcal{M}$, and determined by an initial velocity of the form $\dot{\gamma}_s(0) = v^s := v^0 + sw$, with $v^0, w \in T_{x_0}\mathcal{M}$, where $sw$ represents the perturbation level. A schematic representation is in fig. 2 (a).

Each geodesic $(s, \tau) \rightarrow \gamma_s(\tau)$ is parametrized by two variables: $s \in [0,1]$ which indexes the perturbation of its initial velocity, and $\tau \in [0,1]$, the parameter along one geodesic that links the initial point $\gamma_s(0) = x_0$ to the endpoint $\gamma_s(1) = x_1^s$. For convenience, we denote $\alpha(s, \tau) := \gamma_s(\tau)$ the two-parameter map which simultaneously describes the entire family of perturbed geodesics.

**Definition 4.1** (Jacobi field at a vanishing starting point). *The family of geodesics defined as:*

$$\alpha(s, \tau) := \gamma_s : \tau \rightarrow exp_{x_0}(\tau(v + sw)),$$

*with $s \in [0,1]$ and $\tau \in [0,1]$, $v, w \in T_{x_0}\mathcal{M}$, is a smooth family of shooting geodesics with $\gamma_s(0) = x_0$, $\dot{\gamma}_0(0) = v$ and $\dot{\gamma}_1(0) = v + w$.*

*For each fixed $\tau \in [0,1]$, there exists a vector field, called **Jacobi field**,*

$$J(\tau) := \partial_s \alpha(s, \tau)\big|_{s=0}$$

*along the geodesic $\gamma_s(\tau) := \alpha(s, \tau)$ satisfying the ODE equation: $D_\tau^2 J + R(J, \dot{\gamma}_s)\dot{\gamma}_s = 0$, with $R$ the Riemannian curvature tensor of the manifold. In particular, this Jacobi field is uniquely defined by the initial conditions and at $\tau = 0$ one has the initial conditions: $J(0) = 0$, and $D_\tau J(0) = w$.*

Borrowing the notations from (Chen & Lipman, 2024), we denote the target velocity $v^0 = u_t(x \mid x_1)$, the predicted velocity $v^1 = v_t^\theta(x)$, and their respective endpoints $\gamma_0(1) = x_1$ and $\gamma_1(1) = \mu_t^\theta(x)$. The losses can be formulated in the Jacobi field framework with the following:

**Proposition 4.1** (Loss functions as evaluation of Jacobi fields). *Consider a Jacobi field $J(\tau) := \partial_s \alpha(s, \tau)\big|_{s=0}$ as defined in definition 4.1. We denote $\mathcal{L}_{\text{RFM}}$ the loss function of the (vanilla) Riemannian Flow Matching (Chen & Lipman, 2024), and $\mathcal{L}_{\text{RG-VFM}}$ the loss function for our proposed Riemannian Variational Flow Matching. Then the following equalities hold:*

$$\mathcal{L}_{\text{RFM}}(\theta) = \mathbb{E}_{t,x_1,x}\big[\|u_t(x \mid x_1) - v_t^\theta(x)\|_{\mathbf{g}}^2\big] = \mathbb{E}_{t,x_1,x}\big[\|D_\tau J(0)\|_{\mathbf{g}}^2\big], \tag{10}$$

$$\mathcal{L}_{\text{RG-VFM}}(\theta) = \mathbb{E}_{t,x_1,x}\big[\|\log_{x_1}(\mu_t^\theta(x))\|_{\mathbf{g}}^2\big] = \mathbb{E}_{t,x_1,x}\big[\|J(1)_{\mathbf{g}}\|^2\big]. \tag{11}$$

## 4.2 Relation between RG-VFM and RFM Objectives

Now that we have expressed the losses through the Jacobi fields, we observe that $\mathcal{L}_{\text{RFM}}$ is a first-order approximation of $\mathcal{L}_{\text{RG-VFM}}$ through the following proposition:

**Proposition 4.2.** *$D_\tau J(0)$ is a linear approximation of $J(1)$.*

The proof essentially consists of deriving the Taylor expansion of $J(\tau)$, centered at $\tau = 0$ and evaluated at $\tau = 1$, and identifying $D_\tau J(0)$ as the linear term. By truncating at the linear approximation, curvature information is absent from $D_\tau J(0)$ but remains implicitly encoded in $J(1)$. This distinction directly affects the relationship between the RFM and RG-VFM losses: while they coincide in Euclidean space, their difference in curved spaces is generally nonzero and curvature-dependent.

**Euclidean case.** In Euclidean space, the Taylor expansion reduces to the linear term: $J(\tau) = J(0) + \tau D_\tau J(0)$ which, for $\tau = 1$ and $J(0) = 0$, leads to $J(1) = D_\tau J(0)$. As a consequence,

$$\mathbb{E}_{t,x_1,x}\big[\|D_\tau J(0)\|_2^2\big] = \mathbb{E}_{t,x_1,x}\big[\|J(1)\|_2^2\big] \tag{12}$$

which confirms that $\mathcal{L}_{\text{CFM}}$ and $\mathcal{L}_{\text{VFM}}$ can be reduced to one another, with proper normalization terms.

More generally, the two losses differ by a curvature-dependent term on non-flat manifolds, as shown in the following result as a direct consequence of proposition 4.1 and proposition 4.2:

**Proposition 4.3** (Difference of loss functions as a curvature term). *Consider a Jacobi field $J(\tau) := \partial_s \alpha(s, \tau)\big|_{s=0}$ as defined in definition 4.1 and the equivalences shown in proposition 4.1. The difference between $\mathcal{L}_{\text{RG-VFM}}$ and $\mathcal{L}_{\text{RFM}}$ encodes the manifold curvature through:*

$$\mathcal{L}_{\text{RG-VFM}}(\theta) = \mathcal{L}_{\text{RFM}}(\theta) + \underbrace{\mathbb{E}_{t,x_1,x}[\mathcal{C}(R, D_\tau J(0), v) + \mathcal{E}_{higher}]}_{\text{curvature-dependent term}} \tag{13}$$

*where the leading-order curvature functional is:*

$$\mathcal{C}(R, D_\tau J(0), v) = -\frac{1}{3}\langle R(D_\tau J(0), v)v, D_\tau J(0)\rangle_{\mathbf{g}} - \frac{1}{6}\langle (\nabla_v R)(D_\tau J(0), v)v, D_\tau J(0)\rangle_{\mathbf{g}} \tag{14}$$

$$and \quad \mathcal{E}_{higher} = O(\|D_\tau J(0)\|^2 \|v\|^3), \tag{15}$$

*with $R$ the Riemannian curvature tensor and $v = \dot{\gamma}_0$ the reference geodesic velocity. The higher-order term $\mathcal{E}_{higher}$ encodes curvature variation along geodesics through covariant derivatives of $R$. In terms of the RFM loss terms, $v = u_t(x \mid x_1)$ and $D_\tau J(0) = v_t^\theta(x) - u_t(x \mid x_1)$.*

**Geometric interpretation.** The curvature functional $\mathcal{C}$ captures how the manifold's geometry affects the loss comparison, encoding the first- and second-order effects of curvature on geodesic deviation. Thus, RG-VFM *implicitly* captures the full geometric structure through the exact Jacobi field $J(1)$, while RFM uses only the linear approximation $D_\tau J(0)$. This lack of curvature information results in weaker, less precise supervision in directing the flow toward the actual endpoint, leading in practice to RG-VFM learning the signal more effectively than RFM. Special cases are:

- In Euclidean space, $R = 0$ implies both $\mathcal{C} = 0$ and $\mathcal{E}_{\text{higher}} = 0$. This leads to $\mathcal{L}_{\text{RG-VFM}} = \mathcal{L}_{\text{VFM}} = \mathcal{L}_{\text{CFM}} = \mathcal{L}_{\text{RFM}}$ as expected from eq. (12).
- In spaces of constant curvature (e.g. hyperspheres or hyperbolic spaces) $\nabla R = 0$. In this setting, we can restate the result of proposition 4.3 in terms of the constant sectional curvature $K$. The formulation and proof are given in corollary D.1, and in the experimental section we focus primarily on manifolds that fall within this category.

In summary, we introduced RG-VFM as an alternative to RFM for learning a velocity field on a manifold, providing a variational formulation whose objective fully captures higher-order curvature effects, unlike RFM. This results in generally different objectives on curved manifolds. In Euclidean space, however, the RFM objective reduces to CFM, while RG-VFM reduces to VFM (assuming a Euclidean Gaussian posterior rather than Riemannian), and the CFM and VFM objectives become equivalent under appropriate normalization. These relations are schematized in fig. 1 and fig. 2(b), and their schematic algorithms can be compared in section D.2. In terms of computational costs, extrinsic RG-VFM has the same complexity as VFM during both training and sampling. The only difference between the two methods is that VFM compares endpoints using Euclidean distance, while we use geodesic distance. Since we assume geodesic distance to be in closed-form, this introduces no additional computational overhead compared to VFM. Similarly, implicit RG-VFM maintains the same complexity as RFM, with the main difference being that velocity computation happens during sampling rather than during training (see algorithms 3 to 8).

## 5 EXPERIMENTS

**Goal of the experiments.** The goal of our experiments is twofold. First, we aim to observe the practical implications of proposition 4.3 by studying the behavior of vanilla and variational models, both Euclidean and Riemannian, in a controlled synthetic setting with a visually precise target distribution. Second, we conduct real-world experiments on MOF and protein backbone generation, motivated by a gap in the literature.

**Motivation for material and protein generation and common pattern.** Existing works on protein and material generation – often based on diffusion- and flow-based models with structural losses inspired by Yim et al. (2023b;a), such as Yue et al. (2025); Kim et al. (2024); Guo et al. (2025) – follow a common pattern. Their generation scheme is split between Euclidean and non-Euclidean parameters, where Euclidean parameters are learned through a process that effectively corresponds to variational flow matching, since the model predicts endpoints, minimizes MSE with the target,

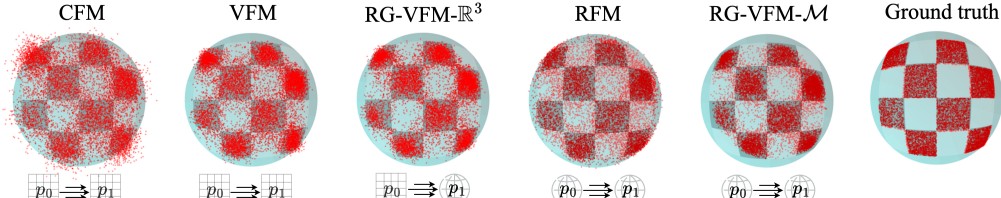

Figure 3: Comparison of the spherical checkerboard distribution generated with CFM, VFM, RFM and our methods RG-VFM-$\mathbb{R}^3$ and RG-VFM-$\mathcal{M}$.

and uses these predictions to compute velocity fields during integration. Non-Euclidean parameters instead employ a partially but not fully variational form of Riemannian FM: endpoints are still predicted, but the loss minimizes the squared distance between ground-truth and predicted velocities, with the latter obtained via the logarithm map of the manifold. *This reveals a room for improvement, as full alignment of the loss components would suggest minimizing the geodesic distance between predicted and target data points in the non-Euclidean case.* Our method directly explores this option. Furthermore, prior works report that endpoint learning improved empirical performance, and we interpret our approach, together with Eijkelboom et al. (2024), as providing complementary theoretical justification for this choice. In this setting, we choose to *variationalize* the losses of two models from distinct applications: MOFFlow (Kim et al., 2024) for MOF generation and ReQFlow (Yue et al., 2025) for protein backbone generation.

### 5.1 Curvature Effects in Synthetic Data

**Dataset and experimental setup.** Inspired by the planar checkerboard benchmark in generative modeling (Grathwohl et al., 2018), we introduce two curved checkerboard distributions as our target $p_1$, whose support is either on the hypersphere $\mathbb{S}^2 \subset \mathbb{R}^3$ or the upper-sheet hyperboloid $\mathbb{H}^2_{-1} \subset \mathbb{R}^3$, which we define in section E.1. The noisy distribution $p_0$ is defined differently for each model: for CFM, VFM, and RG-VFM-$\mathbb{R}^3$, $p_0$ is the standard normal distribution in $\mathbb{R}^3$, while for RG-VFM-$\mathcal{M}$ and RFM, it is obtained by wrapping the standard normal distribution on either $\mathbb{S}^2$ or $\mathbb{H}^2_{-1}$.

We conduct two sets of experiments: we (1) compare the extrinsic models in their ability to capture the correct geometry – assessed by the distance of the generated samples to the data manifold – and (2) evaluate vanilla versus variational models in reproducing the target distribution. For this last point, the evaluation is based on metrics such as Coverage (% of generated points falling within the desired checkerboard area) and Classifier 2-Sample Tests (C2ST) metric (Lopez-Paz & Oquab, 2016; Dalmasso et al., 2020; Lueckmann et al., 2021). The C2ST technique utilizes a neural network classifier to separate true samples from generated ones, where a score of 0.5 indicates the distributions are indistinguishable to the classifier, while scores approaching 1 suggest the distributions are easily separable. Additional experimental details are provided in section E.

**Results.** We observe that (1) Riemannian models better capture manifold geometry by generating points with minimal distance to the manifold compared to Euclidean ones (see Distance columns in table 1), and (2) variational models produce sharper and less blurred distributions than vanilla models, with RG-VFM-$\mathbb{R}^3$ and RG-VFM-$\mathcal{M}$ showing the best visual performance in fig. 3. This is reflected in table 1 in Coverage metric results, where variational models – particularly Riemannian ones – achieve the highest values. For C2ST, no consistent pattern emerges between spherical and hyperbolic cases, except that in both cases standard VFM demonstrates the strongest performance. In essence, emphasizing endpoint accuracy enables variational models to capture fine details of the target distribution's shape, and additional geometric awareness of RG-VFM further enhances the result. We tested setting $\sigma_t(x) = 1$ versus $\sigma_t(x) = 1 - t$ in eq. (8) during training and found negligible differences, so we report only the results obtained with $\sigma_t(x) = 1$. Additionally, preliminary findings suggest that using $L^1$ loss (which corresponds to using a Riemannian Laplace instead of a Riemannian Gaussian distribution in eq. (8)) instead of $L^2$ may enhance performance, particularly in hyperbolic spaces, and we start exploring this option in section E.4.

### 5.2 MOF Generation with MOFFlow

**Dataset and experimental setup: from MOFFlow to V-MOFFlow.** MOFFlow (Kim et al., 2024) is a flow-based generative model for MOF structures operating on rigid building blocks. A

Table 1: **Results of synthetic experiments.** Clipping is applied in the variational setting to stabilize sampling. The distance between generated points and the ground-truth manifold is computed only for extrinsic models since intrinsic ones generate directly on the manifold. Abbreviations: Eucl. = Euclidean, Riem. = Riemannian, Ext. = extrinsic, Int. = intrinsic, Van. = vanilla, Var. = variational.

| | Sphere | | | Hyperboloid | | |
|---|---|---|---|---|---|---|
| | Coverage ↑ | C2ST↓ | Distance↓ | Coverage ↑ | C2ST↓ | Distance↓ |
| Eucl./Ext./Van. (CFM) | 64.97 | 58.36 ± 1.56 | 0.012 ± 0.099 | 69.05 | 57.38 ± 1.30 | 0.008 ± 0.339 |
| Eucl./Ext./Var. (VFM) | 79.08 | **56.33 ± 0.48** | 0.044 ± 0.045 | 75.89 | **57.03 ± 0.59** | 0.061 ± 0.140 |
| Riem./Ext./Var. (Ours) | 83.10 | 56.58 ± 0.28 | **0.010 ± 0.035** | **78.84** | 63.55 ± 0.35 | **0.021 ± 0.056** |
| Riem./Int./Van. (RFM) | 66.83 | 57.99 ± 0.58 | - | 60.75 | 61.66 ± 0.92 | - |
| Riem./Int./Var. (Ours) | **84.21** | 59.72 ± 0.87 | - | 68.38 | 59.73 ± 0.31 | - |

MOF is represented as $S = (\mathcal{B}, q, \tau, \ell)$, where $\mathcal{B}$ denotes building blocks, and the model learns their roto-translations $(q, \tau)$ and lattice parameter $\ell$. The conditional normalizing flow $p_\theta(q, \tau, \ell \mid \mathcal{B})$ uses a re-parameterized training objective predicting clean data $(q_1, \tau_1, \ell_1)$ from intermediate structure $S^{(t)}$. The Euclidean loss minimizes endpoint $L^2$ distance following VFM, while the rotational part computes conditional velocities from predictions and minimizes squared distance to ground-truth velocities, as in RFM. Our contribution makes MOFFlow fully variational by applying our method to its rotational component. A detailed explanation with loss equations is in section F.1.

We evaluate the resulting model, Variational-MOFFlow (V-MOFFlow) against the original MOF-Flow and DiffCSP (Jiao et al., 2023) on MOF structure prediction using the large-scale dataset of Boyd et al. (2019), where structures are decomposed into building blocks and split into train/validation/test set. We follow the experimental setup of Kim et al. (2024), and performance is measured by match rate (MR) and RMSE between original structures and generated samples.

**Results in structure prediction.** We report results in table 2. Our model outperforms all competitors except for MR at stol $= 1.0$, which Kim et al. (2024) consider too lenient for practical use. This validates our theoretical findings that RG-VFM loss guides training more effectively than RVM. We report additional analyses and experimental details in section F.

Table 2: **Structure prediction accuracy.** We report results for DiffCSP and MOFFlow with `TimeBatch` implementation from Kim et al. (2024), and we reproduce MOFFlow and evaluate V-MOFFlow with `Batch` implementation. "stol" is the site-tolerance for matching criteria.

| | # of samples | stol = 0.5 | | stol = 1.0 | |
|---|---|---|---|---|---|
| | | MR (%)↑ | RMSE↓ | MR (%)↑ | RMSE↓ |
| DiffCSP | 1 | 0.09 | 0.3961 | 23.12 | 0.8294 |
| | 5 | 0.34 | 0.3848 | 38.94 | 0.7937 |
| MOFFlow (Paper results) | 1 | 31.69 | 0.2820 | 87.46 | 0.5183 |
| | 5 | 44.75 | 0.2694 | **100.0** | 0.4645 |
| MOFFlow (Reproduced) | 1 | 30.40 | 0.2832 | 83.50 | 0.5255 |
| | 5 | 46.97 | 0.2717 | 95.82 | 0.4603 |
| V-MOFFlow (Ours) | 1 | **33.52** | **0.2789** | **89.08** | **0.5096** |
| | 5 | **50.14** | **0.2629** | 97.18 | **0.4384** |

## 5.3 PROTEIN BACKBONE GENERATION WITH V-REQFLOW

**Dataset and experimental setup: from QFlow & ReQFlow to V-QFlow and V-ReQFlow.** QFlow (Yue et al., 2025) is a flow-based model for protein backbone generation. Unlike previous methods (Yim et al., 2023b; Bose et al., 2023) that represent $SO(3)$ elements with rotation matrices, QFlow uses quaternions, which provide improved training stability. Building on this foundation, ReQFlow (Yue et al., 2025) further enhances QFlow by incorporating rectified flow with re-paired samples and noise, inspired by Liu et al. (2023), improving the designability of generated protein backbone structures. Similar to MOF structure generation, protein backbone structures are represented as sequences of $SE(3)$ elements $\{q_i, t_i\}^N$, where $q_i \in SO(3)$ defines the frame on the

$\alpha$-carbon of each amino acid, and $t_i \in \mathbb{R}^3$ represents the zero-mean coordinate of the $\alpha$-carbon. For further details, see Yim et al. (2023b). The goal is to learn a conditional flow $p_\theta(Q, T|N)$ where $Q = \{q_i\}^N$ and $T = \{t_i\}^N$, with $N$ denoting the number of residues in the desired backbone structure. Like MOFFlow, both QFlow and ReQFlow employ a re-parametrized training objective that predicts endpoints from which vector fields are reconstructed. We apply our method by *variationalizing* the rotational component of their loss function, similarly to section F.1, while maintaining all other implementation details identical to isolate the benefits of our variational objective.

We tested Variational-QFlow (V-QFlow) and Variational-ReQFlow (V-ReQFlow) on filtered Protein Data Bank (Berman et al., 2000) dataset with 23366 protein structure with lengths ranging from 60 to 512. The filtering pipeline follows Yue et al. (2025). For evaluation metrics, we follow Yue et al. (2025), using designability, diversity and novelty to concretely evaluate the quality of the generated protein backbone structures. We trained our V-QFlow with 4 NVIDIA-H100 GPUs for around 260 epochs. For V-ReQFlow, we further finetuned it on our rectified dataset for 10 epochs.

**Results in protein backbone structure generation.** From table 3, we observe that V-QFlow and V-ReQFlow surpass their vanilla counterparts on both designability and folding RMSD, emphasizing the effectiveness of applying variational objectives when learning probability paths on manifolds.

Table 3: **Performance comparison with baseline models on protein backbone generation on PDB dataset.** 50 samples are generated and evaluated for each length in $\{50, 100, 150, 200, 250, 300\}$. For both ReQFlow and V-ReQFlow, we generate the rectified dataset with 20 samples for each length in $[60, 512]$. We filter the generated samples following the procedures in the repo provided by Yue et al. (2025). Samples used for evaluation are generated by flow models trained with 10 epochs on the rectified dataset for both ReQFlow and V-ReQFlow.

| | Efficiency | Designability | | Diversity | Novelty |
|---|---|---|---|---|---|
| | Step | Fraction↑ | scRMSD↓ | TM↓ | TM↓ |
| RFDiffusion | 50 | 0.904 | $1.102_{\pm 1.617}$ | 0.382 | 0.527 |
| Genie2 | 1000 | 0.908 | $1.132_{\pm 1.389}$ | 0.370 | 0.475 |
| FoldFlow2 | 50 | 0.952 | $1.083_{\pm 1.308}$ | 0.373 | **0.527** |
| FrameFlow | 500 | 0.872 | $1.380_{\pm 1.392}$ | **0.346** | 0.562 |
| QFlow (Reproduced) | 500 | 0.924 | $1.252_{\pm 1.302}$ | 0.357 | 0.641 |
| QFlow (Paper results) | 500 | 0.936 | $1.163_{\pm 0.938}$ | 0.356 | 0.635 |
| V-QFlow (Ours) | 500 | 0.968 | $\mathbf{0.923}_{\pm 0.787}$ | 0.387 | 0.647 |
| ReQFlow (Reproduced) | 500 | 0.964 | $0.939_{\pm 0.572}$ | 0.400 | 0.630 |
| ReQFlow (Paper results) | 500 | 0.972 | $1.071_{\pm 0.482}$ | 0.377 | 0.645 |
| V-ReQFlow (Ours) | 500 | **0.980** | $0.961_{\pm 0.832}$ | 0.408 | 0.644 |

# 6 CONCLUSION

We introduce Riemannian Gaussian Variational Flow Matching, which extends VFM to general manifolds through Riemannian Gaussian distributions, unifying RFM and VFM under a common probabilistic framework. Through a reformulation of their objectives using Jacobi vector fields, we demonstrate that RG-VFM captures richer curvature-dependent information compared to standard RFM. In our experiments, we validate that this theoretical advantage translates to more precise supervision and better learned signals: (1) for synthetic spherical and hyperbolic checkerboard distributions, enhanced curvature awareness leads to improved sharpness in learned distributions, and (2) for real-world protein backbone and material generation tasks, applying our variational perspective through a simple modification to the rotational component of existing flow matching losses consistently improves generation quality metrics. A current limitation is that our method is defined for simple geometries with closed-form geodesics. However, most practical tasks involve manifolds with explicit exponential and logarithmic maps, and we believe this framework can be straightforwardly extended to more complex geometries. These results establish RG-VFM as a promising approach for modeling distributions on complex geometries with minimal implementation overhead.

ETHICS STATEMENT

This work aims to advance machine learning and AI for science. Material and protein generation hold great promise for driving scientific discovery and tackling global challenges in medicine, sustainability, and biotechnology. At the same time, the technology raises ethical considerations, including the need for appropriate regulatory oversight as it matures. In terms of readiness, this work remains at an early stage, focusing on foundational computational methods rather than immediate applications, and therefore presents no direct benefits or risks at this time.

REPRODUCIBILITY STATEMENT

To ensure reproducibility and completeness, all required notation, mathematical background, definitions, and proofs of mathematical statements are provided in sections B to D. Experimental and implementation details are included in section 5 and sections E and F.

## 7 ACKNOWLEDGMENTS

This project was funded by the CaLiForNIA project (Marie Skłodowska-Curie Actions Doctoral Network 2022), the Bosch Center for Artificial Intelligence, and the Finnish Center for Artificial Intelligence (FCAI). This publication is also part of the project SIGN with file number VI.Vidi.233.220 of the research programme Vidi which is (partly) financed by the Dutch Research Council (NWO) under the grant https://doi.org/10.61686/PKQGZ71565. The author C.L. was financially supported by Health-Holland, Top Sector Life Sciences & Health (LSH-TKI), project number LSHM22023, which realizes a public-private partnership (PPP) between the University of Amsterdam and Janssen Vaccines & Prevention B.V. We thank SURF (www.surf.nl) for the support in using the National Supercomputer Snellius.

O.Z. would like to thank David Wessels for his great help and advice on the coding and engineering aspects, Nayoung Kim for the valuable conversations about MOFFlow that got her interested in the real-world applications considered here, and Kiyoung Seong for making these conversations happen. O.Z. and A.P. are grateful to their co-authors for the amazing collaboration. Additionally, the authors are indebted to Tullio Levi-Civita for giving the Jacobi equation its modern geometric formulation in terms of curvature.

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

## A    DISCLOSURE OF LLM USAGE

We declare that the use of LLMs for writing this paper was limited to general-purpose writing assistance. Specifically, we used them only to polish the wording of text sections and in no way to generate the research ideas or technical results and proofs presented in this paper.

## B    NOTATIONS

In this section, we report the notations that are used in the paper and the rest of the appendix, summarized in table 4.

| Symbol | Name | Type | Description |
|--------|------|------|-------------|
| $\mathcal{M}$ | manifold | object | Smooth Riemannian manifold $(\mathcal{M}, \mathbf{g})$. |
| $\mathbf{g}$ | metric | tensor | Riemannian metric; $\langle \cdot, \cdot \rangle = \mathbf{g}(\cdot, \cdot)$ and $\lvert \cdot \rvert = \sqrt{\mathbf{g}(\cdot, \cdot)}$. |
| $p$ | base point | point | Fixed point in $\mathcal{M}$; normal coordinates are taken at $p$. |
| $T_p\mathcal{M}$ | tangent space | vector space | Tangent space at $p$; all $v, w, \delta, u_s$ live here. |
| $\exp_p$ | exponential map | map | $\exp_p : T_p\mathcal{M} \supset U \to \mathcal{M}$, a diffeomorphism on a small ball $U$. |
| $\langle \cdot, \cdot \rangle_{\mathbf{g}}$ | inner product | scalar | Inner product on $(\mathcal{M}, \mathbf{g})$. |
| $\mathrm{dist}_{\mathbf{g}}(\cdot, \cdot)$ | distance | scalar | Riemannian distance on $(\mathcal{M}, \mathbf{g})$. |
| $R$ | curvature tensor | tensor | $(1, 3)$-tensor $R(X, Y)Z = \nabla_X \nabla_Y Z - \nabla_Y \nabla_X Z - \nabla_{[X,Y]}Z$. |
| $\nabla$ | Levi–Civita connection | operator | Metric, torsion-free connection; $D_t$ denotes covariant derivative along a curve. |
| $K$ | sectional curvature | scalar | Constant curvature in space forms; for a sphere of radius $r$, $K = 1/r^2$. |
| $v, w$ | tangent vectors | vectors | Elements of $T_p\mathcal{M}$; initial velocities of the two geodesics. |
| $\tau, s$ | parameters | scalars | $\tau$ is the geodesic time (small); $s \in [-\varepsilon, \varepsilon]$ parametrizes the variation within the family of geodesics. |
| $\mathbb{S}^2$ | 2-sphere | object | $\mathbb{S}^2 := \{x \in \mathbb{R}^3 : \langle x, x \rangle_E = 1\}$ |
| $\mathbb{H}^2_{-1}$ | 2-hyperboloid | object | $\mathbb{H}^2_K := \{x \in \mathbb{R}^3 : \langle x, x \rangle_{\mathcal{L}} = -1, x_0 > 0\}$ |
| $\gamma_s(\tau)$ | geodesic | curve | $\gamma_s(\tau) := \exp_p(\tau u_s)$, geodesic with initial velocity $u_s$ at $p$. |
| $\alpha(s, \tau)$ | ruled surface | 2-parameter map | $\alpha(s, \tau) := \gamma_s(\tau)$; two-parameter family used for variations. |
| $J(\tau)$ | Jacobi field | vector field | $J(\tau) := \partial_s \alpha(s, \tau)$ along $\gamma_s$; $J(0) = 0$, $(D_\tau J)(0) = \delta$. |
| $O(\cdot)$ | remainder | notation | Big–O with constants uniform for $v, w$ in a fixed small ball in $T_p\mathcal{M}$. |

Table 4: Notations of objects mentioned in this paper.

## C    GEOMETRIC BACKGROUND

### C.1    RIEMANNIAN MANIFOLDS

In this section, we provide a comprehensive introduction to Riemannian manifolds, establishing all necessary definitions from first principles.

**Basic definitions.**    A manifold $\mathcal{M}$ is a mathematical structure that appears curved globally but looks flat when viewed locally. Formally, a $d$-dimensional manifold can be covered by coordinate charts, where each chart provides a local parameterization. For any point $p \in \mathcal{M}$, there exists a neighborhood that can be mapped smoothly to an open subset of $\mathbb{R}^d$ via coordinate charts.

The tangent space $T_p\mathcal{M}$ at a point $p \in \mathcal{M}$ represents the collection of all possible directions one can move from $p$ while staying on the manifold. This vector space encodes the local linear approximation to the manifold at $p$ and maintains the same dimensionality as the ambient manifold.

**Riemannian metric.** A Riemannian metric $\mathbf{g}$ on $\mathcal{M}$ is a smoothly varying collection of inner products, one for each tangent space. Specifically, for each point $p \in \mathcal{M}$, the metric $\mathbf{g}$ defines an inner product $\langle \cdot, \cdot \rangle_{\mathbf{g}}$ on the tangent space $T_p\mathcal{M}$. This inner product must be:

- *Bilinear*: $\langle av + bw, u \rangle_{\mathbf{g}} = a\langle v, u \rangle_{\mathbf{g}} + b\langle w, u \rangle_{\mathbf{g}}$ for tangent vectors $v, w, u \in T_p\mathcal{M}$ and scalars $a, b$,

- *Symmetric*: $\langle v, w \rangle_{\mathbf{g}} = \langle w, v \rangle_{\mathbf{g}}$,

- *Positive definite*: $\langle v, v \rangle_{\mathbf{g}} > 0$ for all non-zero $v \in T_p\mathcal{M}$.

A manifold $\mathcal{M}$ equipped with a Riemannian metric $\mathbf{g}$ is called a *Riemannian manifold* and is denoted by $(\mathcal{M}, \mathbf{g})$.

The metric enables us to measure lengths of tangent vectors and angles between them. For tangent vectors $v, w \in T_p\mathcal{M}$, their lengths are $\|v\|_{\mathbf{g}} = \sqrt{\langle v, v \rangle_{\mathbf{g}}}$ and $\|w\|_{\mathbf{g}} = \sqrt{\langle w, w \rangle_{\mathbf{g}}}$, respectively.

**Geodesics.** Geodesics are the natural generalization of straight lines to curved spaces. On a Riemannian manifold, a geodesic $\gamma_s(\tau)$ is a curve that maintains constant "speed" and "direction" in the sense defined by the Riemannian metric. Mathematically, geodesics are characterized by having vanishing covariant acceleration.

These curves play a fundamental role as they represent paths of extremal length between nearby points. Given any point $p \in \mathcal{M}$ and initial tangent vector $v \in T_p\mathcal{M}$, there exists a unique geodesic originating at $p$ with initial direction $v$.

**Distance function.** The Riemannian metric induces a natural distance function on the manifold. The Riemannian distance $\text{dist}_{\mathbf{g}}(p, q)$ between two points $p, q \in \mathcal{M}$ is defined as the infimum of the lengths of all piecewise smooth curves connecting $p$ and $q$:

$$\text{dist}_{\mathbf{g}}(p, q) = \inf_{\gamma} \int_0^1 \left\| \frac{d\gamma}{dt}(t) \right\|_{\mathbf{g}} dt \tag{16}$$

where the infimum is taken over all piecewise smooth curves $\gamma : [0, 1] \to \mathcal{M}$ with $\gamma(0) = p$ and $\gamma(1) = q$. Under appropriate completeness conditions, this distance is achieved by geodesics.

**Exponential map.** The exponential map $\exp_p : T_p\mathcal{M} \to \mathcal{M}$ provides a canonical way to translate between the linear tangent space and the curved manifold. For a tangent vector $v \in T_p\mathcal{M}$, the exponential map is defined as:

$$\exp_p(v) = \gamma_v(1) \tag{17}$$

where $\gamma_v(\tau)$ represents the geodesic initiating at $p$ with velocity $v$, evaluated at parameter value $\tau = 1$. This construction allows us to "walk" along geodesics to reach new points on the manifold.

In sufficiently small neighborhoods around any point $p$, the exponential map establishes a smooth bijection between a region in the tangent space and a region on the manifold.

**Logarithmic map.** The logarithmic map $\log_p : \mathcal{M} \to T_p\mathcal{M}$ is the (local) inverse of the exponential map. For a point $q \in \mathcal{M}$ sufficiently close to $p$, the logarithmic map returns the tangent vector $v \in T_p\mathcal{M}$ such that $\exp_p(v) = q$.

In regions where the exponential map is a diffeomorphism, we have $\log_p(\exp_p(v)) = v$ and $\exp_p(\log_p(q)) = q$. The logarithmic map essentially tells us which direction and how far to travel in the tangent space to reach a given nearby point on the manifold.

In this work, we consider complete, connected, and smooth Riemannian manifolds $(\mathcal{M}, \mathbf{g})$, ensuring that geodesics can be extended indefinitely and that the exponential map is well-defined globally.

**Tangent bundle.** By collecting all tangent spaces across the manifold, we obtain the tangent bundle:

$$T\mathcal{M} = \bigcup_{p \in \mathcal{M}} \{p\} \times T_p\mathcal{M} \tag{18}$$

The tangent bundle is itself a smooth manifold of dimension $2d$, where $d$ is the dimension of $\mathcal{M}$. Each element of $T\mathcal{M}$ can be written as $(p, v)$ where $p \in \mathcal{M}$ is a point on the manifold and $v \in T_p\mathcal{M}$ is a tangent vector at that point.

**Vector fields.** A vector field on $\mathcal{M}$ is a smooth section of the tangent bundle, i.e., a smooth map $u : \mathcal{M} \rightarrow T\mathcal{M}$ such that $u(p) \in T_p\mathcal{M}$ for each point $p \in \mathcal{M}$. In local coordinates, a vector field can be expressed as $u = \sum_{i=1}^{d} u^i \frac{\partial}{\partial x^i}$ where the coefficient functions $u^i$ are smooth. We specifically consider *time-dependent vector fields* $\{u_t\}_{t \in I}$, which are smooth families of vector fields parameterized by time $t$. The Riemannian metric $\mathbf{g}$ extends naturally to define pointwise inner products between vector fields: $\langle u, w \rangle_{\mathbf{g}}(p) = \langle u(p), w(p) \rangle_{\mathbf{g}}$ for any two vector fields $u$ and $w$.

**Homogeneous Manifold.** A Riemannian manifold $\mathcal{M}$ is homogeneous if its isometry group acts transitively on $\mathcal{M}$, i.e., for any two points $x, y \in \mathcal{M}$, there exists an isometry $f : \mathcal{M} \rightarrow \mathcal{M}$ such that $f(x) = y$.

## C.2 RIEMANNIAN GAUSSIAN DISTRIBUTIONS

We describe the construction of the Riemannian Gaussian (RG) distribution, which generalizes the familiar Gaussian distribution to the setting of a Riemannian manifold. The definition of the Riemannian Gaussian is a specific instance of the Normal law presented in Pennec (2006):

**Definition C.1** (Normal law Pennec (2006)). *We call Normal law on the manifold $\mathcal{M}$ the maximum–entropy distribution specified by its mean value and covariance. Assuming no continuity or differentiability constraint on the cut locus $C(\bar{x})$ and a symmetric domain $D(\bar{x})$, the probability density function of the Normal law with mean $\bar{x}$ and concentration matrix $\Gamma$ is*

$$N_{(\bar{x}, \Gamma)}(y) = k \, \exp\!\left(-\tfrac{1}{2} \overrightarrow{\bar{x}y}^{\top} \Gamma \overrightarrow{\bar{x}y}\right), \tag{19}$$

*where the normalisation constant $k$ and the covariance $\Sigma$ are related to $\Gamma$ by*

$$k^{-1} = \int_{\mathcal{M}} \exp\!\left(-\tfrac{1}{2} \overrightarrow{\bar{x}y}^{\top} \Gamma \overrightarrow{\bar{x}y}\right) d\mathcal{M}(y), \qquad \Sigma = k \int_{\mathcal{M}} \overrightarrow{\bar{x}y}\, \overrightarrow{\bar{x}y}^{\top} \exp\!\left(-\tfrac{1}{2} \overrightarrow{\bar{x}y}^{\top} \Gamma \overrightarrow{\bar{x}y}\right) d\mathcal{M}(y). \tag{20}$$

By simply defining the concentration matrix $\Gamma$ as $\frac{\mathbf{G}}{\sigma}$, where $\mathbf{G}$ is the metric tensor associated with the chosen metric and $\sigma$ is a fixed variance parameter, we obtain the following definition.

**Definition C.2** (Riemannian Gaussian). *Let $\mathcal{M}$ be a Riemannian manifold endowed with the metric tensor $\mathbf{g}$. The RG distribution is defined by*

$$\mathcal{N}_{Riem}(z \mid \sigma, \mu) = \frac{1}{C} \exp\!\left(-\frac{dist_{\mathbf{g}}(z, \mu)^2}{2\sigma^2}\right), \tag{21}$$

*where $z \in \mathcal{M}$ is a point on the manifold, $\mu \in \mathcal{M}$ plays the role of the mean, and $\sigma > 0$ is a scale parameter controlling the spread of the distribution. Here, $dist_{\mathbf{g}}(z, \mu)$ denotes the geodesic distance between $z$ and $\mu$ as determined by the metric $\mathbf{g}$, and $C$ is a normalization constant chosen so that the total probability integrates to 1 over $\mathcal{M}$:*

$$C = \int_{\mathcal{M}} \exp\!\left(-\frac{dist_{\mathbf{g}}(z, \mu)^2}{2\sigma^2}\right) d\mathcal{M}_z. \tag{22}$$

*The measure $d\mathcal{M}_z$ represents the Riemannian volume element, which in local coordinates takes the form*

$$d\mathcal{M}_z = \sqrt{\det \mathbf{g}(z)} \, dz, \tag{23}$$

*with $dz$ being the standard Lebesgue measure in the coordinate chart and $\mathbf{g}(z)$ is the Riemannian metric tensor at the point $z$. This formulation ensures that the probability density is adapted to the geometric structure of the manifold.*

**Observation.** In the special case where $\mathcal{M} = \mathbb{R}^d$ and the metric is Euclidean (i.e., $\mathbf{g}(z) = \mathbf{I}$), the geodesic distance reduces to the usual Euclidean distance, and the RG distribution becomes the standard multivariate Gaussian with covariance matrix $\sigma^2 \mathbf{I}$. On more general manifolds, however, the curvature and topology are taken into account through the geodesic distance and the volume element, leading to a natural extension of the Gaussian concept. This construction can be applied to spaces such as hyperbolic manifolds, where one can define the distribution in the tangent space at a point $\mu$ and then use the exponential map to project it onto the manifold.

**Comparison to vMF.** A closely related distribution is the von Mises–Fisher (vMF) distribution, which is traditionally defined on the sphere $S^n$ by

$$\mathrm{vMF}(z \mid \mu, \kappa) \propto \exp\big(\kappa \langle z, \mu \rangle\big),$$

with $\mu \in S^n$ and $\langle \cdot, \cdot \rangle$ denoting the standard dot product. The vMF distribution is based on the notion of directional data and an inner product structure that measures alignment. In contrast, the RG distribution is inherently tied to the Riemannian metric, making it applicable to a much wider class of manifolds. Generalizing the idea behind the vMF distribution to other geometries often requires embedding the manifold into a larger ambient space and defining a suitable bilinear form (such as the Minkowski inner product in hyperbolic geometry). In this sense, the RG approach offers a more natural and geometrically intrinsic formulation.

In summary, the Riemannian Gaussian distribution is defined in terms of the geodesic distance and the corresponding volume element, and it adapts to the underlying geometry of any Riemannian manifold.

# D RG-VFM AND LINK WITH RFM

## D.1 DETAILED DERIVATION OF RG-VFM OBJECTIVE

**Proposition D.1.** *If the manifold* $(\mathcal{M}, \mathbf{g})$ *is homogeneous, the normalization constant*

$$C = \int_{\mathcal{M}} \exp\left(-\frac{dist_{\mathbf{g}}(z, \mu)^2}{2\sigma^2}\right) d\mathcal{M}_z \tag{24}$$

*is independent of the mean* $\mu$.

*Proof.* We can initially rename the normalization constant $C$ by making the dependency on the mean explicit, referring to it as $C(\mu)$. In this setting, we want to prove that for two arbitrary mean values $\bar{\mu}$ and $\tilde{\mu}$, we have $C(\bar{\mu}) = C(\tilde{\mu})$.

By definition, a Riemannian manifold $\mathcal{M}$ is homogeneous if $\forall x, y \in \mathcal{M}, \exists f : \mathcal{M} \to \mathcal{M}$ such that $f(x) = y$ and with $f$ being an isometry, meaning that $\mathrm{dist}_{\mathbf{g}}(x, y) = \mathrm{dist}_{\mathbf{g}}(f(x), f(y))$.

We can then assume that $f$ satisfies $\bar{\mu} = f(\tilde{\mu})$, getting the following:

$$C(\tilde{\mu}) = \int_{\mathcal{M}} \exp\left(-\frac{\mathrm{dist}_{\mathbf{g}}(z, \tilde{\mu})^2}{2\sigma^2}\right) d\mathcal{M}_z,$$

$$C(\bar{\mu}) = \int_{\mathcal{M}} \exp\left(-\frac{\mathrm{dist}_{\mathbf{g}}(y, \bar{\mu})^2}{2\sigma^2}\right) d\mathcal{M}_y,$$

$$C(f(\tilde{\mu})) = \int_{\mathcal{M}} \exp\left(-\frac{\mathrm{dist}_{\mathbf{g}}(y, f(\tilde{\mu}))^2}{2\sigma^2}\right) d\mathcal{M}_y,$$

with $C(\bar{\mu}) = C(f(\tilde{\mu}))$.

Let's suppose that $y := f(s)$, for some $s \in \mathcal{M}$. By the definition of isometry, we have $\mathrm{dist}_{\mathbf{g}}(y, f(\tilde{\mu})) = \mathrm{dist}_{\mathbf{g}}(f(s), \tilde{\mu}) = \mathrm{dist}_{\mathbf{g}}(s, \bar{\mu})$. Furthermore, for any integrable scalar function $\phi : \mathcal{M} \to \mathbb{R}$ and isometry $f$:

$$\int_{\mathcal{M}} \phi(y) \, d\mathcal{M}_y = \int_{\mathcal{M}} \phi\big(f(s)\big) \, d\mathcal{M}_s.$$

By applying these two facts to our case, we obtain the following series of equalities:

$$
\begin{aligned}
C(\bar{\mu}) &= \int_{\mathcal{M}} \exp\left(-\frac{\mathrm{dist}_{\mathbf{g}}(y, f(\tilde{\mu}))^2}{2\sigma^2}\right) d\mathcal{M}_y \\
&= \int_{\mathcal{M}} \exp\left(-\frac{\mathrm{dist}_{\mathbf{g}}(f(s), f(\tilde{\mu}))^2}{2\sigma^2}\right) d\mathcal{M}_s \\
&= \int_{\mathcal{M}} \exp\left(-\frac{\mathrm{dist}_{\mathbf{g}}(s, \tilde{\mu})^2}{2\sigma^2}\right) d\mathcal{M}_s = C(\tilde{\mu}).
\end{aligned}
$$

$\square$

**Proposition 3.1.** *Let $\mathcal{M}$ be a homogeneous manifold with closed-form geodesics. Then, the RG-VFM objective reduces to*

$$
\mathcal{L}_{\mathrm{RG\text{-}VFM}}(\theta) = \mathbb{E}_{t,x_1,x}\left[|| \log_{x_1}(\mu_t^\theta(x))||_{\mathbf{g}}^2\right] = \mathbb{E}_{t,x_1,x}\left[\mathrm{dist}_{\mathbf{g}}(x_1, \mu_t^\theta(x))^2\right], \tag{9}
$$

*where $\log$ denotes the **logarithmic map** on the manifold and $\mathrm{dist}_{\mathbf{g}}$ is the geodesic distance.*

*Proof.* The objective of VFM is defined as

$$
\mathcal{L}_{\mathrm{VFM}}(\theta) = -\mathbb{E}_{t,x_1,x}\left[\log q_t^\theta(x_1|x)\right].
$$

We define the objective function of RG-VFM by setting the posterior probability as the Riemannian Gaussian, i.e.,

$$
q_t^\theta(x_1|x) = \mathcal{N}_{\mathrm{Riem}}(x_1 \mid \mu_t^\theta(x), \sigma_t(x)),
$$

so that

$$
\mathcal{L}_{\mathrm{RG\text{-}VFM}}(\theta) = -\mathbb{E}_{t,x_1,x}\left[\log \mathcal{N}_{\mathrm{Riem}}(x_1 \mid \mu_t^\theta(x), \sigma_t(x))\right].
$$

More explicitly, we have

$$
\begin{aligned}
\mathcal{L}_{\mathrm{RG\text{-}VFM}}(\theta) &= -\mathbb{E}_{t,x_1,x}\left[\log q_t^\theta(x_1|x)\right] \\
&= -\mathbb{E}_{t,x_1,x}\left[\log \mathcal{N}_{\mathrm{Riem}}(x_1 \mid \mu_t^\theta(x), \sigma_t(x))\right] \\
&= -\mathbb{E}_{t,x_1,x}\left[\log\left(\frac{1}{C(\mu_t^\theta(x))} \exp\left(-\frac{\mathrm{dist}_{\mathbf{g}}(x_1, \mu_t^\theta(x))^2}{2\sigma_t(x)^2}\right)\right)\right] \\
&= -\mathbb{E}_{t,x_1,x}\left[\log\left(\frac{1}{C(\mu_t^\theta(x))}\right) - \frac{\mathrm{dist}_{\mathbf{g}}(x_1, \mu_t^\theta(x))^2}{2\sigma_t(x)^2}\right] \\
&= -\mathbb{E}_{t,x_1,x}\left[\log\left(\frac{1}{C(\mu_t^\theta(x))}\right)\right] + \mathbb{E}_{t,x_1,x}\left[\frac{\mathrm{dist}_{\mathbf{g}}(x_1, \mu_t^\theta(x))^2}{2\sigma_t(x)^2}\right],
\end{aligned}
$$

where $\mathrm{dist}_{\mathbf{g}}()$ denotes the geodesic distance induced by the Riemannian metric $\mathbf{g}$.

Without any regularity assumptions on $\mathcal{M}$, no further simplification is possible. However, under the following assumptions the objective becomes more tractable:

1. **Homogeneity:** If the manifold $(\mathcal{M}, \mathbf{g})$ is homogeneous, the normalization constant

$$
C = \int_{\mathcal{M}} \exp\left(-\frac{\mathrm{dist}_{\mathbf{g}}(z, \mu)^2}{2\sigma^2}\right) d\mathcal{M}_z
$$

is independent of the mean $\mu$ (see proposition D.1). Hence, defining

$$
K := -\mathbb{E}_{t,x_1,x}\left[\log\left(\frac{1}{C(\mu_t^\theta(x))}\right)\right],
$$

which is constant with respect to $\theta$, we obtain

$$
\mathcal{L}_{\mathrm{RG\text{-}VFM}}(\theta) = K + \mathbb{E}_{t,x_1,x}\left[\frac{\mathrm{dist}_{\mathbf{g}}(x_1, \mu_t^\theta(x))^2}{2\sigma_t(x)^2}\right].
$$

Since $K$ is a constant that is independent of the model's parameters $\theta$, the minimization objective becomes

$$
\mathcal{L}_{\mathrm{RG\text{-}VFM}}(\theta) = \mathbb{E}_{t,x_1,x}\left[\frac{\mathrm{dist}_{\mathbf{g}}(x_1, \mu_t^\theta(x))^2}{2\sigma_t(x)^2}\right].
$$

2. **Closed-form Geodesics:** If the geometry allows closed-form expressions for geodesics, namely

$$\gamma(t) = \exp_x\Big( t \cdot \log_x(y) \Big),$$

then the geodesic distance between two points is given by:

$$\text{dist}_{\mathbf{g}}(z, \mu) = \| \log_z(\mu) \|_{\mathbf{g}}.$$

In this setting, we can write

$$\text{dist}_{\mathbf{g}}(x_1, \mu_t^\theta(x))^2 = \| \log_{x_1}(\mu_t^\theta(x)) \|_{\mathbf{g}}^2,$$

so that the objective becomes

$$\mathcal{L}_{\text{RG-VFM}}(\theta) = -\mathbb{E}_{t,x_1,x}\left[ \log\left( \frac{1}{C(\mu_t^\theta(x))} \right) \right] + \mathbb{E}_{t,x_1,x}\left[ \frac{1}{2\sigma_t(x)^2} \| \log_{x_1}(\mu_t^\theta(x)) \|_{\mathbf{g}}^2 \right].$$

3. **Combined Assumptions:** If both conditions hold, the objective simplifies to

$$\mathcal{L}_{\text{RG-VFM}}(\theta) = \mathbb{E}_{t,x_1,x}\left[ \frac{1}{2\sigma_t(x)^2} \| \log_{x_1}(\mu_t^\theta(x)) \|_{\mathbf{g}}^2 \right].$$

If we further assume that $\sigma_t(x)$ is constant, this reduces to

$$\mathcal{L}_{\text{RG-VFM}}(\theta) = \mathbb{E}_{t,x_1,x}\left[ \| \log_{x_1}(\mu_t^\theta(x)) \|_{\mathbf{g}}^2 \right].$$

$\square$

**Remark on the definition of $\sigma_t(x)$.** In the previous proof, the result is obtained by assuming $\sigma_t(x)$ to be constant. More in general, we could maintain the presence of $\sigma_t(x)$ explicit in the loss, obtaining $\mathcal{L}_{\text{RG-VFM}}(\theta) = \mathbb{E}_{t,x_1,x}\left[ \frac{1}{2\sigma_t(x)^2} \| \log_{x_1}(\mu_t^\theta(x)) \|_{\mathbf{g}}^2 \right]$. By being time dependent, $\sigma_t(x)$ can for example be defined as the normalization constant $\frac{1}{1-t}$. Despite the generality that it allows, for the sake of simplicity we make the choice to assume $\sigma_t(x)$ being constant, or implicit in the loss definition.

**Examples of simple geometries.** A homogeneous manifold does not necessarily imply that geodesics admit closed-form expressions. Conversely, the simple geometries with closed-form geodesics considered in the RFM setting—such as hyperspheres $\mathbb{S}^n$, hyperbolic spaces $\mathbb{H}^n$, flat tori $T^n = [0, 2\pi]^n$, and the space of SPD matrices $\mathcal{S}_d^+$ with the affine-invariant metric—are homogeneous. Thus, when restricting to these geometries for comparison with RFM, we are in the combined case.

**Special case: euclidean space.** In the Euclidean case (which also falls into the combined case), the objective simplifies further to

$$\mathcal{L}_{\text{RG-VFM}}(\theta) = \mathbb{E}_{t,x_1,x}\left[ \| \mu_t^\theta(x) - x_1 \|^2 \right].$$

### D.2  How does RG-VFM fit in the Existing Flow Matching Framework?

Figure 2 (left) illustrates how RG-VFM fits within the framework of related FM models. In VFM, a *probabilistic* generalization of CFM is obtained by making the posterior distribution explicit and customizable, obtaining standard CFM under the choice of a specific Gaussian (see Eijkelboom et al. (2024)). In contrast, RFM serves as a *geometric* generalization of CFM, where the model and its objective take into account the intrinsic properties and metric of the underlying Riemannian manifold. The same happens for the variational models: VFM with a Gaussian posterior is a particular instance of RG-VFM when the geometry is Euclidean. In Euclidean space, $\| \log_{x_1}(\mu_t^\theta(x)) \|_{\mathbf{g}}^2$ reduces to $\| \mu_t^\theta(x) - x_1 \|_2^2$, thereby recovering the VFM objective.

A further comparison can be made between the simplified version of RFM and RG-VFM-$\mathcal{M}$, where $\mathcal{M}$ is a homogeneous manifold with closed-form geodesics. The variational model (RG-VFM) is *not*

a direct generalization of vanilla RFM because, unlike in Euclidean space, *tangent spaces at different points on a manifold do not coincide*. This difference is reflected in the models' outputs (fig. 2): vanilla models predict velocity fields, which are integrated as ODEs to construct flows, whereas variational models predict endpoint distributions, ideally aligning with the target distribution $p_1$.

In Euclidean space, the difference between two vectors starting at $x$ and pointing to different endpoints is simply the vector between those endpoints, leading to identical $L_2$ terms in the objectives, i.e. $\|\mu_t^\theta(x) - x_1\|_2^2$ for VFM and $\|u_t(x \mid x_1) - v_t^\theta(x)\|^2$ for CFM. However, since $T_x\mathcal{M} \neq T_{x_1}\mathcal{M}$ in general, in their geometric counterparts this equivalence no longer holds: indeed, the difference vector in the RFM objective, $v_t^\theta(x) - \frac{\log_x(x_1)}{(1-t)}$, is in $T_x\mathcal{M}$, while $\log_{x_1}(\mu_t^\theta(x))$ is in $T_{x_1}\mathcal{M}$. This fundamental distinction separates RG-VFM from RFM. More details are in the following Section.

---

**Algorithm 3** CFM

**Require:** base $p \in \mathbb{R}^d$, target $q \in \mathbb{R}^d$.
  # **Training Phase**
  Initialize parameters $\theta$ of $v_t$
  **while** not converged **do**
      sample $t \sim \mathcal{U}(0,1)$, $x_0 \sim p$, $x_1 \sim q$
      compute linear interpolation:
          $x_t = t \cdot x_1 + (1-t) \cdot x_0$
      compute corresponding velocity:
          $\dot{x}_t = (x_1 - x_t)/(1-t)$
      $\ell(\theta) = \mathbb{E}_{t,x_1,x}\left[\|v_t(x_t;\theta) - \dot{x}_t\|_g^2\right]$
      $\theta = \texttt{optimizer\_step}(\ell(\theta))$
  **end while**

  # **Generation Phase**
  sample noise $x_0 \sim p$
  $x_1 = \texttt{solve\_ODE}([0,1], x_0, v_t(x_t;\theta))$

---

**Algorithm 4** VFM

**Require:** base $p \in \mathbb{R}^d$, target $q \in \mathbb{R}^d$.
  # **Training Phase**
  Initialize parameters $\theta$ of $\mu_t$
  **while** not converged **do**
      sample $t \sim \mathcal{U}(0,1)$, $x_0 \sim p$, $x_1 \sim q$
      compute linear interpolation:
          $x_t = t \cdot x_1 + (1-t) \cdot x_0$
      $\ell(\theta) = \mathbb{E}_{t,x_1,x}\left[\|\mu_t(x_t;\theta) - x_1\|_g^2\right]$
      $\theta = \texttt{optimizer\_step}(\ell(\theta))$
  **end while**

  # **Generation Phase**
  sample noise $x_0 \sim p$
  compute corresponding velocity:
  $\dot{x}_t = \frac{\mu_t(x_t;\theta) - x_t}{1-t}$
  $x_1 = \texttt{solve\_ODE}([0,1], x_0, \dot{x}_t)$

---

**Algorithm 5** RFM

**Require:** base $p \in \mathcal{M}$, target $q \in \mathcal{M}$.
  # **Training Phase**
  Initialize parameters $\theta$ of $v_t$
  **while** not converged **do**
      sample $t \sim \mathcal{U}(0,1)$, $x_0 \sim p$, $x_1 \sim q$
      compute geodesic interpolation:
          $x_t = \exp_{x_0}(t \cdot \log_{x_0}(x_1))$
      compute corresponding velocity:
          $\dot{x}_t = \texttt{PT}_{x_0 \to x_t}(\log_{x_0}(x_1))$
      $\ell(\theta) = \mathbb{E}_{t,x_1,x}\left[\|v_t(x_t;\theta) - \dot{x}_t\|_g^2\right]$
      $\theta = \texttt{optimizer\_step}(\ell(\theta))$
  **end while**

  # **Generation Phase**
  sample noise $x_0 \sim p$
  $x_1 = \texttt{solve\_ODE}([0,1], x_0, v_t(x_t;\theta))$

---

**Algorithm 6** RG-VFM (general)

**Require:** base $p$, target $q \in \mathcal{M}$.
  # **Training Phase**
  Initialize parameters $\theta$ of $\mu_t$
  **while** not converged **do**
      sample $t \sim \mathcal{U}(0,1)$, $x_0 \sim p$, $x_1 \sim q$
      compute interpolation:
          $x_t = \texttt{int}(t, x_0, x_1)$
      $\ell(\theta) = \mathbb{E}_{t,x_1,x}\left[\texttt{dist}_g^2(x_1, \mu_t(x_t;\theta))\right]$
      $\theta = \texttt{optimizer\_step}(\ell(\theta))$
  **end while**

  # **Generation Phase**
  sample noise $x_0 \sim p$
  compute corresponding velocity $\dot{x}_t$
  $x_1 = \texttt{solve\_ODE}([0,1], x_0, \dot{x}_t)$

---

**Algorithm 7** RG-VFM intrinsic

**Require:** base $p \in \mathcal{M}$, target $q \in \mathcal{M}$.
  # **Training Phase**
  Initialize parameters $\theta$ of $\mu_t$
  **while** not converged **do**
    sample $t \sim \mathcal{U}(0,1)$, $x_0 \sim p$, $x_1 \sim q$
    compute geodesic interpolation:
      $x_t = \exp_{x_0}(t \cdot \log_{x_0}(x_1))$
    $\ell(\theta) = \mathbb{E}_{t,x_1,x}\left[\text{dist}_g^2(x_1, \mu_t(x_t; \theta))\right]$
    $\theta = \texttt{optimizer\_step}(\ell(\theta))$
  **end while**

  # **Generation Phase**
  sample noise $x_0 \sim p$
  compute corresponding velocity:
  $\dot{x}_t = \frac{\log_{x_t}(\mu_t(x_t; \theta))}{1-t}$
  $x_1 = \texttt{solve\_ODE}([0,1], x_0, \dot{x}_t)$

**Algorithm 8** RG-VFM extrinsic

**Require:** base $p \in \mathbb{R}^d$, target $q \in \mathcal{M}$.
  # **Training Phase**
  Initialize parameters $\theta$ of $\mu_t$
  **while** not converged **do**
    sample $t \sim \mathcal{U}(0,1)$, $x_0 \sim p$, $x_1 \sim q$
    compute linear interpolation:
      $x_t = t \cdot x_1 + (1-t) \cdot x_0$
    $\ell(\theta) = \mathbb{E}_{t,x_1,x}\left[\text{dist}_g^2(x_1, \mu_t(x_t; \theta))\right]$
    $\theta = \texttt{optimizer\_step}(\ell(\theta))$
  **end while**

  # **Generation Phase**
  sample noise $x_0 \sim p$
  compute corresponding velocity:
  $\dot{x}_t = \frac{\mu_t(x_t; \theta) - x_t}{1-t}$
  $x_1 = \texttt{solve\_ODE}([0,1], x_0, \dot{x}_t)$

## D.3 RG-VFM vs RFM on Homogeneous Spaces with Closed-Form Geodesics

The objective of RG-VFM is defined as

$$\mathcal{L}_{\text{RG-VFM}}(\theta) = \mathbb{E}_{t,x_1,x}\left[\|\log_{x_1}(\mu_t^\theta(x))\|_{\mathbf{g}}^2\right],$$

while the objective of RFM, in the case of closed-form geodesics, is given by

$$\mathcal{L}_{\text{RFM}}(\theta) = \mathbb{E}_{t,x_1,x}\left[\|v_t^\theta(x) - \log_x(x_1)/(1-t)\|_{\mathbf{g}}^2\right],$$

with $\mathbf{g}$ being the metric tensor at $x \sim p_t(x|x_1)$.

Ignoring multiplicative constants that depend only on $t$ and $x$, comparing the two losses reduces to comparing the quantities

$$\|\log_{x_1}(\mu_t^\theta(x))\|_{\mathbf{g}}^2 \quad \text{and} \quad \|v_t^\theta(x) - \log_x(x_1)\|_{\mathbf{g}}^2.$$

**Euclidean space.** In Euclidean space $\mathbb{R}^n$, the tangent space at each point is naturally identified with $\mathbb{R}^n$. In this setting,

$$\log_{x_1}(\mu_t^\theta(x)) = \mu_t^\theta(x) - x_1 \quad \text{and} \quad \log_x(x_1) = x_1 - x.$$

Notice that

$$\mu_t^\theta(x) - x_1 = \mu_t^\theta(x) - x + x - x_1 = (\mu_t^\theta(x) - x) - (x_1 - x) = (\mu_t^\theta(x) - x) - \log_x(x_1).$$

If we define (ignoring multiplicative constants such as $1/(1-t)$)

$$v_t^\theta(x) = \log_x(\mu_t^\theta(x)) = \mu_t^\theta(x) - x,$$

then it follows that

$$\log_{x_1}(\mu_t^\theta(x)) = \log_x(\mu_t^\theta(x)) - \log_x(x_1),$$

implying

$$\|\log_{x_1}(\mu_t^\theta(x))\|_{\mathbf{g}}^2 = \|v_t^\theta(x) - \log_x(x_1)\|_{\mathbf{g}}^2.$$

Thus, $\mathcal{L}_{\text{RG-VFM}}(\theta)$ and $\mathcal{L}_{\text{RFM}}(\theta)$ are equivalent up to an additive constant. This result is consistent with the known equivalence between $\mathcal{L}_{\text{VFM}}(\theta)$ and $\mathcal{L}_{\text{CFM}}(\theta)$.

**General geometries.** In non-Euclidean spaces, however, the quantities

$$\|\log_{x_1}(\mu_t^\theta(x))\|_{\mathbf{g}}^2 \quad \text{and} \quad \|v_t^\theta(x) - \log_x(x_1)/(1-t)\|_{\mathbf{g}}^2$$

are not necessarily equal. This is because $\log_{x_1}(\mu_t^\theta(x))$ is a vector in $T_{x_1}\mathcal{M}$, while $\log_x(\mu_t^\theta(x)) - \log_x(x_1)$ lies in $T_x\mathcal{M}$, and in general $T_{x_1}\mathcal{M} \neq T_x\mathcal{M}$. Establishing a relation between these vectors is not straightforward and can be illustrated by comparing the law of cosines in Euclidean, hyperbolic spaces, and on hyperspheres.

### D.4 A Comparison based on Jacobi Fields

In this section, we report the notations used for explaining the comparison based on Jacobi fields, in table 5, as well as the proofs of the propositions in section 4.

| Symbol | Name | Description |
|---|---|---|
| $x_{\tau=0}^s = x_0^s = x_0$ | base point | Base point obtained at $\tau = 0$. |
| $x_1^1, x_1^0$ | generated and target end point | Predicted and reference data points after training. |
| $v^1, v^0$ | generated and target vector | Predicted and reference vectors after training. |
| $\mathcal{L}_{\text{RG-VFM}} = \mathbb{E}_{t,x,x_1}[\text{dist}(x_1^1, x_1^0)^2]$ | RG-VFM loss | Geodesic distance used in the Variational Riemannian FM loss function. |
| $\mathcal{L}_{\text{RFM}} = \mathbb{E}_{t,x,x_1}[\|v^1 - v^0\|_{\mathbf{g}}^2]$ | RFM loss | Norm of vector fields used in the Vanilla Riemannian FM loss function. |

Table 5: Synthetic notations used in this section for the Jacobi field and Riemannian flow matching losses.

**Definition 4.1** (Jacobi field at a vanishing starting point). *The family of geodesics defined as:*

$$\alpha(s, \tau) := \gamma_s : \tau \to exp_{x_0}(\tau(v + sw)),$$

*with $s \in [0, 1]$ and $\tau \in [0, 1]$, $v, w \in T_{x_0}\mathcal{M}$, is a smooth family of shooting geodesics with $\gamma_s(0) = x_0$, $\dot{\gamma}_0(0) = v$ and $\dot{\gamma}_1(0) = v + w$.*

*For each fixed $\tau \in [0, 1]$, there exists a vector field, called **Jacobi field**,*

$$J(\tau) := \partial_s \alpha(s, \tau)\big|_{s=0}$$

*along the geodesic $\gamma_s(\tau) := \alpha(s, \tau)$ satisfying the ODE equation: $D_\tau^2 J + R(J, \dot{\gamma}_s) \dot{\gamma}_s = 0$, with $R$ the Riemannian curvature tensor of the manifold. In particular, this Jacobi field is uniquely defined by the initial conditions and at $\tau = 0$ one has the initial conditions: $J(0) = 0$, and $D_\tau J(0) = w$.*

*Proof.* Lee (2018)[Lemma 10.9. and Proposition 10.2.]

Since $\tau \to \alpha(s, \tau)$ is a geodesic for each $s$, we have $D_\tau^2 \alpha = 0$. Differentiate with respect to $s$ and use the torsion-free, metric connection to get $D_\tau^2(\partial_s \alpha) + R(\partial_s \alpha, \partial_\tau \alpha)\partial_\tau \alpha = 0$, which is the Jacobi equation for $J(\tau) = \partial_s \alpha(s, \tau)|_{s=0}$. Because $\alpha(s, 0) = x_0$ for all $s$, we get $J(0) = 0$. Also $\partial_\tau \alpha(s, 0) = v + sw$, so $D_\tau J(0) = \partial_s(v + sw)|_{s=0} = w|_{s=0} = w$.

The Jacobi equation is a linear second-order ODE along $\gamma$ with smooth coefficients, there is a unique solution with any prescribed initial data $(J(0), D_\tau J(0)) = (0, w)$. $\square$

For $s = 0$, we are interested in the geodesic $\gamma_0 : \tau \to \exp_{x_0}(\tau v^0)$, with $v^0$ the target velocity, and for $s = 1$, the geodesic $\gamma_1 : \tau \to \exp_{x_0}(\tau v^1)$ is defined with $v^1$ the velocity learned by the model. $w = v^0 - v^1$ corresponds to the discrepancy between the learned and the conditional initial velocity, and their norm is exactly what is minimized in the vanilla Riemannian Flow Matching (Chen & Lipman, 2024).

**Proposition 4.1** (Loss functions as evaluation of Jacobi fields). *Consider a Jacobi field $J(\tau) := \partial_s \alpha(s, \tau)\big|_{s=0}$ as defined in definition 4.1. We denote $\mathcal{L}_{\text{RFM}}$ the loss function of the (vanilla) Riemannian Flow Matching (Chen & Lipman, 2024), and $\mathcal{L}_{\text{RG-VFM}}$ the loss function for our proposed Riemannian Variational Flow Matching. Then the following equalities hold:*

$$\mathcal{L}_{\text{RFM}}(\theta) = \mathbb{E}_{t,x_1,x}[\|u_t(x \mid x_1) - v_t^\theta(x)\|_{\mathbf{g}}^2] = \mathbb{E}_{t,x_1,x}[\|D_\tau J(0)\|_{\mathbf{g}}^2], \tag{10}$$

$$\mathcal{L}_{\text{RG-VFM}}(\theta) = \mathbb{E}_{t,x_1,x}[\|\log_{x_1}(\mu_t^\theta(x))\|_{\mathbf{g}}^2] = \mathbb{E}_{t,x_1,x}[\|J(1)_{\mathbf{g}}\|^2]. \tag{11}$$

*Proof.* In RFM, the goal is to learn a velocity field $v_t^\theta$ that transports the base distribution at $t = 0$ into the target distribution at $t = 1$. The loss penalizes the discrepancy between the conditional velocity $u_t(x \mid x_1)$ and the model's prediction $v_t^\theta(x)$, averaged over all time steps $t$, target samples $x_1 \sim p_{\text{data}}$, and intermediate samples $x \sim p_t(x \mid x_1)$.

When introducing Jacobi fields, we first define them for a fixed generative step $t$, a specific target point $x_1 = x_1^{s=0}$, and an associated intermediate point. As an abuse of notation, we suppose here that the Jacobi fields are obtained for all $t, x_1, x$, allowing us to take expectations over these variables.

1. Let us prove that:
$$\mathcal{L}_{\text{RFM}}(\theta) = \mathbb{E}_{t,x_1,x}\big[\|u_t(x \mid x_1) - v_t^\theta(x)\|_{\mathbf{g}}^2\big] = \mathbb{E}_{t,x_1,x}\big[\|D_\tau J(0)\|_{\mathbf{g}}^2\big],$$

   By definition of the initial conditions of our Jacobi field, we have the target velocity field defined as $\dot{\gamma}_{s=0}(0) := v^0 = u_t(x \mid x_1)$ and the learned velocity field defined as $\dot{\gamma}_{s=1}(0) := v^1 = v_t^\theta(x)$. By definition 4.1,
   $\|u_t(x \mid x_1) - v_t^\theta(x)\| = \|v^0 - v^1\| = \|-w\| = \|w\| = \|D_\tau J(0)\|$.

2. We want to prove the other equality:
$$\mathcal{L}_{\text{RG-VFM}}(\theta) = \mathbb{E}_{t,x_1,x}\big[\|\log_{x_1}(\mu_t^\theta(x))\|_{\mathbf{g}}^2\big] = \mathbb{E}_{t,x_1,x}\big[\|J(1)\|_{\mathbf{g}}^2\big].$$

   We observe that $\log_{x_1}(\mu_t^\theta(x)) = \log_{\gamma_0(1)}(\gamma_1(1))$. Let $p := \gamma_0(1)$ and $q := \gamma_1(1)$, the respective end points of the geodesics $\gamma_0$ and $\gamma_1$. If $q$ lies in the injectivity radius of $p$, then there is a unique minimizing geodesic $\sigma : \rho \in [0,1] \to \exp_p(\rho u)$ with $u = \log_p(q)$, with $\sigma(0) = p$ to $\sigma(1) = q$.

   a) We can then consider the Taylor expansion for the exponential map $\exp_p$ and consequently of the geodesic defined from it $\sigma(\rho)$, as in Monera et al. (2014):
   $$\sigma(\rho) = \sigma(0) + \sigma'(0)\rho + \tfrac{1}{2}\sigma''(0)\rho^2 + O(\|\rho u\|^3).$$
   By substituting the values, we obtain:
   $$\sigma(\rho) = p + u\rho + \tfrac{1}{2}\sigma''(0)\rho^2 + O(\|\rho u\|^3).$$
   We want now to reparametrize the geodesic with respect to the variable $s$, instead of $\rho \in [0,1]$. For this, we reparametrize it with a new initial velocity vector $w$ such that $\sigma_w(s) = \sigma_u(\rho(s))$ for a smooth reparametrization function $\rho = \rho(s)$. In this setting, we still have $\sigma_w(0) = p$ and $\sigma_w(1) = q$. Hence
   $$\sigma_w(s) = p + ws + \tfrac{1}{2}\sigma''(0)s^2 + O(\|sw\|^3),$$
   and for $s = 1$:
   $$q = \sigma_w(1) = p + w + \tfrac{1}{2}\sigma''(0) + O(\|w\|^3).$$

   b) From the perspective of the family of geodesics
   $$\gamma_s(\tau) = \exp_{x_0}(\tau(v + sw))$$
   and the corresponding Jacobi field $J(\tau)$, we can instead derive:
   $$\gamma_s(1) = \gamma_0(1) + J(1)s + \tfrac{1}{2}J'(1)s^2 + O(\|J(1)s\|^3),$$
   that for $s = 1$ gives:
   $$q = \gamma_1(1) = p + J(1) + \tfrac{1}{2}J'(1) + O(\|J(1)\|^3).$$

   We can observe that, for $s = 1$, the two Taylor expansions in (a) and (b) coincide, and we also have for the linear terms:
   $$\log_p q = w = J(1).$$

   From this, we derive the fundamental equality $\log_{x_1}(\mu_t^\theta(x)) = \log_p q = J(1)$, which exactly corresponds to what we want to prove.

$$\square$$

**Proposition 4.2.** $D_\tau J(0)$ *is a linear approximation of* $J(1)$.

*Proof.* Let $J^{(k)}(0) := D_\tau^k J(\tau)|_{\tau=0}$ be the $k$-th derivative of $J$ with respect to $\tau$, then evaluated at 0. The value of the Jacobi field at timestep $\tau$ can be computed through Taylor expansion of the Jacobi field centered in $\tau = 0$, as

$$J(\tau) = \tau J^{(1)}(0) + \frac{\tau^2}{2} J^{(2)}(0) + \frac{\tau^3}{6} J^{(3)}(0) + \mathcal{O}(\|w\|\tau^4)$$

Since the exponential map is smooth, the associated geodesics and Jacobi fields are smooth functions depending on $\tau$. The Taylor's theorem guarantees that we can approximate $J(\tau)$ around $\tau = 0$, with the big-O term quantifying the size of the error. In particular, the expansion centered at $\tau = 0$ remains valid when evaluated at any $\tau$, provided that the exponential map stays well-defined and smooth.

In our setting, $J(0) = 0$, and we are interested in the timestep $\tau = 1$, for which we get

$$J(1) = J^{(1)}(0) + \frac{1}{2} J^{(2)}(0) + \frac{1}{6} J^{(3)}(0) + \mathcal{O}(\|w\|)$$

and if we want to stop at the linear term:

$$J(1) \approx J^{(1)}(0),$$

in the sense that $J^{(1)}(0)$ is a linear approximation of $J(1)$. $\square$

**Proposition 4.3** (Difference of loss functions as a curvature term). *Consider a Jacobi field* $J(\tau) := \partial_s \alpha(s, \tau)|_{s=0}$ *as defined in definition 4.1 and the equivalences shown in proposition 4.1. The difference between* $\mathcal{L}_{\text{RG-VFM}}$ *and* $\mathcal{L}_{\text{RFM}}$ *encodes the manifold curvature through:*

$$\mathcal{L}_{\text{RG-VFM}}(\theta) = \mathcal{L}_{\text{RFM}}(\theta) + \underbrace{\mathbb{E}_{t,x_1,x}[\mathcal{C}(R, D_\tau J(0), v) + \mathcal{E}_{higher}]}_{\textit{curvature-dependent term}} \quad (13)$$

*where the leading-order curvature functional is:*

$$\mathcal{C}(R, D_\tau J(0), v) = -\frac{1}{3}\langle R(D_\tau J(0), v)v, D_\tau J(0)\rangle_{\mathbf{g}} - \frac{1}{6}\langle(\nabla_v R)(D_\tau J(0), v)v, D_\tau J(0)\rangle_{\mathbf{g}} \quad (14)$$

$$and \quad \mathcal{E}_{higher} = O(\|D_\tau J(0)\|^2 \|v\|^3), \quad (15)$$

*with* $R$ *the Riemannian curvature tensor and* $v = \dot{\gamma}_0$ *the reference geodesic velocity. The higher-order term* $\mathcal{E}_{higher}$ *encodes curvature variation along geodesics through covariant derivatives of* $R$. *In terms of the RFM loss terms,* $v = u_t(x \mid x_1)$ *and* $D_\tau J(0) = v_t^\theta(x) - u_t(x \mid x_1)$.

*Proof.* The proof consists of three steps: (1) we will detail the Taylor series of the norm of the Jacobi field, noted $S := \|J\|^2$, (2) we will compute up to the 5th order of those terms evaluated at $\tau = 0$, and (3) we will simplify the expression to have the approximation.

1. Let's look at the Taylor expansion of the Jacobi fields. The Jacobi fields are indefinitely differentiable at a point $\tau = 0$, and so have a Taylor series on this point. They can be expressed as

$$J(\tau) = \sum_{k=0}^n \frac{\tau^k}{k!} J^{(k)}(0) + \int_0^\tau \frac{(\tau - \rho)^n}{n!} J^{n+1}(\rho)d\rho,$$

   with $J^{(k)}(0) := D_\tau^k J(\tau)|_{\tau=0}$, the $k$-th derivative of $J$ with respect to $\tau$, then evaluated at 0. $R_n = \int_0^\tau \frac{(\tau-\rho)^n}{n!} J^{(n+1)}(\rho)d\rho$ is the remainder term.

We further note $S(\tau)$ the linear product of two Jacobi fields, defined as:

$$S(\tau) := \|J(\tau)\|^2 = \langle J(\tau), J(\tau) \rangle = \langle \sum_{m \geqslant 0} \frac{\tau^m}{m!} J^{(m)}(0), \sum_{n \geqslant 0} \frac{\tau^n}{n!} J^{(n)}(0) \rangle.$$

Note that here and in the following, $\langle \cdot, \cdot \rangle := \langle \cdot, \cdot \rangle_{\mathbf{g}}$, but we omit $\mathbf{g}$ to avoid overcharging equations.

By bilinearity and using the Cauchy product, we have:

$$S(\tau) = \sum_{m,n \geqslant 0} \frac{\tau^{m+n}}{m!n!} \langle J^{(m)}(0), J^{(n)}(0) \rangle = \sum_{r \geqslant 0} \frac{\tau^r}{r!} S^{(r)}(0)$$

with $S^{(r)}(0) := \sum_{l=0}^{r} \binom{r}{l} \langle J^{(l)}(0), J^{(r-l)}(0) \rangle$.

2. Let us compute the Jacobi terms $J^{(k)}(0) := D_\tau^k J(\tau)|_{\tau=0}$ and their norm $S^k(0)$.

We know that the Jacobi fields satisfy the following ODE equation

$$J^{(2)} + AJ = 0,$$

with $A(\tau)J(\tau) = R(J(\tau), \dot\gamma(\tau))\dot\gamma(\tau)$, with $\dot\gamma(0) = v$ is the initial velocity and $D_\tau \dot\gamma(\tau) = 0$ since $\gamma$ is a geodesic. We know the initial condition $J(0) = 0$ and $J^{(1)}(0) := D_\tau J(0) = w$. Noting $A^{(k)} := D_\tau^k A(\tau)$, we can compute with the chain rule:

$$
\begin{aligned}
D_\tau A(\tau) J(\tau) &= \nabla_{\dot\gamma} [R(J(\tau), \dot\gamma)\dot\gamma] \\
&= (\nabla_{\dot\gamma} R)(J(\tau), \dot\gamma)\dot\gamma + R(D_\tau J(\tau), \dot\gamma)\dot\gamma + R(J(\tau), D_\tau \dot\gamma)\dot\gamma + R(J(\tau), \dot\gamma)D_\tau \dot\gamma \\
&= (\nabla_{\dot\gamma} R)(J(\tau), \dot\gamma)\dot\gamma + R(D_\tau J(\tau), \dot\gamma)\dot\gamma \qquad \text{since} \quad D_\tau \dot\gamma = 0 \\
D_\tau^2 A(\tau) J(\tau) &= D_\tau(\nabla_{\dot\gamma} R)(J(\tau), \dot\gamma)\dot\gamma + D_\tau(R(D_\tau J(\tau), \dot\gamma)\dot\gamma) \\
&= (\nabla_{\dot\gamma}^2 R)(J(\tau), \dot\gamma)\dot\gamma + (\nabla_{\dot\gamma} R)(D_\tau J(\tau), \dot\gamma)\dot\gamma + (\nabla_{\dot\gamma} R)(D_\tau J(\tau), \dot\gamma)\dot\gamma + R(D_\tau^2 J(\tau), \dot\gamma)\dot\gamma \\
&= (\nabla_{\dot\gamma}^2 R)(J(\tau), \dot\gamma)\dot\gamma + 2(\nabla_{\dot\gamma} R)(D_\tau J(\tau), \dot\gamma)\dot\gamma + R(D_\tau^2 J(\tau), \dot\gamma)\dot\gamma
\end{aligned}
$$

For all t, we can express the derivatives of $A(\tau)J(\tau)$ as
$D_\tau [A(\tau)J(\tau)] = D_\tau [A(\tau)] J(\tau) + A(\tau)D_\tau [J(\tau)]$, with in general:
$D_\tau^k [A(\tau)] J(\tau) = (\nabla_{\dot\gamma}^k R)(J(\tau), \dot\gamma)\dot\gamma$ and $A(\tau)D_\tau^k [J(\tau)] = R(D_\tau^k J(\tau), \dot\gamma)\dot\gamma$.

$$
\begin{aligned}
J^{(0)}(0) &= 0 \\
J^{(1)}(0) &= w \\
J^{(2)}(0) &= -A(0)J(0) = 0 \\
J^{(3)}(0) &= -A^{(1)}(0)J(0) - A(0)J^{(1)}(0) = -R(w, v)v \\
J^{(4)}(0) &= -A^{(2)}(0)J(0) - 2A^{(1)}(0)J^{(1)}(0) - A(0)J^{(2)}(0) = -2(\nabla_v R)(w, v)v
\end{aligned}
$$

And we have the following norms:

$$S^{(0)}(0) = \langle J(0), J(0) \rangle = 0$$

$$S^{(1)}(0) = 2\langle J(0), J^{(1)}(0) \rangle = 0$$

$$S^{(2)}(0) = 2\langle J(0), J^{(2)}(0) \rangle + 2\langle J^{(1)}(0), J^{(1)}(0) \rangle = 2\|w\|^2$$

$$S^{(3)}(0) = 2\langle J(0), J^{(3)}(0) \rangle + 6\langle J^{(1)}(0), J^{(2)}(0) \rangle = 0$$

$$S^{(4)}(0) = 2\langle J(0), J^{(4)}(0) \rangle + 8\langle J^{(1)}(0), J^{(3)}(0) \rangle + 6\langle J^{(2)}(0), J^{(2)}(0) \rangle$$

$$= 8\langle J^{(1)}(0), J^{(3)}(0) \rangle = -8\langle R(w, v)v, w \rangle$$

$$S^{(5)}(0) = 2\langle J(0), J^{(5)}(0) \rangle + 10\langle J^{(1)}(0), J^{(4)}(0) \rangle + 20\langle J^{(2)}(0), J^{(3)}(0) \rangle$$

$$= 10\langle J^{(1)}(0), J^{(4)}(0) \rangle = -20\langle (\nabla_v R)(w, v)v, w \rangle$$

3. We can finally express $\|J\|^2$ in terms of Taylor series

$$\|J(\tau)\|^2 = \sum_{r \geqslant 0} \frac{\tau^r}{r!} S^{(r)}(0)$$

$$= S^{(0)}(0) + \tau S^{(1)}(0) + \frac{\tau^2}{2} S^{(2)}(0) + \frac{\tau^3}{6} S^{(3)}(0) + \frac{\tau^4}{24} S^{(4)}(0) + \frac{\tau^5}{120} S^{(5)}(0) + \text{remainder}$$

$$= \tau^2 \|w\|^2 - \tau^4 \frac{1}{3} \langle (R(w, v)v, w \rangle - \tau^5 \frac{1}{6} \langle (\nabla_v R)(w, v)v, w \rangle + \mathcal{O}(\tau^6 \|w\|^2 \|v\|^3)$$

with $\langle (\nabla_v R)(w, v)v, w \rangle \leqslant \|\nabla R\| \|v\|^3 \|w\|^2$, and we assume the curvature of our manifold being bounded, so $\nabla R \leqslant M$, with $M \in \mathbb{R}$. Setting now $\tau = 1$ and $w = D_\tau J(0)$, we get

$$\|J(\tau)\|^2 = \|D_\tau J(0)\|^2 + \mathcal{C}(R, D_\tau J(0), v) + \mathcal{E}_{\text{higher}}$$

with

$$\mathcal{C}(R, D_\tau J(0), v) = -\frac{1}{3} \langle R(D_\tau J(0), v)v, D_\tau J(0) \rangle_{\mathbf{g}} - \frac{1}{6} \langle (\nabla_v R)(D_\tau J(0), v)v, D_\tau J(0) \rangle_{\mathbf{g}}$$

$$\text{and} \quad \mathcal{E}_{\text{higher}} = O(\|D_\tau J(0)\|^2 \|v\|^3),$$

where the higher-order term $\mathcal{E}_{\text{higher}}$ encodes curvature variation along geodesics through covariant derivatives of $R$.

Taking the expectation with respect to the variables $t, x_1, x$ and considering the result of proposition 4.1, we obtain the desired result

$$\mathcal{L}_{\text{RG-VFM}}(\theta) = \mathcal{L}_{\text{RFM}}(\theta) + \mathbb{E}_{t, x_1, x}[\mathcal{C}(R, D_\tau J(0), v) + \mathcal{E}_{\text{higher}}]. \tag{25}$$

$\square$

**Corollary D.1.** *For $v^0 = u_t(x \mid x_1)$ and $v^1 = v_t^\theta(x)$, the following holds:*

$$\mathcal{C}(R, D_\tau J(0), v) = \mathcal{C}(R, v^0, v^1) = -\frac{1}{3} K(v^1, v^0) \|v^1 \wedge v^0\|^2 - \frac{1}{6} \langle (\nabla_{v^0} R)(v^1, v^0)v^0, v^1 \rangle. \tag{26}$$

*with $K$ the sectional curvature, $\wedge$ the wedge-product, $R$ the Riemannian curvature and $\nabla$ the covariant derivative. For constant sectional curvature $K$, we further have $\nabla_{v^0} R = 0$:*

$$\mathcal{L}_{\text{RG-VFM}}(\theta) = \mathcal{L}_{\text{RFM}}(\theta) - \frac{1}{3} K \|v^1 \wedge v^0\|^2 + \mathcal{O}(\|v^0\|^3 \|v^1\|^2). \tag{27}$$

*Proof.* The sectional curvature is a way to measure locally the normalized deviation between two geodesics. It is defined as:

$$K(w, v) := \frac{\langle R(w, v)v, w \rangle}{\|w \wedge v\|^2}$$

with $\|w \wedge v\|$ the area spanned by the vector $w$ and $v$. Furthermore, for the sphere and the hyperboloid, the sectional curvature is constant, being $K = 1$ and $K = -1$ respectively.

With this definition, we can first of all re-express, in point (2) of proposition 4.3:

$$S^{(4)}(0) = -8\langle R(w,v)v,w \rangle = -8K(w,v)\|w \wedge v\|^2.$$

Consequently, we get the following expression in part (3):

$$
\begin{aligned}
\|J(\tau)\|^2 &= \sum_{r \geqslant 0} \frac{\tau^r}{r!} S^{(r)}(0) \\
&= S^{(0)}(0) + \tau S^{(1)} + \frac{\tau^2}{2} S^{(2)}(0) + \frac{\tau^3}{6} S^{(3)}(0) + \frac{\tau^4}{24} S^{(4)}(0) + \frac{\tau^5}{120} S^{(5)}(0) + \text{remainder} \\
&= \tau^2 \|w\|^2 - \tau^4 \frac{1}{3} K(w,v)\|w \wedge v\|^2 - \tau^5 \frac{1}{6}\langle (\nabla_v R)(w,v)v,w \rangle + \mathcal{O}(\tau^6 \|w\|^2 \|v\|^3)
\end{aligned}
$$

With the initial velocity vectors $v = v^0$ and $w = v^1 - v^0$, we can express the Riemannian curvature tensor $\langle R(w,v)v,w \rangle = \langle R(v^1,v^0)v^0,v^1 \rangle$

$$\|J(\tau)\|^2 = \tau^2 \|v^1 - v^0\|^2 - \tau^4 \frac{1}{3} K(v^1,v^0)\|v^1 \wedge v^0\|^2 - \tau^5 \frac{1}{6}\langle (\nabla_{v^0} R)(v^1,v^0)v^0,v^1 \rangle + \mathcal{O}(\tau^6 \|v^0\|^3 \|v^1\|^2)$$

Hence, we have, evaluating the Taylor expansion at $\tau = 1$ and considering it in expectation:

$$\mathcal{L}_{\text{RG-VFM}} = \mathcal{L}_{\text{RFM}} - \mathbb{E}_{t,x_1,x} \left[ \frac{1}{3} K(v^1,v^0)\|v^1 \wedge v^0\|^2 + \frac{1}{6}\langle (\nabla_{v^0} R)(v^1,v^0)v^0,v^1 \rangle + \mathcal{O}(\|v^0\|^3 \|v^1\|^2) \right].$$

with $K$ the sectional curvature, $\wedge$ the wedge-product, $R$ the Riemannian curvature and $\nabla$ the covariant derivative.

For constant sectional curvature $K$, we further have $\nabla_{v^0} R = 0$:

$$\mathcal{L}_{\text{RG-VFM}} = \mathcal{L}_{\text{RFM}} - \frac{1}{3} K \|v^1 \wedge v^0\|^2 + \mathcal{O}(\|v^0\|^3 \|v^1\|^2).$$

Let us consider the angle between the vectors defined as $\cos(\theta) = \frac{\langle v^0, v^1 \rangle}{\|v^0\|\|v^1\|}$. In that case:

$$\|v^1 \wedge v^0\|^2 = \|v^1\|^2 \|v^0\|^2 - (\|v^1\|\|v^0\|\cos\theta)^2 = \|v^1\|^2 \|v^0\|^2 (1 - \cos^2\theta) = \|v^1\|^2 \|v^0\|^2 \sin^2\theta$$

For a sphere $\mathbb{S}^2$ ($K = +1$), we have:

$$\text{dist}(\gamma_0(1),\gamma_1(1))^2 = \|v^1 - v^0\|^2 - \frac{1}{3}\|v^1\|^2 \|v^0\|^2 \sin^2\theta + \mathcal{O}(\|v^0\|^3 \|v^1\|^2).$$

For a hyperboloid $\mathbb{H}^2$ ($K = -1$), we have:

$$\text{dist}(\gamma_0(1),\gamma_1(1))^2 = \|v^1 - v^0\|^2 + \frac{1}{3}\|v^1\|^2 \|v^0\|^2 \sin^2\theta + \mathcal{O}(\|v^0\|^3 \|v^1\|^2).$$

$\square$

# E    SYNTHETIC EXPERIMENTS ON HYPERSPHERE AND HYPERBOLOID

In this section, we present further results from the experiments described in section 5.1.

## E.1    DEFINITION OF THE MANIFOLDS

The hypersphere is defined as $\mathbb{S}^n := \{x \in \mathbb{R}^{n+1} : \langle x, x \rangle_E = 1\}$, with the standard Euclidean inner product $\langle x, y \rangle_E = x_0 y_0 + x_1 y_1 + x_2 y_2 + \cdots + x_n y_n$. Instead, we adopt the Lorentz model for the hyperbolic space, which is defined as the upper sheet of the hyperboloid embedded in Minkowski space. The Minkowski space is the manifold $\mathbb{R}^{n+1}$ equipped with the Lorentzian inner product $\langle x, y \rangle_{\mathcal{L}} = -x_0 y_0 + x_1 y_1 + x_2 y_2 + \cdots + x_n y_n$. In this setting, the Lorentz hyperbolic model is defined as $\mathbb{H}^n_K := \{x \in \mathbb{R}^{n+1} : \langle x, x \rangle_{\mathcal{L}} = 1/K, x_0 > 0, K < 0\}$, where we set $K = -1$.

## E.2    EXPERIMENTAL SETUP

In all experiments, the target distribution $p_1$ is the spherical checkerboard, so its support is $\mathbb{S}^2$. The noisy distribution $p_0$ varies by model: for CFM, VFM, and RG-VFM-$\mathbb{R}^3$, $p_0$ is the standard normal distribution in $\mathbb{R}^3$, while for RG-VFM-$\mathcal{M}$ and RFM, it is obtained by wrapping the standard normal distribution on either $\mathbb{S}^2$ or $\mathbb{H}^2_{-1}$. In every case, we train a five-layer MLP with 64/128 hidden features for 3000 epochs on 10000 training samples, that we use to generate 10000 samples using a Euler ODE solver. For the intrinsic geometric models, the Euler solver is manifold-aware, meaning that it's defined with the log and exp maps on the manifold. Additionally, for the variational models we used a clipping technique during sampling, in order to make sure that the normalization term $\frac{1}{1-t}$ would not be too high for values of $t$ approaching 1.

## E.3    RESULTS

Figure 4, fig. 10 and fig. 11 illustrate the generative flow trajectories over time, from the initial distribution $p_0$ to the generated distribution at $t = 1$.

Figures 6 and 13 displays the generated distributions unwrapped onto a flat surface for easier visualization and comparison. These results visually confirm the observations presented in section 5.1.

Finally, figs. 8 and 15 show histograms of the norm values of the generated samples. As discussed in section 5.1, this metric differentiates the Euclidean models (CFM and VFM) from the others. Ideally, points should have a Euclidean norm of 1 if lying on $\mathbb{S}^2$, or a Minkowski norm of -1 if on $\mathbb{H}^2_{-1}$. However, because the Euclidean models lack explicit geometric information, their points deviate slightly from the ideal norm, with CFM exhibiting a larger divergence. In contrast, the geometric models consistently generate points that lie almost exactly on the sphere.

## E.4    LAPLACE POSTERIOR PROBABILITY

We explore the definition of the VFM training loss as the absolute value of the geodesic distance, instead of the squared geodesic distance, which would be obtained by ideally defining the posterior distribution $q_t^\theta$ in the VFM loss (eq. (5)) as the Riemannian version of the Laplace distribution. This would be defined as in eq. (7), by replacing the $L^2$ norm of the geodesic distance with the $L^1$ norm, obtaining

$$\mathcal{L}^{Lap}_{\text{RG-VFM}}(\theta) = \mathbb{E}_{t,x_1,x} \left[ || \log_{x_1}(\mu_t^\theta(x)) ||_{\mathbf{g}} \right] = \mathbb{E}_{t,x_1,x} \left[ \text{dist}_{\mathbf{g}}(x_1, \mu_t^\theta(x)) \right]. \tag{28}$$

We observe that using a Laplace distribution as the posterior for the Riemannian VFM models yields better results, both visually and with respect to the considered metrics. This effect is particularly evident in the hyperbolic case, and we hypothesize that it arises from the different impacts of using $L^1$ versus $L^2$ norms in this space. The numerical results are reported in table 6, the probability paths in figs. 5 and 12, the sampled densities in figs. 7 and 14 and the norm histograms in figs. 9 and 16.

Table 6: **Results of synthetic experiments with Laplace posterior.** Abbreviations: Eucl. = Euclidean, Riem. = Riemannian, Ext. = extrinsic, Int. = intrinsic, Van. = vanilla, Var. = variational.

| | Sphere | | | Hyperboloid | | |
|---|---|---|---|---|---|---|
| | Coverage ↑ | C2ST↓ | Distance↓ | Coverage ↑ | C2ST↓ | Distance↓ |
| Eucl/Ext/Var (VFM) | 89.92 | 59.98 ± 0.56 | 0.034 ± 0.042 | 87.63 | 57.26 ± 0.59 | **0.001 ± 0.133** |
| Riem/Ext/Var (Ours) | **95.04** | 61.33 ± 0.23 | **0.008 ± 0.034** | **91.98** | 62.55 ± 0.30 | 0.041 ± 0.113 |
| Riem/Int/Var (Ours) | 90.56 | **57.39 ± 0.70** | - | 86.23 | **56.04 ± 0.41** | - |

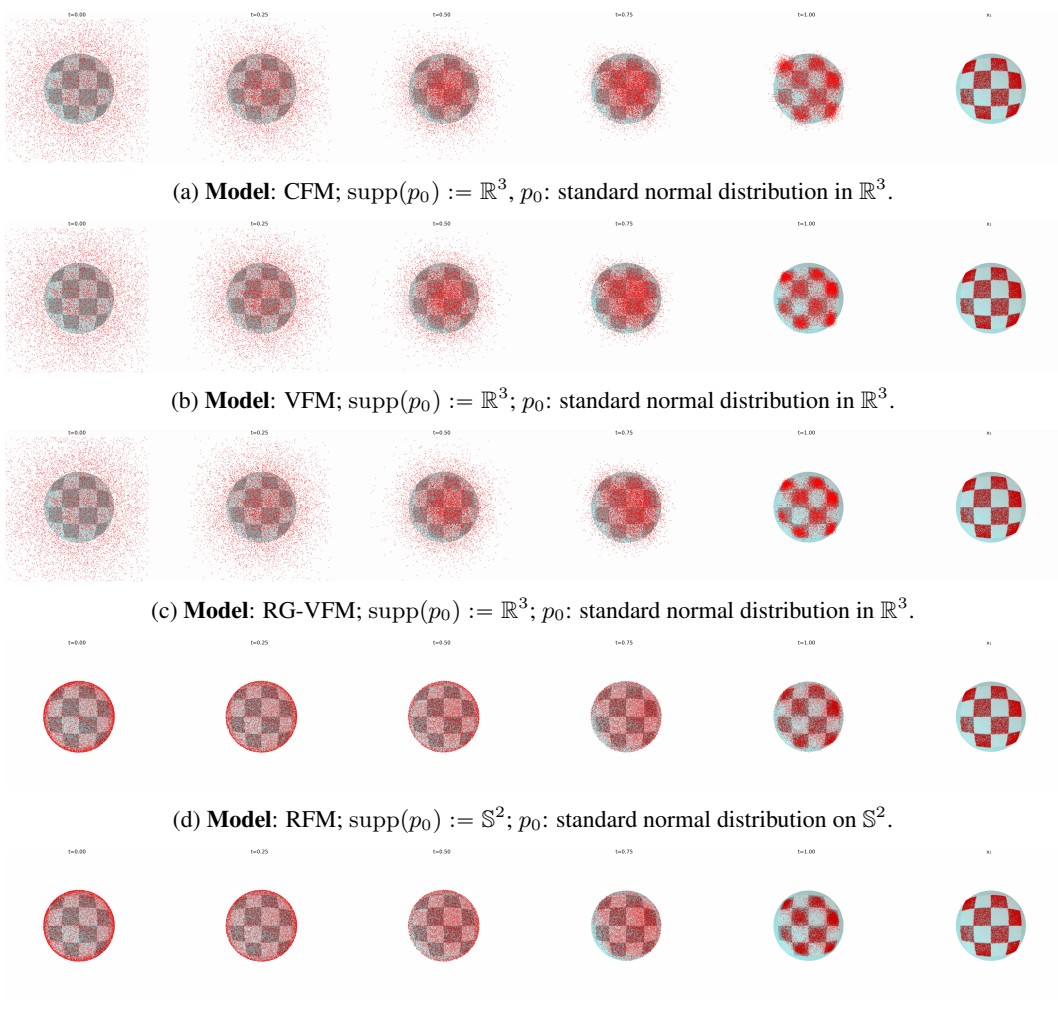

(a) **Model**: CFM; $\mathrm{supp}(p_0) := \mathbb{R}^3$, $p_0$: standard normal distribution in $\mathbb{R}^3$.

(b) **Model**: VFM; $\mathrm{supp}(p_0) := \mathbb{R}^3$; $p_0$: standard normal distribution in $\mathbb{R}^3$.

(c) **Model**: RG-VFM; $\mathrm{supp}(p_0) := \mathbb{R}^3$; $p_0$: standard normal distribution in $\mathbb{R}^3$.

(d) **Model**: RFM; $\mathrm{supp}(p_0) := \mathbb{S}^2$; $p_0$: standard normal distribution on $\mathbb{S}^2$.

(e) **Model**: RG-VFM; $\mathrm{supp}(p_0) := \mathbb{S}^2$; $p_0$: standard normal distribution on $\mathbb{S}^2$.

Figure 4: Flow trajectories of 10,000 samples, initially drawn from the noisy distribution $p_0$ at $t = 0$, evolving to reach their final configuration by $t = 1$. In all variational cases, the posterior distribution is **Normal**, and $p_1$ is the checkerboard distribution on $\mathbb{S}^2$.

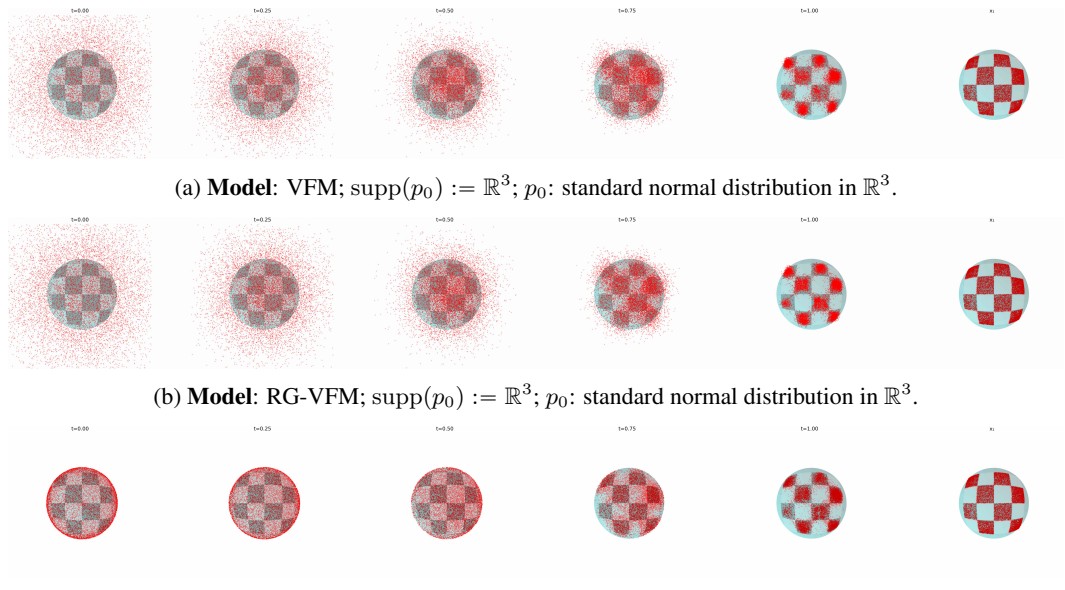

(a) **Model**: VFM; $\mathrm{supp}(p_0) := \mathbb{R}^3$; $p_0$: standard normal distribution in $\mathbb{R}^3$.

(b) **Model**: RG-VFM; $\mathrm{supp}(p_0) := \mathbb{R}^3$; $p_0$: standard normal distribution in $\mathbb{R}^3$.

(c) **Model**: RG-VFM; $\mathrm{supp}(p_0) := \mathbb{S}^2$; $p_0$: standard normal distribution on $\mathbb{S}^2$.

Figure 5: Flow trajectories of 10,000 samples, initially drawn from the noisy distribution $p_0$ at $t = 0$, evolving to reach their final configuration by $t = 1$. In all variational cases, the posterior distribution is **Laplace**, and $p_1$ is the checkerboard distribution on $\mathbb{S}^2$.

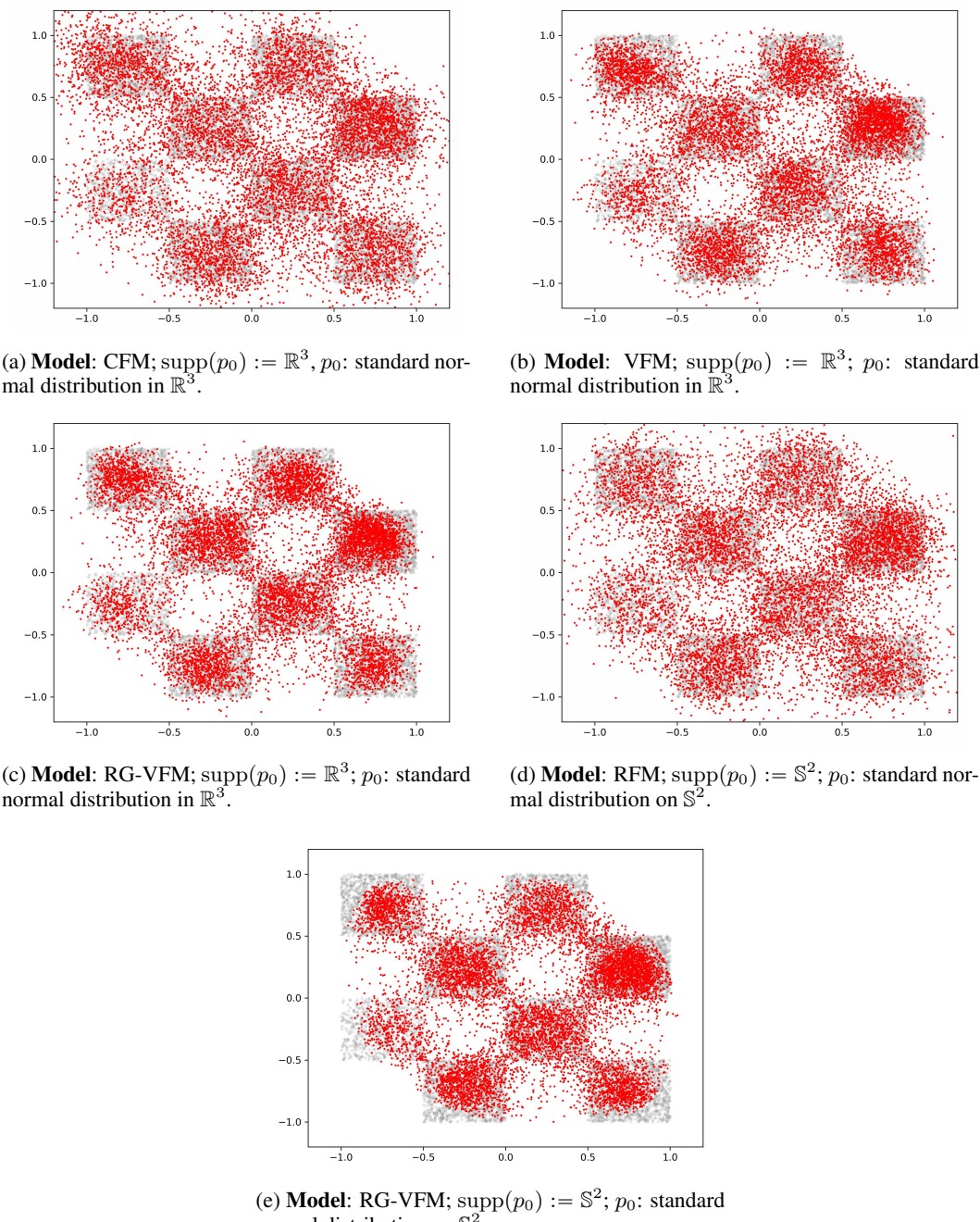

(a) **Model**: CFM; $\mathrm{supp}(p_0) := \mathbb{R}^3$, $p_0$: standard normal distribution in $\mathbb{R}^3$.

(b) **Model**: VFM; $\mathrm{supp}(p_0) := \mathbb{R}^3$; $p_0$: standard normal distribution in $\mathbb{R}^3$.

(c) **Model**: RG-VFM; $\mathrm{supp}(p_0) := \mathbb{R}^3$; $p_0$: standard normal distribution in $\mathbb{R}^3$.

(d) **Model**: RFM; $\mathrm{supp}(p_0) := \mathbb{S}^2$; $p_0$: standard normal distribution on $\mathbb{S}^2$.

(e) **Model**: RG-VFM; $\mathrm{supp}(p_0) := \mathbb{S}^2$; $p_0$: standard normal distribution on $\mathbb{S}^2$.

Figure 6: Sample distributions generated by different models (representing the flow configuration at $t = 1$) unwrapped from $\mathbb{S}^2$ to $\mathbb{R}^2$ for improved visualization. The true checkerboard distribution is shown in gray in the background. In all variational cases, the posterior distribution is **Normal**, and $p_1$ is the checkerboard distribution on $\mathbb{S}^2$.

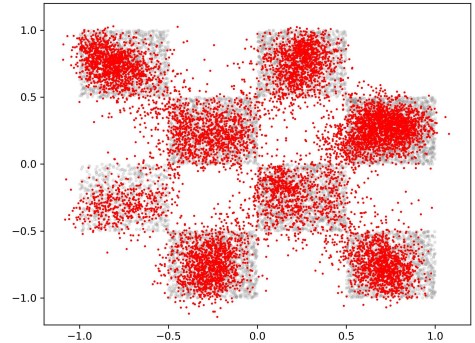

(a) **Model**: VFM; $\text{supp}(p_0) := \mathbb{R}^3$; $p_0$: standard normal distribution in $\mathbb{R}^3$.

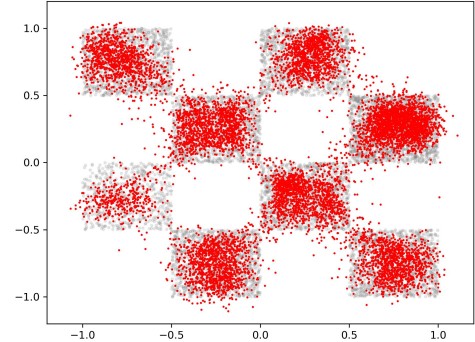

(b) **Model**: RG-VFM; $\text{supp}(p_0) := \mathbb{R}^3$; $p_0$: standard normal distribution in $\mathbb{R}^3$.

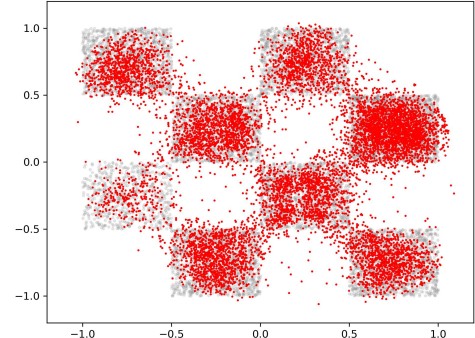

(c) **Model**: RG-VFM; $\text{supp}(p_0) := \mathbb{S}^2$; $p_0$: standard normal distribution on $\mathbb{S}^2$.

Figure 7: Sample distributions generated by different models (representing the flow configuration at $t = 1$) unwrapped from $\mathbb{S}^2$ to $\mathbb{R}^2$ for improved visualization. The true checkerboard distribution is shown in gray in the background. In all variational cases, the posterior distribution is **Laplace**, and $p_1$ is the checkerboard distribution on $\mathbb{S}^2$.

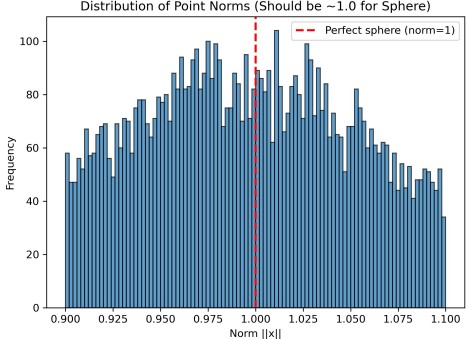

(a) **Model**: CFM; $\mathrm{supp}(p_0) := \mathbb{R}^3$, $p_0$: standard normal distribution in $\mathbb{R}^3$.

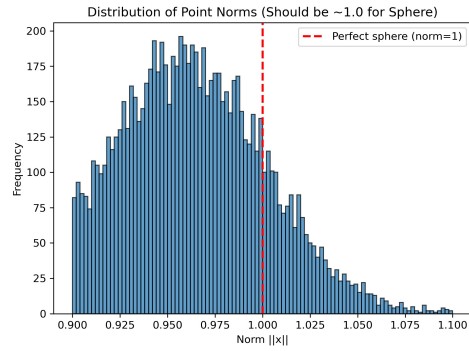

(b) **Model**: VFM; $\mathrm{supp}(p_0) := \mathbb{R}^3$; $p_0$: standard normal distribution in $\mathbb{R}^3$.

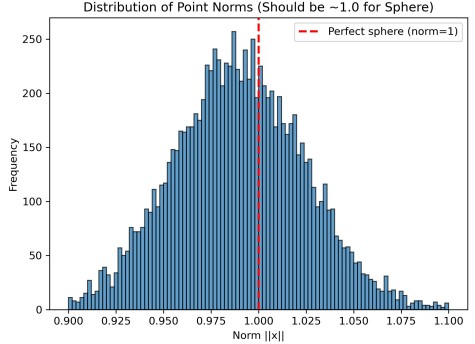

(c) **Model**: RG-VFM; $\mathrm{supp}(p_0) := \mathbb{R}^3$; $p_0$: standard normal distribution in $\mathbb{R}^3$.

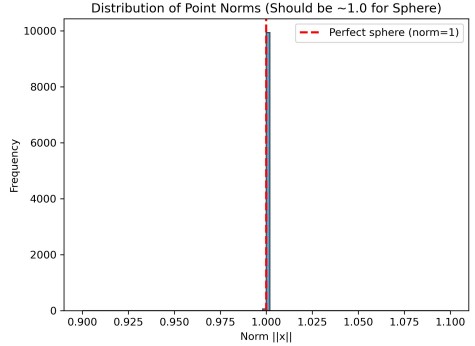

(d) **Model**: RFM; $\mathrm{supp}(p_0) := \mathbb{S}^2$; $p_0$: standard normal distribution on $\mathbb{S}^2$.

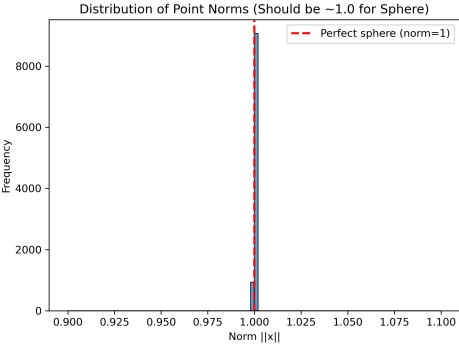

(e) **Model**: RG-VFM; $\mathrm{supp}(p_0) := \mathbb{S}^2$; $p_0$: standard normal distribution on $\mathbb{S}^2$.

Figure 8: Histogram of the norm values of the 10,000 samples describing the generated distribution. In all variational cases, the posterior distribution is **Normal**, and $p_1$ is the checkerboard distribution on $\mathbb{S}^2$.

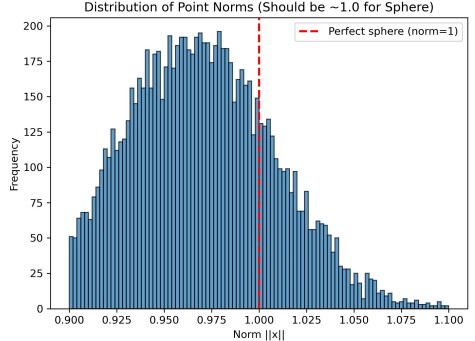

(a) **Model**: VFM; $\operatorname{supp}(p_0) := \mathbb{R}^3$; $p_0$: standard normal distribution in $\mathbb{R}^3$.

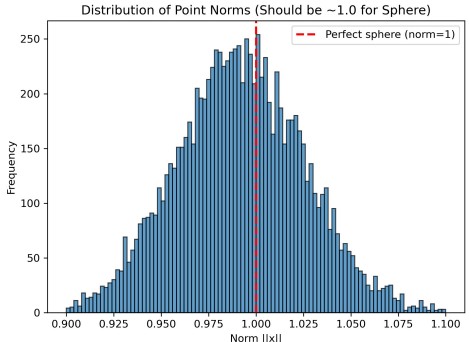

(b) **Model**: RG-VFM; $\operatorname{supp}(p_0) := \mathbb{R}^3$; $p_0$: standard normal distribution in $\mathbb{R}^3$.

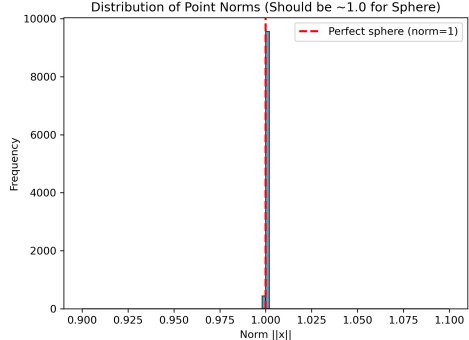

(c) **Model**: RG-VFM; $\operatorname{supp}(p_0) := \mathbb{S}^2$; $p_0$: standard normal distribution on $\mathbb{S}^2$.

Figure 9: Histogram of the norm values of the 10,000 samples describing the generated distribution. In all variational cases, the posterior distribution is **Laplace**, and $p_1$ is the checkerboard distribution on $\mathbb{S}^2$.

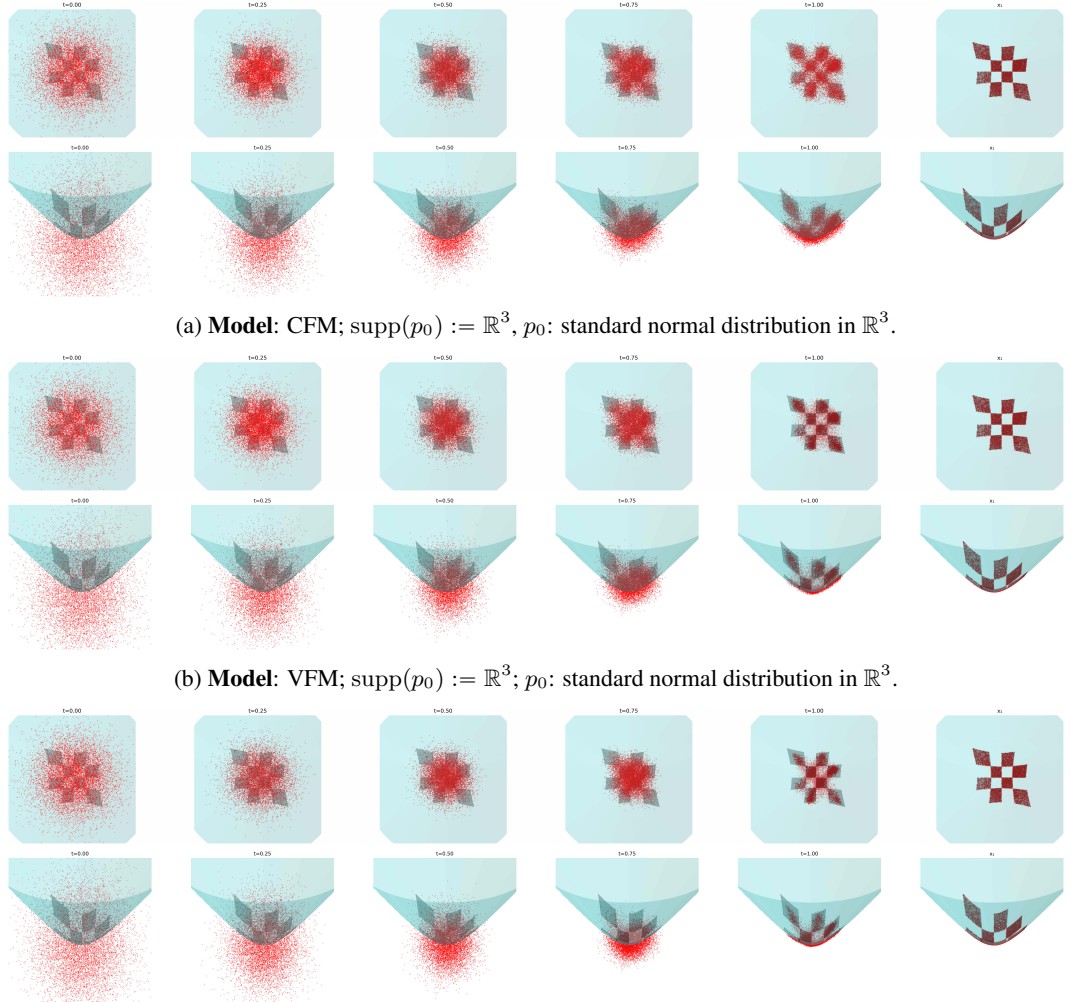

(a) **Model**: CFM; $\mathrm{supp}(p_0) := \mathbb{R}^3$, $p_0$: standard normal distribution in $\mathbb{R}^3$.

(b) **Model**: VFM; $\mathrm{supp}(p_0) := \mathbb{R}^3$; $p_0$: standard normal distribution in $\mathbb{R}^3$.

(c) **Model**: RG-VFM; $\mathrm{supp}(p_0) := \mathbb{R}^3$; $p_0$: standard normal distribution in $\mathbb{R}^3$.

Figure 10: Flow trajectories of 10,000 samples, initially drawn from the noisy distribution $p_0$ at $t = 0$, evolving to reach their final configuration by $t = 1$. In all variational cases, the posterior distribution is **Normal**, and $p_1$ is the checkerboard distribution on $\mathbb{H}^2_{-1}$.

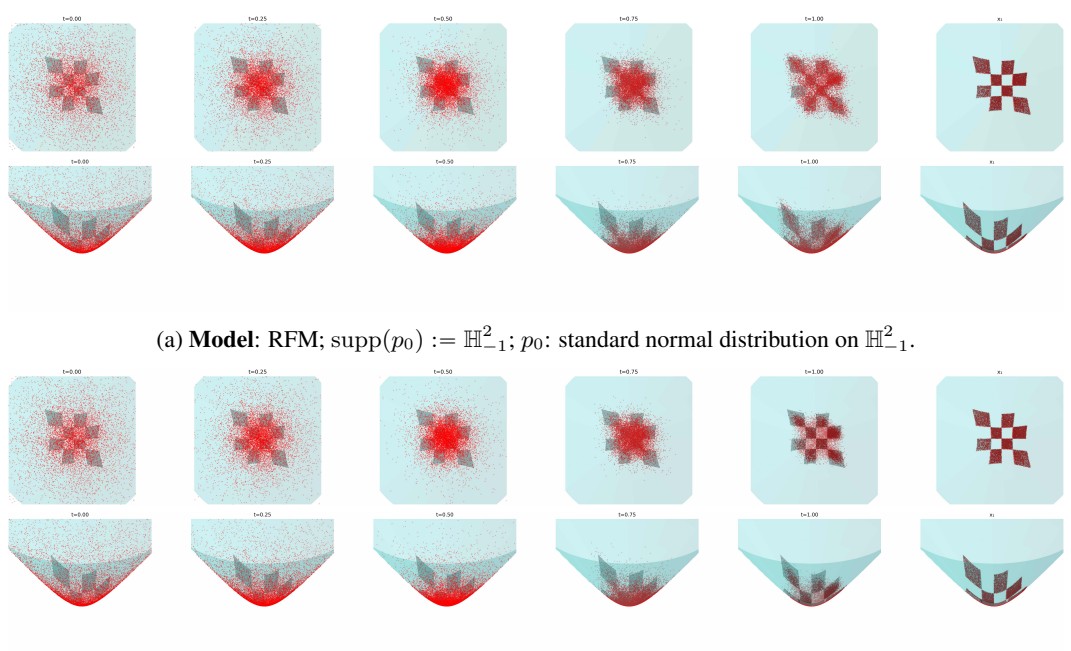

(a) **Model**: RFM; $\mathrm{supp}(p_0) := \mathbb{H}^2_{-1}$; $p_0$: standard normal distribution on $\mathbb{H}^2_{-1}$.

(b) **Model**: RG-VFM; $\mathrm{supp}(p_0) := \mathbb{H}^2_{-1}$; $p_0$: standard normal distribution on $\mathbb{H}^2_{-1}$.

Figure 11: Flow trajectories of 10,000 samples, initially drawn from the noisy distribution $p_0$ at $t = 0$, evolving to reach their final configuration by $t = 1$. In all variational cases, the posterior distribution is **Normal**, and $p_1$ is the checkerboard distribution on $\mathbb{H}^2_{-1}$.

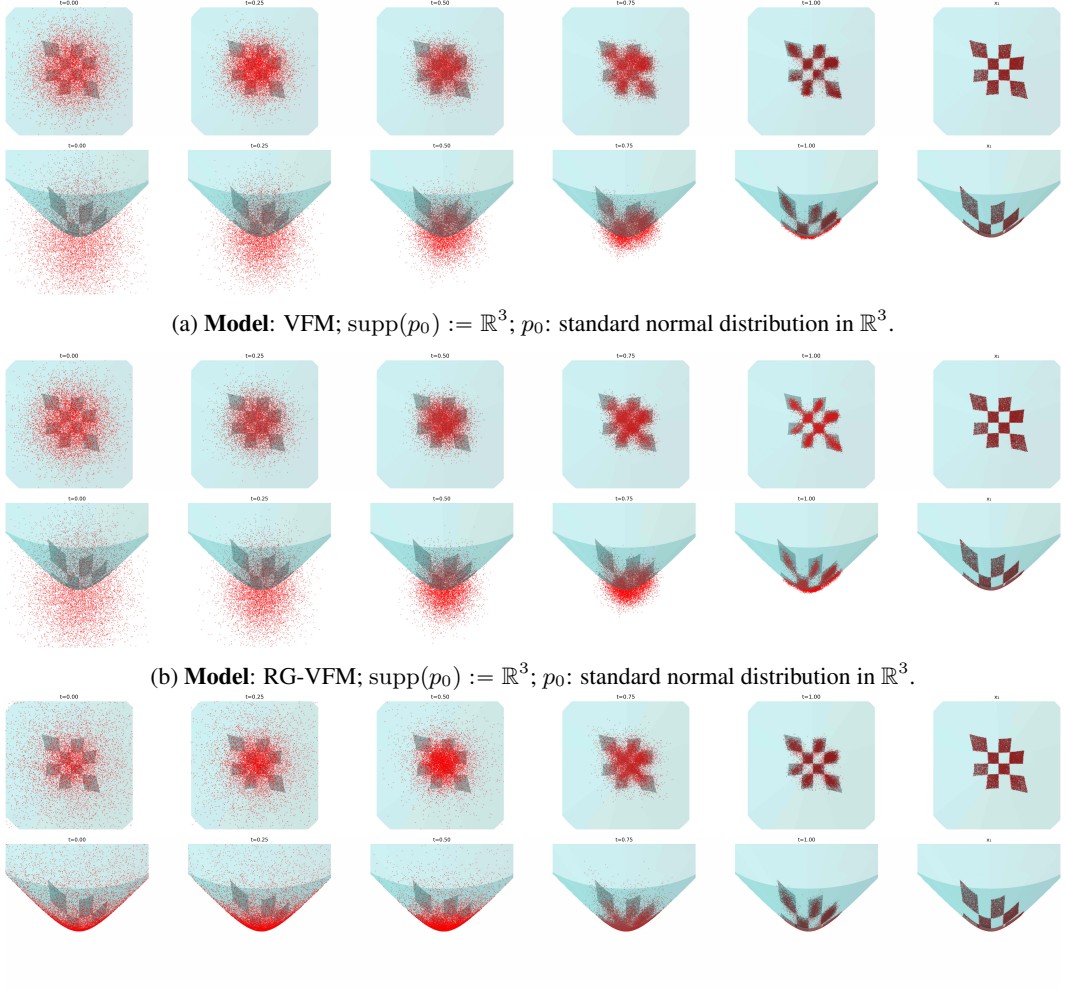

(a) **Model**: VFM; $\mathrm{supp}(p_0) := \mathbb{R}^3$; $p_0$: standard normal distribution in $\mathbb{R}^3$.

(b) **Model**: RG-VFM; $\mathrm{supp}(p_0) := \mathbb{R}^3$; $p_0$: standard normal distribution in $\mathbb{R}^3$.

(c) **Model**: RG-VFM; $\mathrm{supp}(p_0) := \mathbb{H}^2_{-1}$; $p_0$: standard normal distribution on $\mathbb{H}^2_{-1}$.

Figure 12: Flow trajectories of 10,000 samples, initially drawn from the noisy distribution $p_0$ at $t = 0$, evolving to reach their final configuration by $t = 1$. In all variational cases, the posterior distribution is **Laplace**, and $p_1$ is the checkerboard distribution on $\mathbb{H}^2_{-1}$.

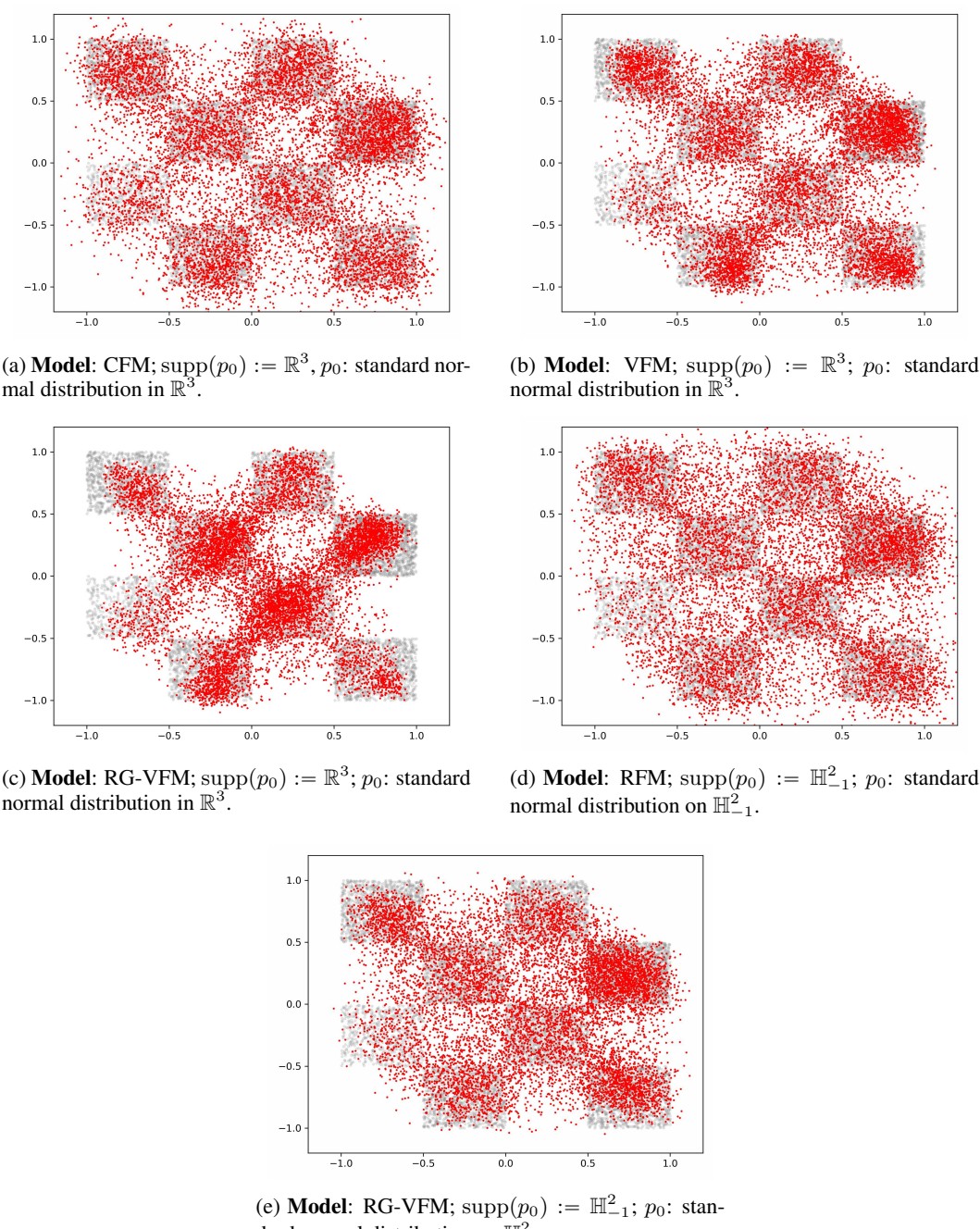

(a) **Model**: CFM; $\mathrm{supp}(p_0) := \mathbb{R}^3$, $p_0$: standard normal distribution in $\mathbb{R}^3$.

(b) **Model**: VFM; $\mathrm{supp}(p_0) := \mathbb{R}^3$; $p_0$: standard normal distribution in $\mathbb{R}^3$.

(c) **Model**: RG-VFM; $\mathrm{supp}(p_0) := \mathbb{R}^3$; $p_0$: standard normal distribution in $\mathbb{R}^3$.

(d) **Model**: RFM; $\mathrm{supp}(p_0) := \mathbb{H}^2_{-1}$; $p_0$: standard normal distribution on $\mathbb{H}^2_{-1}$.

(e) **Model**: RG-VFM; $\mathrm{supp}(p_0) := \mathbb{H}^2_{-1}$; $p_0$: standard normal distribution on $\mathbb{H}^2_{-1}$.

Figure 13: Sample distributions generated by different models (representing the flow configuration at $t = 1$) unwrapped from $\mathbb{H}^2_{-1}$ to $\mathbb{R}^2$ for improved visualization. The true checkerboard distribution is shown in gray in the background. In all variational cases, the posterior distribution is **Normal**, and $p_1$ is the checkerboard distribution on $\mathbb{H}^2_{-1}$.

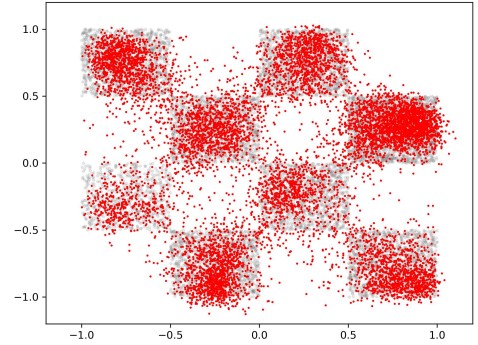

(a) **Model**: VFM; $\mathrm{supp}(p_0) := \mathbb{R}^3$; $p_0$: standard normal distribution in $\mathbb{R}^3$.

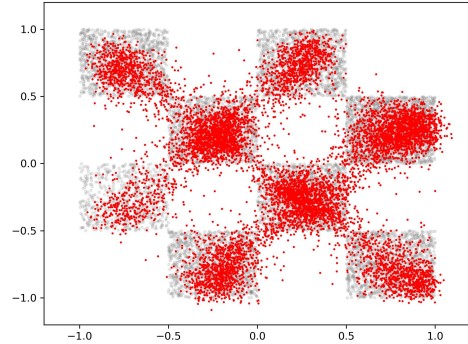

(b) **Model**: RG-VFM; $\mathrm{supp}(p_0) := \mathbb{R}^3$; $p_0$: standard normal distribution in $\mathbb{R}^3$.

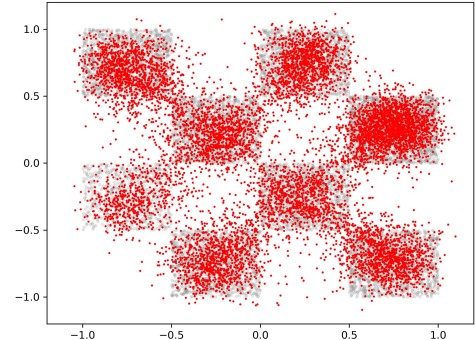

(c) **Model**: RG-VFM; $\mathrm{supp}(p_0) := \mathbb{H}^2_{-1}$; $p_0$: standard normal distribution on $\mathbb{H}^2_{-1}$.

Figure 14: Sample distributions generated by different models (representing the flow configuration at $t = 1$) unwrapped from $\mathbb{H}^2_{-1}$ to $\mathbb{R}^2$ for improved visualization. The true checkerboard distribution is shown in gray in the background. In all variational cases, the posterior distribution is **Laplace**, and $p_1$ is the checkerboard distribution on $\mathbb{H}^2_{-1}$.

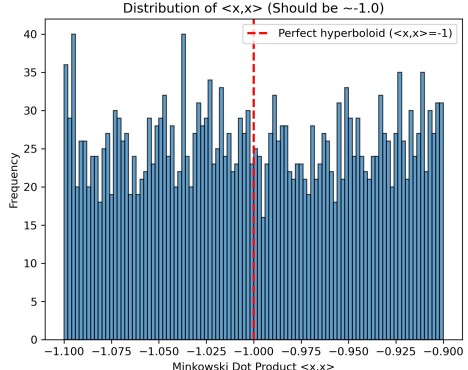

(a) **Model**: CFM; $\mathrm{supp}(p_0) := \mathbb{R}^3$, $p_0$: standard normal distribution in $\mathbb{R}^3$.

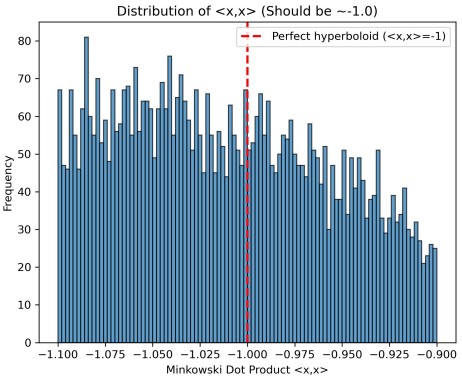

(b) **Model**: VFM; $\mathrm{supp}(p_0) := \mathbb{R}^3$; $p_0$: standard normal distribution in $\mathbb{R}^3$.

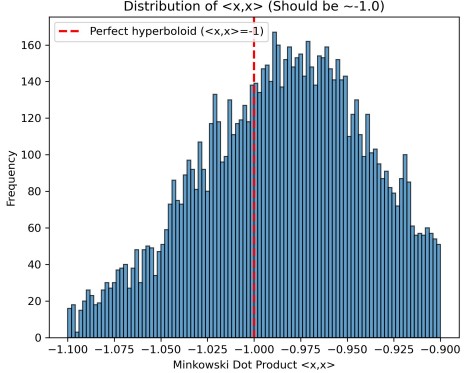

(c) **Model**: RG-VFM; $\mathrm{supp}(p_0) := \mathbb{R}^3$; $p_0$: standard normal distribution in $\mathbb{R}^3$.

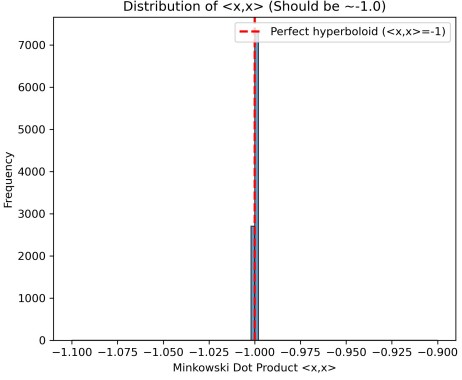

(d) **Model**: RFM; $\mathrm{supp}(p_0) := \mathbb{H}^2_{-1}$; $p_0$: standard normal distribution on $\mathbb{H}^2_{-1}$.

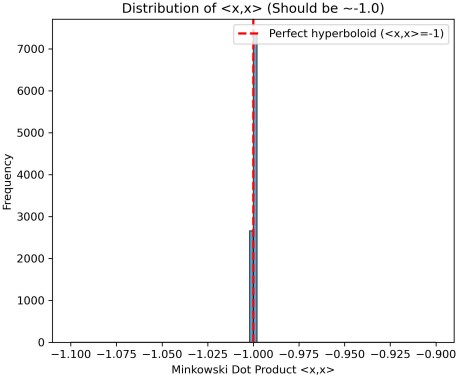

(e) **Model**: RG-VFM; $\mathrm{supp}(p_0) := \mathbb{H}^2_{-1}$; $p_0$: standard normal distribution on $\mathbb{H}^2_{-1}$.

Figure 15: Histogram of the norm values of the 10,000 samples describing the generated distribution. In all variational cases, the posterior distribution is **Normal**, and $p_1$ is the checkerboard distribution on $\mathbb{H}^2_{-1}$.

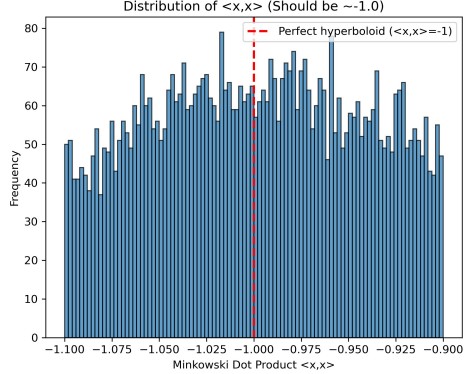

(a) **Model**: VFM; $\mathrm{supp}(p_0) := \mathbb{R}^3$; $p_0$: standard normal distribution in $\mathbb{R}^3$.

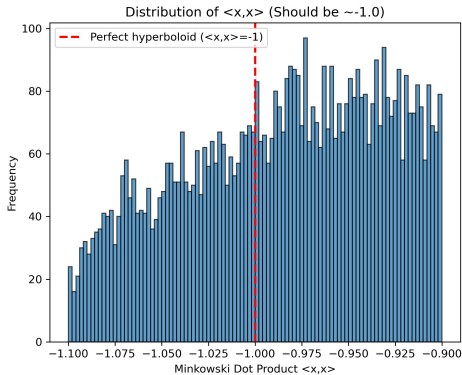

(b) **Model**: RG-VFM; $\mathrm{supp}(p_0) := \mathbb{R}^3$; $p_0$: standard normal distribution in $\mathbb{R}^3$.

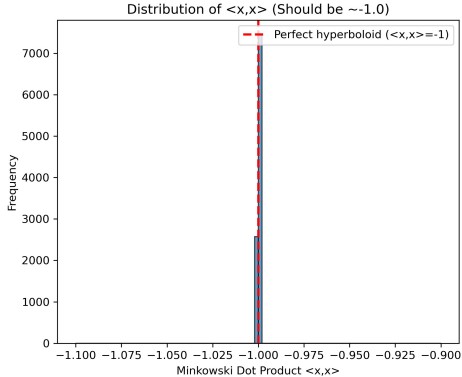

(c) **Model**: RG-VFM; $\mathrm{supp}(p_0) := \mathbb{H}^2_{-1}$; $p_0$: standard normal distribution on $\mathbb{H}^2_{-1}$.

Figure 16: Histogram of the norm values of the 10,000 samples describing the generated distribution. In all variational cases, the posterior distribution is **Laplace**, and $p_1$ is the checkerboard distribution on $\mathbb{H}^2_{-1}$.

## F  MOF GENERATION WITH MOFFLOW

### F.1  EXPERIMENTAL SETUP

**MOFFlow.**  MOFFlow (Kim et al., 2024) is a flow-based generative model for MOF structures that operates at the level of rigid building blocks, i.e., metal nodes and organic linkers. A MOF is represented as $S = (\mathcal{B}, q, \tau, \ell)$, where $\mathcal{B}$ denotes the set of building blocks, and the model learns their roto-translations $(q, \tau)$ together with the lattice parameter $\ell$. Instead of predicting atom-level coordinates, MOFFlow (Kim et al., 2024) treats building blocks as rigid bodies, reducing the search space. The generative model is defined as a conditional normalizing flow $p_\theta(q, \tau, \ell \mid \mathcal{B})$, trained with the Riemannian flow matching framework. Specifically, conditional flows are defined along geodesics in $SE(3)$ and the lattice space as

$$q^{(t)} = \exp_{q^{(0)}}\big(t \log_{q^{(0)}}(q^{(1)})\big), \quad \tau^{(t)} = (1-t)\tau^{(0)} + t\tau^{(1)}, \quad \ell^{(t)} = (1-t)\ell^{(0)} + t\ell^{(1)}, \quad (29)$$

leading to the conditional vector fields

$$u_t(q^{(t)} \mid q^{(1)}) = \frac{\log_{q^{(t)}}(q^{(1)})}{1-t}, \quad u_t(\tau^{(t)} \mid \tau^{(1)}) = \frac{\tau^{(1)} - \tau^{(t)}}{1-t}, \quad u_t(\ell^{(t)} \mid \ell^{(1)}) = \frac{\ell^{(1)} - \ell^{(t)}}{1-t}. \quad (30)$$

Rather than directly modeling these vector fields, Kim et al. (2024) uses a re-parameterized training objective that predicts the clean data $(q_1, \tau_1, \ell_1)$ from an intermediate structure $S^{(t)}$:

$$\mathcal{L}_{\text{MOFFlow}}(\theta) = \mathbb{E}_{\mathbf{S}^{(1)} \sim \mathcal{D}, t \sim \mathcal{U}(0,1)} \big[\lambda_1 \mathcal{L}_q(\theta) \quad + \quad \lambda_2 \mathcal{L}_\tau(\theta) \quad + \quad \lambda_3 \mathcal{L}_l(\theta)\big] = \quad (31)$$

$$= \mathbb{E}_{\mathbf{S}^{(1)} \sim \mathcal{D}, t \sim \mathcal{U}(0,1)} \left[ \lambda_1 \frac{\left\| \log_{q^{(t)}}(\hat{q}_1) - \log_{q^{(t)}}(q_1) \right\|_{SO(3)}^2}{(1-t)^2} + \lambda_2 \frac{\|\hat{\tau}_1 - \tau_1\|_{\mathbb{R}^3}^2}{(1-t)^2} + \lambda_3 \frac{\left\| \hat{\ell}_1 - \ell_1 \right\|_{\mathbb{R}^3}^2}{(1-t)^2} \right]$$

At generation time, samples are drawn from priors on rotations, translations, and lattice parameters, which are then mapped to the full MOF structure by applying the predicted blockwise roto-translations to the input building blocks.

**V-MOFFlow.**  Our contribution consists in adopting a variational perspective in the rotational component of $\mathcal{L}_{\text{V-MOFFlow}}(\theta)$, by only substituting $\mathcal{L}_q(\theta)$ with the following:

$$\tilde{\mathcal{L}}_q(\theta) = \left\| \log_{\hat{q}_1}(q_1) \right\|_{SO(3)}^2, \quad (32)$$

which corresponds to the squared geodesic distance between $q_1$ and $\hat{q}_1$ in $SO(3)$. The definition of the vector fields is unchanged from eqs. (29) and (30), as well as the sampling algorithm.

**Implementation details.**  For reproducing the MOFFlow results (training from scratch) and evaluating our V-MOFFlow model, we follow the exact experimental procedure described in Kim et al. (2024) using their codebase and hyperparameter values, with only the following differences:

1. We use the `Batch` implementation introduced in Kim et al. (2024) instead of `TimeBatch` (Yim et al., 2023b), which processes multiple data instances per batch, leading to reduced computational requirements in terms of training and generation time in GPU hours.

2. In terms of computational resources, we use $2 \times$ 24GB NVIDIA RTX A5000 GPUs instead of $8 \times$ 24GB RTX 3090 GPUs.

Regarding dataset details and train/validation/test split information, we refer the reader to Kim et al. (2024). Furthermore, we choose not to report inference times in table 2, as we find the differences negligible compared to the reported MOFFlow values.

### F.2  ADDITIONAL RESULTS

**Results in property evaluation.**  Following Kim et al. (2024), we evaluate the quality of generated MOF structures beyond match rate and RMSE by analyzing eight key properties: volumetric surface area (VSA), gravimetric surface area (GSA), largest cavity diameter (LCD),

pore limiting diameter (PLD), void fraction (VF), density (DST), accessible volume (AV), and unit cell volume (UCV). We use the same experimental implementation and code as MOFFlow, evaluating models with RMSE and distributional differences.

Results in table 7 compare our model against reported DiffCSP and MOFFlow results from Kim et al. (2024), as well as our reproduced MOFFlow model trained with the `Batch` implementation. V-MOFFlow achieves improved RMSE for half the properties compared to the original MOFFlow paper. Moreover, the reproduced MOFFlow yields slightly higher property values than both the original MOFFlow and V-MOFFlow results. Overall, we believe that the magnitude of most values is too high for meaningful comparison across methods.

Table 7: **Property evaluation.** We report results for DiffCSP and MOFFlow as they are in Kim et al. (2024), and we compute from scratch the properties of the generated samples with the re-trained MOFFlow and V-MOFFlow, that make use of the `Batch` implementation. Average RMSE is computed between the ground-truth and generated structures.

|  | RMSE ↓ | | | |
| --- | --- | --- | --- | --- |
|  | DiffCSP | MOFFlow (Paper) | MOFFlow (Reproduced) | V-MOFFlow (Ours) |
| VSA ($m^2/cm^3$) | 796.9 | **264.5** | 289.9 | 265.0 |
| GSA ($m^2/g$) | 1561.9 | 331.6 | 473.2 | **328.8** |
| AV ($Å^3$) | 3010.2 | **530.5** | 1935.1 | 714.2 |
| UCV ($Å^3$) | 3183.4 | **569.5** | 2108.5 | 785.8 |
| VF | 0.2167 | 0.0285 | 0.0379 | **0.0263** |
| PLD (Å) | 4.0581 | 1.0616 | 1.2434 | **1.0337** |
| LCD (Å) | 4.5180 | 1.1083 | 1.2613 | **1.0888** |
| DST ($g/cm^3$) | 0.3711 | **0.0442** | 0.0747 | 0.0446 |

**Effect on integration steps.** Following Kim et al. (2024), we investigate how the number of sampling integration steps affects both V-MOFFlow and our reproduced MOFFlow model (the one from section 5.2). We randomly select 1000 structures from the test set and evaluate match rate and RMSE across varying integration steps: [2, 5, 7, 10, 50, 100, 200, 500, 1000], using the same experimental procedure as Kim et al. (2024). The results in fig. 17 show that both models exhibit similar trends, with performance peaking around 10 and 50 integration steps before slightly declining at higher step counts. The main difference between the models is the performance gap rather than the overall trend, reflecting the difference in accuracies obtained on the entire test set (table 2).

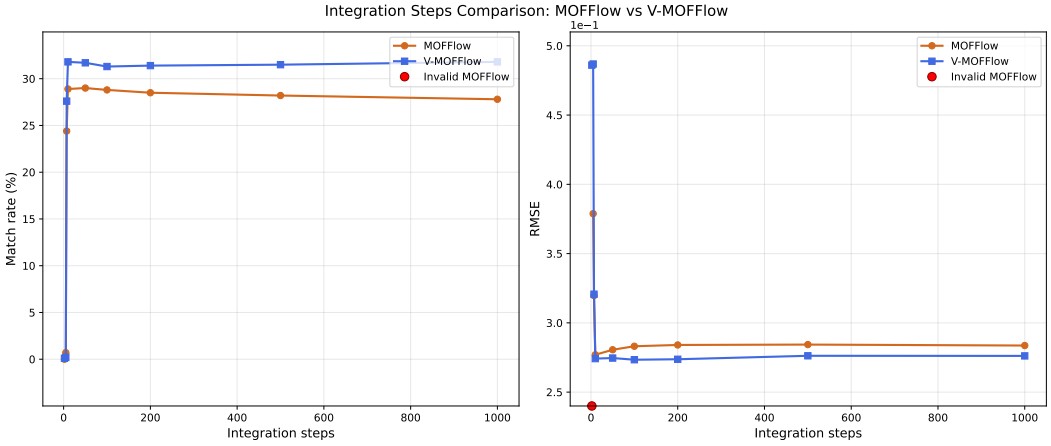

Figure 17: Comparison between MOFFlow (the reproduced one from section 5.2) and V-MOFFlow in terms of match rate and RMSE over different timestep values: [2, 5, 7, 10, 50, 100, 200, 500, 1000].

