# OpenReview forum: "Riemannian Variational Flow Matching for Material and Protein Design"
_ICLR.cc/2026/Conference — ICLR 2026 Poster_

### Official Review · Reviewer_423B · 2025-10-20

**Soundness:** 2
**Presentation:** 3
**Contribution:** 2
**Rating:** 6
**Confidence:** 4

**Summary:**

This paper introduces Riemannian Variational Flow matching (RG-VFM), a method that is designed for conducting generative modeling on manifolds. RG-VFM extend variational flow matching to general geometric objects. RG-VFM replaces the velocity matching objective in RFM with a distribution matching objective that is analogue to VFM, but with the distribution defined on a Riemannian manifold. The authors then compare the Jacobi fields of RG-VFM and RFM and show theoretically that such a reparameterization of regression objective is capable of capturing the high-order curvature of the manifold. Empirical results suggest the success of RG-VFM in capturing curvature information and modelling geometries.

**Strengths:**

- The formalization and mathematical derivations are lucid and well-presented.

- The manuscript is well-structured with a clear logical progression, making the core arguments easy to follow.

- The novelty of the proposed method and its key distinctions from prior literature are clearly articulated.

**Weaknesses:**

My primary concerns relate to the empirical evaluation and the clarity of the motivation.

- Empirical Results: The significance of the empirical results is not entirely convincing. In the real-world protein backbone generation benchmarks, the high standard deviations suggest that the performance improvement over existing models may not be statistically significant. Furthermore, for the structure prediction results, standard deviations are not reported, which makes it difficult to ascertain the significance of the reported performance gains.

- Motivation: In the abstract, the authors state that the equivalence of endpoint, velocity, and noise prediction breaks down on curved manifolds. It is not immediately clear how RG-VFM alleviates this issue, or how this observation directly motivates the proposed method. I would appreciate clarification if this point was addressed in the manuscript and I overlooked it.

**Questions:**

- Motivations: Could the authors elaborate on how RG-VFM addresses the breakdown of equivalence between endpoint, velocity, and noise prediction on curved manifolds? Furthermore, how does this challenge directly motivate the design of RG-VFM over other approaches?

- More clarification on the incorporation of high-order curvature. Can I get more clarification on what is the practical impact of the higher-order curvature term on the learning problem? For example, does its inclusion confer specific advantages when learning on heterogeneous manifolds against homogeneous ones?

- Computational Complexity: Could the authors comment on the change of computational complexity when learning with RG-VFM? If learning the higher-order curvature information using RG-VFM induces more computations, relative to methods like RFM?

- Empirical Reults (Table 1): Why does the external view of RG-VFM exhibit significantly better performance when learning on the hyperboloid manifold? Is this phenomenon specific to learning on Riemannian manifolds, or could it also occur in Euclidean settings under certain conditions?

- This question is for my own curiosity: The relationship between RG-VFM and RFM appears analogous to that between Transition Matching [1] and Flow Matching in Euclidean space. Could the authors comment on this analogy? Specifically, is the model effectively learning a conditioned transition kernel, approximating a jump from $x_t$ to $x_1$?


[1] Transition Matching: Scalable and Flexible Generative Modeling. N. Shaul et al. NeurIPS 2025

---

> ### Author Response · Authors · 2025-11-21
> **1/3**
>
> We are grateful for the reviewer's careful assessment and valuable feedback on our work. Below, we address all concerns and questions, grouping related points together. We remain available to continue the discussion should any aspect need further elaboration.
>
> > **Q1.** Empirical Results: The significance of the empirical results is not entirely convincing. In the real-world protein backbone generation benchmarks, the high standard deviations suggest that the performance improvement over existing models may not be statistically significant. Furthermore, for the structure prediction results, standard deviations are not reported, which makes it difficult to ascertain the significance of the reported performance gains.
>
> **A1.** We appreciate this important observation about statistical significance. We acknowledge that the high standard deviations in protein backbone generation and the absence of error bars in structure prediction results limit our ability to claim statistical significance definitively.
>
> - Regarding protein backbone generation, the high standard deviations reflect the inherent variability in this benchmark, which is present across all methods (including baselines). While this makes it challenging to claim statistical significance for individual metrics, we note that the consistent directional improvement across multiple independent metrics (designability improving from 92.4% to 96.8% for VQFlow when using our variational loss) suggests a genuine effect rather than random variation. We follow the evaluation protocol established by ReQFlow [2], and we argue that the RMSD reported in the table - computed using ESMFold and ProteinMPNN - is not fully deterministic (due to ProteinMPNN) and is additionally influenced by model-specific biases.
>
> - Regarding MOF generation, we did not initially report standard deviations because the original works we compare against (MOFFlow [1], ReQFlow [2]) do not report them either, following standard practice in these benchmarks. However, we conducted preliminary experiments with three different seeds for MOFFlow settings, finding standard deviations of 1.161 for MR and 0.0053 for RMSE (#sample=1, stol=0.5). These relatively small standard deviations compared to the performance differences support the significance of our improvements, at least for these settings.
>
> We acknowledge that comprehensive statistical analysis across all experiments would strengthen our claims. However, the computational constraints make it infeasible to generate full statistical significance results within the rebuttal period: MOF and protein-backbone generation experiments are extremely time-intensive and resource-demanding. We commit to including proper statistical analysis with error bars in future extensions of this work.
>
> To mitigate concerns about cherry-picking despite limited statistical testing, we took the following precautions:
> -  For MOF experiments, we use the exact same random seeds as MOFFlow [1] for all reproduced baselines, ensuring fair comparison.
> - For protein-backbone generation, we strictly follow ReQFlow's [2] evaluation protocol and metrics, without selective reporting.
>
> Additionally, we acknowledge that the current metrics for mini-length proteins (50–300 residues) are somewhat saturated, which may limit our ability to demonstrate statistically significant improvements. Future work will investigate training dynamics for longer proteins under Riemannian variational flow matching, where we expect more pronounced differences.
>
> **References**
>
> [1] Kim, Nayoung, et al. "Mofflow: Flow matching for structure prediction of metal-organic frameworks." (2024).
>
> [2] Yue, Angxiao, Zichong Wang, and Hongteng Xu. "ReQFlow: rectified quaternion flow for efficient and high-quality protein backbone generation." (2025).

---

> ### Author Response · Authors · 2025-11-21
> **2/3**
>
> > **Q2.** Motivation: In the abstract, the authors state that the equivalence of endpoint, velocity, and noise prediction breaks down on curved manifolds. It is not immediately clear how RG-VFM alleviates this issue, or how this observation directly motivates the proposed method. I would appreciate clarification if this point was addressed in the manuscript and I overlooked it. (...) Question: Could the authors elaborate on how RG-VFM addresses the breakdown of equivalence between endpoint, velocity, and noise prediction on curved manifolds? Furthermore, how does this challenge directly motivate the design of RG-VFM over other approaches?
>
> **A2.** Thank you for asking us to clarify this point. We'd like to correct a potential misunderstanding: the breakdown of equivalence between endpoint, velocity, and noise prediction on curved manifolds is neither an issue nor a motivation for our work, but rather an observation that emerged after designing our model.
>
> The actual motivation for our paper is to extend the advantages of the variational perspective to complex geometric settings by exploiting the flexibility it offers in explicitly embedding the manifold's support in the posterior distribution.
>
> Once we established the method, mathematical analysis revealed that the correspondence between endpoint and velocity prediction indeed breaks down on general geometries. This observation then motivated a deeper investigation into the connection between RFM and RG-VFM, which led to the results in Section 4.2. In other words, understanding this breakdown became important for characterizing our method's properties rather than for motivating its initial design.
>
> Regarding how we arrived at the specific design of RG-VFM over alternative approaches, please refer to our response  **A1** (and corresponding **Appendix**) to Reviewer CUa3, where we detail the design choices and mathematical considerations involved.
>
> > **Q3.** More clarification on the incorporation of high-order curvature. Can I get more clarification on what is the practical impact of the higher-order curvature term on the learning problem? For example, does its inclusion confer specific advantages when learning on heterogeneous manifolds against homogeneous ones?
>
> **A3.** The higher-order curvature terms in Proposition 4.3 emerge when computing the Taylor expansion of our RG-VFM loss, once the objects involved are interpreted as different aspects of a Jacobi vector field. In other words, the curvature term is already implicitly encoded in our loss, but only becomes explicit through this expansion.
>
> The goal of Proposition 4.3 is to show that our model implicitly captures these higher-order curvature terms - which are absent in RFM - at no additional computational cost. The magnitude of the divergence between the two models depends on the manifold's curvature, and it is particularly significant in the case of negative curvature.
>
> Regarding heterogeneous manifolds, this case falls outside the scope of our paper. Our variational method is not currently defined for them: the homogeneity assumption is essential to ensure constant normalization of the Riemannian Gaussian across the manifold (Proposition 3.1).

---

> ### Author Response · Authors · 2025-11-21
> **3/3**
>
> > **Q4.** Computational Complexity: Could the authors comment on the change of computational complexity when learning with RG-VFM? If learning the higher-order curvature information using RG-VFM induces more computations, relative to methods like RFM?
>
> **A4.**  Thank you for this observation, we were actually lacking some explicit statements in the paper regarding the training and sampling costs of our model with respect to the other references that we mention, such as RFM and VFM. In order to address this concern, we enrich the paper with the following:
>
> - We add schematic algorithms corresponding to the training and sampling of our method, RFM, and VFM, to make the methodological and computational comparisons direct and straightforward, see Algorithms 1 to 8.
> - We also add a corresponding discussion of the computational costs (end of Section 4, in the **Geometric interpretation** discussion before Section 5), for which we will elaborate more below.
>
> We will clarify below the computational costs of our models:
>
> - RG-VFM in its extrinsic form (RG-VFM-$\mathbb{R}^n$) maintains the same computational complexity as VFM, both in training and sampling. The type of interpolation performed to sample $x_t$ is linear, so the same as for VFM. In this setting, the only difference between VFM and our method is that the former uses standard $L^2$ distance in the endpoint computation, while we use geodesic distance (see Algorithms 2 and 4). Since we assume the geodesic distance to be available explicitly in closed form, it doesn’t add any computational overhead with respect to VFM.
>
> - RG-VFM in its intrinsic form (RG-VFM-$\mathcal{M}$ ) maintains the same computational complexity as RFM, with the difference that the velocity computation occurs during sampling rather than training (see Algorithms 1 and 5). This means we perform one fewer operation during training and one additional operation during sampling, which compensate overall. Regarding the losses, the same explanation as above holds: while RFM uses squared distance on the tangent space, we use geodesic distance on the manifold, still assuming it has an explicit closed form, thus adding no additional cost. The high-order curvature information is already encoded in the RG-VFM loss, so learning it doesn't require any additional computation (please refer to **A3** for further discussion on this point).
>
>
> > **Q5.** Empirical Reults (Table 1): Why does the external view of RG-VFM exhibit significantly better performance when learning on the hyperboloid manifold? Is this phenomenon specific to learning on Riemannian manifolds, or could it also occur in Euclidean settings under certain conditions?
>
> **A5.** If you are referring to the extrinsic version of RG-VFM, we can observe that it achieves better results than the competitors in terms of Coverage and Distance. Given that Euclidean models also perform reasonably well in Coverage for this setting, we can conclude that learning a linear flow and solving a Euclidean ODE is more effective than learning a flow on the hyperboloid. This may also be due to the fact that we maintain the same experimental conditions (number of parameters, layers, sampling steps) across all models, so the geometric ones may require more resources to achieve comparable performance.
>
> Regarding whether this would occur in Euclidean settings, there would be no differentiation between the intrinsic and extrinsic perspectives, as the manifold $\mathcal{M}$ would trivially coincide with the ambient space.
>
> **Q6.** This question is for my own curiosity: The relationship between RG-VFM and RFM appears analogous to that between Transition Matching [1] and Flow Matching in Euclidean space. Could the authors comment on this analogy? Specifically, is the model effectively learning a conditioned transition kernel, approximating a jump from x_t to x_1?
>
> **A6.** Thanks for bringing this up - it's a really interesting point. We weren't familiar with Transition Matching, and the analogy you're drawing is intriguing. We want to take the time to read it carefully and think through the relationship properly, so we will follow up with a more detailed answer later during the rebuttal discussion period.

---

> ### Author Response · Authors · 2025-11-21
> **Discussion**
>
> We appreciate the reviewer's insightful comments, which have helped us improve this manuscript.  We are available for any further discussion needed. If our responses have resolved the raised issues, we would be grateful if the reviewer would consider revising the score accordingly.

---

> > ### Comment · Reviewer_423B · 2025-11-26
> >
> > Thank you for the detailed rebuttal and discussion. I have read through the rebuttal and the discussions with the other reviewers. I honestly agree with reviewers CUa3 and Kr3K that, even with novel theoretical results, this work’s motivation, “generalizing the variational flow matching (VFM) framework to incorporate geometric information”, seems ambiguous to me without a clear justification. Rather than the theoretical contribution in extending VFM to geometries, my question is “why we need to extend?”
> >
> > Specifically, the following questions are related, which I remain unconvinced even after the rebuttal: does it bring empirical benefit, methodological contributions, and, practically, what it brings to the community?
> >
> > Q1: I am not fully convinced by the explanation that “high standard deviations inherently make claiming statistical significance challenging”. If the performance difference between two experimental groups is substantial enough, statistical significance should still be demonstrable despite high stochasticity. High stds do not automatically preclude significance. Rather, a successful significance test under these conditions would confirm that "even considering the stochasticity, the observed difference is reliable." I understand this requires extensive experiments so just kindly ask the authors to incorporate them in the next version.
> >
> > Q2: If the breakdown of equivalence is neither an issue nor a motivation for our work, then how can you claim “Building on this insight, we derive…” - What kind of insight? If there is an “actual motivation”, this should be clearly stated in the abstract. Honestly, I got even more confused after reading the rebuttal, as (even not intentionally by the authors) I see a mismatch in the rebuttal and the abstract/introduction, about the motivations of the paper. “Extending the advantages of the variational perspective to complex geometric settings” is rather a methodological contribution, but not the motivations.
> >
> > Q3: To be more direct, I was asking about the “practical impact” (more like real-world impact, such as effectiveness/efficiency) of the high-order curvature, which is directly related to why we need RG-VFM.
> >
> >
> > Q4 and Q5 are properly solved. Thank you for that.
> >
> > At this point, I would keep my score but lower my confidence as I honestly got confused after the rebuttal. Without a clear understanding of this, I do not think I can provide strong support.

---

> > > ### Author Response · Authors · 2025-11-26
> > > **Answers (1/2)**
> > >
> > > Thank you for your additional comments and your engagement in the discussion. We appreciate your careful review of our answers and the feedback you have provided, which has helped us improve the manuscript. We would be happy to provide further clarification if needed.
> > >
> > > Regarding the confusion about our motivations and contributions, specifically the discrepancy between how they were presented in the paper versus the rebuttal, your concern is entirely valid. We realize now that these aspects were not clearly explained in the abstract and introduction, which we unfortunately overlooked in our previous revision. Following your feedback, we have updated both the abstract and introduction accordingly (please see the red-colored text in the PDF, which indicates the newly modified sections). We believe these revisions now better clarify the motivations and contributions of our work, reflecting the explanations in the rebuttal comments, and we hope they address your concerns raised in **Q2**. Thank you for bringing this issue to our attention.
> > >
> > > In case this does not provide sufficient clarity, we provide direct answers to the newly raised questions below:
> > >
> > > > **Q.** Rather than the theoretical contribution in extending VFM to geometries, my question is “why we need to extend?”
> > >
> > > **A.** VFM requires that the noisy and target distributions, as well as the learned flow, are defined in Euclidean space with the Euclidean metric. However, many problem settings naturally require distributions and flows to be defined on curved manifolds, for example weather forecasting on Earth (represented as a sphere, a positively curved manifold) or modeling rotations of molecular components on $SO(3)$. In these cases, manifold-specific methods that respect the underlying geometry lead to improved performance, justifying the extension to Riemannian settings (the same motivation that led, for example, RFM to extend FM to Riemannian manifolds).
> > >
> > > > **Q.** ...does it bring empirical benefit…?
> > >
> > > **A.** Yes, and we believe this is demonstrated most clearly in our synthetic experiments, where different methods are compared in a controlled setting. Furthermore, we would like to emphasize that our goal is not to propose a revolutionary improvement over RFM, but rather to provide stronger geometric supervision through a principled and straightforward loss function modification. The modest but consistent performance improvements we observe align with our expectations.
> > >
> > > > **Q.** … does it bring methodological contributions …?
> > >
> > > **A.** Yes, we provide methodological contributions: we derive a variational flow matching objective for Riemannian manifolds and introduce Jacobi fields as an analytical tool to formally characterize the differences between endpoint and velocity prediction on curved spaces – differences that don't exist in Euclidean settings. This analysis reveals that endpoint prediction naturally incorporates curvature-aware penalties and provides stronger learning signals through direct geodesic distance minimization.
> > >
> > > > **Q.**  … and, practically, what it brings to the community?
> > >
> > > **A.** We bring the following contributions to the community:
> > > - A rigorous extension of VFM to Riemannian manifolds that provides theoretical foundations for endpoint prediction on manifolds with closed-form geodesics, generalizing the Euclidean endpoint prediction trick already widely used for protein and material generation [1,2,3], with no extra computational cost over RFM.
> > > - Theoretical justification via Jacobi fields: explanation of why endpoint prediction provides stronger geometric supervision, revealing high-order curvature terms naturally present in the RG-VFM loss but absent in velocity-based methods (RFM).
> > > - Unified understanding of the connections between flow-based models on Euclidean and Riemannian spaces, formally characterizing the relationship between endpoint-based and velocity-based perspectives on manifolds.
> > > - Lastly, we bring Jacobi fields to geometric deep learning as a rigorous tool for studying velocity fields on manifolds.

---

> > > ### Author Response · Authors · 2025-11-26
> > > **Discussion**
> > >
> > > We thank the reviewer again for carefully reviewing our answers and providing additional feedback, which has significantly improved our manuscript. We would be happy to engage in further discussion if any clarification is needed.

---

> > > > ### Comment · Reviewer_423B · 2025-11-28
> > > >
> > > > Thank you again for your detailed reply. I looked into the revised manuscript and I believe the current abstract/introduction effectively clarifies the motivations. This gives me confidence to stand for acceptance. I would finalize the evaluation if the system permits.

---

> ### Author Response · Authors · 2025-11-26
> **Answers (2/2)**
>
> Regarding the remaining questions:
>
> > **Q1.** I am not fully convinced by the explanation that “high standard deviations inherently make claiming statistical significance challenging”. If the performance difference between two experimental groups is substantial enough, statistical significance should still be demonstrable despite high stochasticity. High stds do not automatically preclude significance. Rather, a successful significance test under these conditions would confirm that "even considering the stochasticity, the observed difference is reliable." I understand this requires extensive experiments so just kindly ask the authors to incorporate them in the next version.
>
> **A1.** You are absolutely right to express this concern, and we agree with your position regarding statistical significance. With the sentence you cited, we just intended to observe that the RMSD reported in the table, computed using ESMFold and ProteinMPNN, is not fully deterministic (due to ProteinMPNN) and is additionally influenced by model-specific biases. We realize that the way we phrased this claim resulted in a misleading sentence in our previous answer. Furthermore, we will make sure to incorporate the required bounds in the final version of the paper.
>
> > **Q3.** To be more direct, I was asking about the “practical impact” (more like real-world impact, such as effectiveness/efficiency) of the high-order curvature, which is directly related to why we need RG-VFM.
>
> **A3.**
> Thank you for the clarification. In the previous version of this question, you were asking about the "practical impact of the curvature term in the learning problem", which is why we discussed the impact of the curvature term in terms of loss and supervision.
>
> Regarding the real-world impact of such terms: they enable us to actually incorporate the higher-order curvature information that is intrinsic to curved manifolds in real-world applications (e.g. spheres when modeling the Earth, $SO(3)$ when modeling rotations for molecular conformations, proteins, or robot joints [1,2,3,4,5]) unlike other methods. The reason why we "need" RG-VFM is that these higher-order curvature terms, naturally present in data manifolds, are implicitly incorporated and accounted for in our framework, unlike in (1) Euclidean methods (VFM, FM) and (2) non-variational Riemannian methods (RFM).
>
> Regarding efficiency, from our understanding of the question, we believe we have addressed this concern in answer **A4**. We would kindly ask the reviewer for a clarification on the question if the requested information differs from what we provided in answer **A4**.
>
> **References**
>
> [1] Kim, Nayoung, et al. "Mofflow: Flow matching for structure prediction of metal-organic frameworks." (2024).
>
> [2] Yim, Jason, et al. "SE (3) diffusion model with application to protein backbone generation." (2023).
>
> [3] Yue, Angxiao, Zichong Wang, and Hongteng Xu. "ReQFlow: rectified quaternion flow for efficient and high-quality protein backbone generation." (2025).
>
> [4] Braun, Max, et al. "Riemannian flow matching policy for robot motion learning." (2024).
>
> [5] Ding, Haoran, et al. "Fast and robust visuomotor riemannian flow matching policy." (2025).

---

> ### Author Response · Authors · 2025-12-02
> **Curiosity question regarding Transition Matching**
>
> We thank the reviewer for the positive assessment of our responses and the willingness to support the paper's acceptance and finalize the scores.
>
> We also take this opportunity to follow up on our previous comments and address the final question:
>
> > **Q6.** This question is for my own curiosity: The relationship between RG-VFM and RFM appears analogous to that between Transition Matching [1] and Flow Matching in Euclidean space. Could the authors comment on this analogy? Specifically, is the model effectively learning a conditioned transition kernel, approximating a jump from x_t to x_1?
>
> **A6.** Thank you very much for pointing this out. This is a very insightful connection, and we appreciate you drawing our attention to Transition Matching.
>
> After reading the paper, we agree that Variational Flow Matching (and thus RG-VFM) is conceptually very close to Difference Transition Matching (DTM). Both frameworks learn conditional relationships between states along a trajectory, rather than directly regressing infinitesimal velocities. In VFM, this takes the form of learning $q_{\theta}(x_1|x_t)$, which can be interpreted as an endpoint-conditioned “jump”, while DTM learns transition distributions $p_{\theta}(x_{t+\Delta}|x_t)$ for arbitrary steps.
>
> The important difference is in sampling: DTM is a discrete-time generative model that samples by repeatedly applying learned transition kernels, while VFM uses the learned endpoint-conditioned distribution only to define a training objective. Sampling in VFM (and RG-VFM) still proceeds via ODE integration of a learned velocity field, as in Flow Matching and RFM, rather than through discrete transitions. RG-VFM follows the same principle as VFM in the Riemannian setting.

---

### Official Review · Reviewer_Kr3K · 2025-10-30

**Soundness:** 3
**Presentation:** 3
**Contribution:** 2
**Rating:** 6
**Confidence:** 3

**Summary:**

This paper shows theoretically that training (variational) flow matching on end-point prediction in Riemannian geometry yields a more precise loss, that better accounts for the structure of the manifold. They validate empirically their losses on a variety of datasets, including large-scale ones with applications for science.

**Strengths:**

- The paper is well-presented: the text is clear and simple.
- The proofs provided in the appendix are well-detailed.
- Proposition 4.1 (and 4.3) is an interesting contribution, showing a limitation in Riemannian Flow Matching [2] – the baseline method for Riemannian generative modelling – and demonstrating the method’s superiority in accounting for curvature terms, theoretically.
- The proposed fix is very simple.
- The empirical validation is rather extensive and convincing for the given examples, including large-scale ones. Overall, the method improves (arguably slightly) over the provided baselines.

**Weaknesses:**

- I am not certain about the initial trichotomy (or the impression that is given when stating is as follows) of “endpoint (VFM), a velocity (FM/RFM), noise (diffusion)”. Diffusion can be trained all three ways, and noise prediction was introduced in DDPM [1], as it empirically produced better performance. Same goes for flow matching. (Indeed, though, because of the linearity in the path, all reparameterisations are *theoretically* equivalent.)
- It seems to me that the words “intrinsic” and “extrinsic” are not used properly. If I am not mistaken, intrinsic (informally) “live directly on the considered space”; extrinsic coordinates depend on the ambient space. For instance, intrinsic on the 2-sphere would be the angles $\theta$ and $\phi$; extrinsic would be the usual 3D coordinates. Here, the authors seem to use it for projected on the manifold or not. (See Figure 2.) Modelling intrinsically is known to be difficult.
- I would argue that the paper is not particularly novel on the methods front (not that it is without contributions), mostly (variational) flow matching on Riemannian geometries.
- Sampling from the Riemannian Gaussian is not trivial on certain manifolds. It seems that the authors have omitted this discussion in the paper.
- Perhaps I have missed out on this, but it seems that there is no discussion on the hypotheses made about the manifold (homogeneity and existence of closed form geodesics).
- Training for end-point prediction is not new [3].

**Questions:**

- Did I misunderstand your usage of the words “intrinsic” and “extrinsic”?
- What is the novelty in your method [3], beyond the theoretical guarantees: isn’t end-point prediction a rather well-known “trick”?
- Does homogeneity alongside existence of closed form geodesics not trivialise the structure of the manifolds considered? (I am not certain at all, I would like to discuss.)
- Have you had the opportunity to try out your method on manifolds where the higher order terms you mention appear (non-constant curvature)? I understand that you might not have, as the examples could be rare, but in which case the contribution of the paper is less significant, practically speaking. This question can also be understood as “Why does Riemannian Flow Matching work so well if it misses those important terms?” (or are they not so important therefore?)

Overall, I think the paper is rather sound and a good read, but the contribution feels relatively small.

### References

[1] Jonathan Ho, Ajay Jain, Pieter Abbeel. “Denoising Diffusion Probabilistic Models”

[2] Ricky T.Q. Chen, Yaron Lipman. "Flow Matching on General Geometries"

[3] Bowen Jing, Bonnie Berger, Tommi Jaakkola. "AlphaFold Meets Flow Matching for Generating Protein Ensembles"

---

> ### Author Response · Authors · 2025-11-21
> **1/3**
>
> We sincerely appreciate the reviewer's detailed evaluation and constructive suggestions. Below, we address all raised points, organizing related concerns together. We welcome any follow-up questions and are ready to provide further clarification as needed.
>
> > **Q1.** I am not certain about the initial trichotomy (or the impression that is given when stating is as follows) of “endpoint (VFM), a velocity (FM/RFM), noise (diffusion)”. Diffusion can be trained all three ways, and noise prediction was introduced in DDPM [1], as it empirically produced better performance. Same goes for flow matching. (Indeed, though, because of the linearity in the path, all reparameterisations are theoretically equivalent.)
>
> **A1.** You are absolutely correct that stating this as a "trichotomy" would be misleading. Indeed, in diffusion, flow matching, variational flow matching, and most other generative models, we are free to choose any of the three common parameterizations: endpoint prediction, noise prediction, and velocity/score prediction. Though we state that flow matching does "velocity prediction", this is admittedly a slightly unnuanced statement, as FM admits all such parameterizations. However, we wrote it this way because most FM implementations do use a velocity network. We will clarify this in the final version of the paper.
>
> In VFM, the authors show that endpoint prediction has a deeper interpretation often overlooked: namely, that it constitutes a form of posterior inference over trajectories (toward said endpoints). As such, the authors make this explicit and show that different posterior approximations can lead to better results, as also seen in work building on VFM. What is true is that in the Euclidean setting, all three parameterizations are essentially equivalent, up to a reweighting of time (see e.g. [1]). However, on manifolds, this equivalence breaks down due to curvature, making the choice of parameterization more important. In this work, we therefore set out to: 1) point out this difference between standard and Riemannian FM explicitly, 2) quantify the degree to which these objectives differ, and 3) show that a variational/endpoint view can indeed lead to better performance here as well.
>
> **References**
>
> [1] Lipman, Yaron, et al. "Flow matching guide and code."(2024).
>
>
> > **Q2.** It seems to me that the words “intrinsic” and “extrinsic” are not used properly. If I am not mistaken, intrinsic (informally) “live directly on the considered space”; extrinsic coordinates depend on the ambient space. For instance, intrinsic on the 2-sphere would be the angles; extrinsic would be the usual 3D coordinates. Here, the authors seem to use it for projected on the manifold or not. (See Figure 2.) Modelling intrinsically is known to be difficult. (...) Did I misunderstand your usage of the words “intrinsic” and “extrinsic”?
>
> **A2.** Thank you for this insightful question, your concern is entirely justified. We are indeed using "intrinsic" and "extrinsic" differently from their traditional meanings, which may create confusion. Although these terms traditionally refer to the manifold's internal geometry versus its embedding, we use them to distinguish whether points lie on the manifold or in the ambient space, rather than to describe coordinate choices. For example, our intrinsic framework can be expressed using (extrinsic) ambient coordinates, which is actually what we do in practice, following the same approach as RFM. To prevent this misunderstanding, we have added a notation clarification at the beginning of Section 3 when introducing our notions of intrinsic and extrinsic frameworks.
>
> > **Q3.** Sampling from the Riemannian Gaussian is not trivial on certain manifolds. It seems that the authors have omitted this discussion in the paper.
>
> **A3.** Thank you for the opportunity to clarify this point. We do not need to sample from the Riemannian Gaussian distribution during generation: this distribution appears only as the posterior $q_t^\theta(x_1 | x)$ in our variational objective (Equation 8), not as the base distribution from which we sample noisy points.
>
> The choice of base distribution is flexible and application-dependent, which is why we don't discuss a specific approach in the main text. In our experiments, we use a simple and practical strategy: we sample from a standard Gaussian in the ambient Euclidean space and then project these samples onto the manifold using the exponential map. This makes sampling straightforward while maintaining the geometric structure.

---

> ### Author Response · Authors · 2025-11-21
> **2/3**
>
> > **Q4.** I would argue that the paper is not particularly novel on the methods front (not that it is without contributions), mostly (variational) flow matching on Riemannian geometries. (...) What is the novelty in your method [3], beyond the theoretical guarantees: isn’t end-point prediction a rather well-known “trick”? (...) Training for end-point prediction is not new [3].
>
> **A4.** The theoretical analyses and derived guarantees, as well as the accompanying experiments, should indeed be considered our main contributions. Regarding the end-point prediction "trick", which is arguably well-known, its Riemannian formulation is nevertheless non-trivial (for non-triviality arguments, please refer to our answer **A1** -and corresponding **Appendix**- to Reviewer CUa3).
>
> The methods using this trick [1-4] treat Euclidean and non-Euclidean parameters differently: Euclidean parameters use what effectively corresponds to variational flow matching (predicting endpoints, minimizing MSE), while non-Euclidean parameters use a partially variational form - endpoints are predicted, but the loss minimizes squared distance between ground-truth and predicted velocities (via the logarithm map) rather than between data points. This reveals a gap: full alignment would suggest minimizing geodesic distance between predicted and target data points in the non-Euclidean case, which is exactly what our method does.
>
> Since prior work shows that endpoint learning improves empirical performance, we see our approach - together with [5] - as providing theoretical justification for this choice. Framing endpoint prediction within the variational flow matching context may seem like a minor contribution, but it's necessary for enabling rigorous theoretical analysis.
>
> **References**
>
> [1] Kim, Nayoung, et al. "Mofflow: Flow matching for structure prediction of metal-organic frameworks." (2024).
>
> [2] Yim, Jason, et al. "SE (3) diffusion model with application to protein backbone generation." (2023).
>
> [3] Yue, Angxiao, Zichong Wang, and Hongteng Xu. "ReQFlow: rectified quaternion flow for efficient and high-quality protein backbone generation." (2025).
>
> [4] Bowen Jing, Bonnie Berger, Tommi Jaakkola. "AlphaFold Meets Flow Matching for Generating Protein Ensembles"
>
> [5] Eijkelboom, Floor, et al. "Variational flow matching for graph generation." (2024).
>
>
> > **Q5.** Have you had the opportunity to try out your method on manifolds where the higher order terms you mention appear (non-constant curvature)? I understand that you might not have, as the examples could be rare, but in which case the contribution of the paper is less significant, practically speaking. This question can also be understood as “Why does Riemannian Flow Matching work so well if it misses those important terms?” (or are they not so important therefore?)
>
>
>
> **A5.** Thank you for this interesting question, it helps us clarify an important aspect of our contribution. We address your concerns in two parts:
>
> - Regarding the curvature-dependent terms in our experiments: they remain nonzero for constant curvature manifolds, as shown in Corollary D.1. All our experiments use constant curvature manifolds (we have now made this explicit in the **Geometric interpretation** following Proposition 4.3), and this allows us to empirically validate the practical difference between RFM (which lacks the curvature-dependent terms) and RG-VFM (which captures them).
> - Regarding why RFM still performs well despite missing curvature terms: this observation actually highlights the nature of our contribution. Rather than proposing a revolutionary improvement over RFM, we provide stronger geometric supervision through a principled and simple loss function modification. The key insight is that incorporating the curvature term requires only a straightforward modification to RFM's loss (compare Algorithms 5 and 7) with no additional computational cost, simply using geodesic distance in the objective. The modest but consistent performance improvements align with our expectations: RFM already performs well by approximating the geometry, and our method refines this approximation cost-free. The magnitude of improvement depends on the manifold's curvature, but the cost-benefit ratio is decisive: better geometric fidelity at no expense. This is especially valuable in applications like protein and materials design, where incremental improvements in sample quality can meaningfully impact downstream scientific outcomes.

---

> ### Author Response · Authors · 2025-11-21
> **3/3**
>
> > **Q6.** Perhaps I have missed out on this, but it seems that there is no discussion on the hypotheses made about the manifold (homogeneity and existence of closed form geodesics). (...) Does homogeneity alongside existence of closed form geodesics not trivialise the structure of the manifolds considered? (I am not certain at all, I would like to discuss.)
>
> **A6.** Thank you for raising this important point. We agree that we did not sufficiently discuss what these assumptions mean in practice and how they relate to the manifolds typically encountered in applications. To address this, we have extended the discussion in Section 3.1, immediately before Proposition 3.1.
>
> These assumptions do not trivialize the manifold structure as much as one might initially expect. First, we note that homogeneity and closed-form geodesics are independent properties in general: homogeneous manifolds do not necessarily admit closed-form geodesics, and conversely, manifolds with explicit geodesics need not be homogeneous.
>
> However, for the simple geometries commonly used in Riemannian flow matching applications, the closed-form geodesic property naturally brings homogeneity along with it, so both of them hold. The canonical spaces - hyperspheres $\mathbb{S}^n$, hyperbolic spaces $\mathbb{H}^n$, flat tori $\mathbb{T}^n$, SPD matrices $\mathcal{S}_d^+$ (with affine-invariant metric), rotation groups $SO(3)$, and rigid transformations $SE(3)$ - all satisfy both conditions. These manifolds cover a wide range of practical applications: materials and protein design on $SO(3)$ [1,2,3], geometry-aware robotic planning on end-effector space $SE(3)$ [4,5] and joint-configuration space $\mathbb{T}^n$ [6]. Since our theoretical analysis focuses on comparison with RFM on these standard geometries, working under both assumptions is not restrictive in practice, it simply formalizes the structure already present in the problems of interest. The homogeneity assumption additionally ensures that the curvature-dependent term in our analysis (Proposition 3.1) behaves uniformly across the manifold, enabling clean theoretical comparisons between methods.
>
> **References**
>
> [1] Kim, Nayoung, et al. "Mofflow: Flow matching for structure prediction of metal-organic frameworks." (2024).
>
> [2] Yim, Jason, et al. "SE (3) diffusion model with application to protein backbone generation." (2023).
>
> [3] Yue, Angxiao, Zichong Wang, and Hongteng Xu. "ReQFlow: rectified quaternion flow for efficient and high-quality protein backbone generation." (2025).
>
> [4] Braun, Max, et al. "Riemannian flow matching policy for robot motion learning." (2024).
>
> [5] Ding, Haoran, et al. "Fast and robust visuomotor riemannian flow matching policy." (2025).
>
> [6] Chatzipantazis, Evangelos, Nishanth Rao, and Kostas Daniilidis. "STRiDE: STate-space Riemannian Diffusion for Equivariant Planning." (2025).

---

> ### Author Response · Authors · 2025-11-21
> **Discussion**
>
> We thank the reviewer for their constructive feedback, which has strengthened our work. We welcome any additional questions or clarifications. If the provided responses have adequately addressed the concerns raised, we would appreciate a reassessment of the evaluation score.

---

### Official Review · Reviewer_fzbJ · 2025-10-31

**Soundness:** 3
**Presentation:** 4
**Contribution:** 3
**Rating:** 6
**Confidence:** 3

**Summary:**

This paper introduces **Riemannian Gaussian Variational Flow Matching (RG-VFM)**, a geometric extension of Variational Flow Matching (VFM) for generative modeling on manifolds.
The authors argue that on curved manifolds, endpoint prediction (as in VFM) provides a stronger learning signal than velocity prediction (as in Flow Matching, FM, or Riemannian FM, RFM) because it directly minimizes **geodesic distances**.

Key contributions:
1. **A variational flow matching objective on general Riemannian manifolds**, employing Riemannian Gaussian distributions with closed-form geodesics.
2. **A formal analysis linking RG-VFM to RFM** via Jacobi fields, showing that curvature-dependent terms naturally emerge in the RG-VFM objective but are absent in RFM.
3. **Experiments** on synthetic (spherical/hyperbolic checkerboards), **materials (MOF)**, and **protein backbone generation** tasks. Results indicate improved geometric consistency and generation quality over Euclidean and velocity-based baselines.

**Strengths:**

1. The paper provides a clear and well-motivated theoretical bridge between **variational** and **geometric** generative modeling. The idea of introducing curvature-dependent terms via Jacobi fields to analyze flow-matching losses is novel and elegant.
2. The derivation is rigorous, with propositions and proofs connecting RG-VFM and RFM. The mathematical treatment of curvature effects is insightful, especially Proposition 4.3, showing curvature-dependent correction terms.
3. The paper is well written and technically organized, with clear mathematical notation and well-illustrated figures (e.g., Fig. 1–3 showing geometric intuition).

**Weaknesses:**

1. The method is limited to manifolds with **closed-form geodesics**. This assumption restricts applicability to simple spaces (e.g., \(S^n\), \(H^n\)), leaving open how to handle more general manifolds.
2. While results are positive, the experiments lack deeper ablations (e.g., sensitivity to curvature magnitude, loss variants, or effect of variance parameter σ in the Riemannian Gaussian).
3. The paper briefly mentions that RG-VFM maintains simplicity of linear flows but does not analyze training/sampling cost relative to RFM or diffusion models.

**Questions:**

1. How would the approach generalize to manifolds without analytic exponential/log maps (e.g., learned manifolds from data)?
2. Could the curvature-dependent term be approximated or regularized for arbitrary manifolds, enabling broader applicability?
3. In Table 3, could the authors provide variance/error bars for the protein design metrics to quantify statistical significance?
4. Is there any insight on the trade-off between extrinsic (RG-VFM-R³) and intrinsic (RG-VFM-M) variants beyond the empirical performance?

---

> ### Author Response · Authors · 2025-11-21
> **1/4**
>
> We thank the reviewer for the positive feedback on our work and for taking the time to evaluate our manuscript, we truly appreciate it. Below, we address all concerns and questions, grouping those on common topics together, and we hope to continue the discussion if any points require further clarification.
>
> > **Q1.1.** The method is limited to manifolds with closed-form geodesics. This assumption restricts applicability to simple spaces (e.g., (S^n), (H^n)), leaving open how to handle more general manifolds.
>
> **A1.1.** You are correct that our model is designed for closed-form geodesics, which we acknowledge as a scope limitation in multiple sections of the manuscript. However, we'd like to clarify why this is not necessarily a significant constraint (or at least, not only specific to our work):
>
> - This design choice is intentional. For the applications in our experiments, extending the method to manifolds without closed-form geodesics proved unnecessary, and we intentionally avoided introducing additional theory or model variants that would make the methodology overly complex and fragmented. Importantly, a substantial portion of Riemannian Flow Matching applications involve manifolds that are simple enough to have closed-form geodesics available. This includes not only materials and protein design on $SO(3)$ [1-3], but also geometry-aware robotic planning, where both end-effector space ($SE(3)$) [4,5] and joint-configuration space ( $\mathbb{T}^n$) [6] have closed-form geodesics.
> - While manifolds without closed-form geodesics fall outside our current scope, our method could be straightforwardly extended to more complex geometries using the same approaches employed by other geometric generative methods - for instance, through geodesic simulations [7].
>
> > **Q1.2.** How would the approach generalize to manifolds without analytic exponential/log maps (e.g., learned manifolds from data)?
>
> **A1.2.** Regarding generalization to manifolds without closed-form exponential and logarithmic maps: this would be definitely possible by adapting the approach from RFM [7], which handles such cases through geodesic simulation. However, developing a rigorous method for general manifolds requires substantial mathematical development, that would be infeasible within the rebuttal timeline, and also beyond the scope of the current paper. This would be an interesting direction to explore in future work, especially when motivated by specific applications that demonstrate the need for such generalization.
>
> Additionally, our extrinsic formulation, RG-VFM-$\mathbb{R}^n$ , already relaxes these constraints significantly. In the extrinsic setting, the only manifold-specific operation required is the geodesic distance for loss computation - exponential and logarithmic maps are not needed.
>
> **References**
>
> [1] Kim, Nayoung, et al. "Mofflow: Flow matching for structure prediction of metal-organic frameworks." (2024).
>
> [2] Yim, Jason, et al. "SE (3) diffusion model with application to protein backbone generation." (2023).
>
> [3] Yue, Angxiao, Zichong Wang, and Hongteng Xu. "ReQFlow: rectified quaternion flow for efficient and high-quality protein backbone generation." (2025).
>
> [4] Braun, Max, et al. "Riemannian flow matching policy for robot motion learning." (2024).
>
> [5] Ding, Haoran, et al. "Fast and robust visuomotor riemannian flow matching policy." (2025).
>
> [6] Chatzipantazis, Evangelos, Nishanth Rao, and Kostas Daniilidis. "STRiDE: STate-space Riemannian Diffusion for Equivariant Planning." (2025).
>
> [7] Chen, Ricky TQ, and Yaron Lipman. "Flow matching on general geometries." (2023).

---

> ### Author Response · Authors · 2025-11-21
> **2/4**
>
> > **Q2.** While results are positive, the experiments lack deeper ablations (e.g., sensitivity to curvature magnitude, loss variants, or effect of variance parameter σ in the Riemannian Gaussian).
>
> **A2.** Thank you for pointing out this weakness, your concern is entirely valid. We added clarifications on these points in Section 3.1 and Section 5.1. We address the lack of ablations from several perspectives:
>
> - In terms of curvature magnitude, we test our model on constant curvature spaces with $K=1$ and $K=-1$, which we believe represent the range of curvatures typically encountered in applications. While additional curvature values could be explored, these reference cases provide strong evidence for the method's behavior across positive and negative curvature regimes.
> - We actually evaluate two loss variants in the synthetic experiments, corresponding to squared $L^2$ versus non-squared $L^1$ geodesic distance in the Riemannian Gaussian. Equivalently, this corresponds to using either a Riemannian Gaussian or Riemannian Laplace as the posterior distribution. Since the Laplace variant is somewhat tangential to our main contributions, we only mention this in the main text (Section 5.1) but just properly discuss it in Appendix E.4.
> - You are correct that we did not adequately discuss the variance parameter, although we actually observed interesting behaviors. The variance could be set to a constant $\sigma_t(x) = 1$, or its parameterization can incorporate time-dependent normalization (e.g., it can be defined as $\sigma_t(x) = 1-t$). In our synthetic experiments, we tested variants with and without the $(1-t)^2$ normalization factor in the loss and found that its presence makes negligible difference in practice. We have now added clarifications on this in Section 3.1 and Section 5.1.
>
> For MOF and protein-backbone generation, our focus was on isolating the effect of the loss function by making minimal changes to the RFM [1] framework, which limited the scope for extensive ablations. However, we note that MOFFlow-2 [2], a follow-up work of MOFFlow [3], found that removing the $1/(1-t)^2$ normalization can be beneficial for endpoint-prediction objectives, which likely extends to our setting, though this claim would require further investigation in real-world applications.
>
> **References**
>
> [1] Chen, Ricky TQ, and Yaron Lipman. "Flow matching on general geometries." (2023).
>
> [2] Kim, Nayoung, Seongsu Kim, and Sungsoo Ahn. "Flexible MOF Generation with Torsion-Aware Flow Matching." (2025).
>
> [3] Kim, Nayoung, et al. "Mofflow: Flow matching for structure prediction of metal-organic frameworks." (2024).
>
>
>
>
>
> > **Q3.1.** The paper briefly mentions that RG-VFM maintains simplicity of linear flows but does not analyze training/sampling cost relative to RFM or diffusion models.
>
> **A3.1.** Thank you for this observation, we were actually lacking some explicit statements in the paper regarding the training and sampling costs of our model with respect to the other references that we mention, such as RFM and VFM. In order to address this concern, we enrich the paper with the following:
>
> - We add schematic algorithms corresponding to the training and sampling of our method, RFM, and VFM, to make the methodological and computational comparisons direct and straightforward, see Algorithms 1 to 8.
> - We also add a corresponding discussion of the computational costs (end of Section 4, in the **Geometric interpretation** discussion before Section 5), for which we will elaborate more below.
>
> By "linear flow", we suppose you are referring to the following claim in Section 3: *The extrinsic framework maintains the simplicity and efficiency of a linear flow, avoiding the need for the manifold's exponential and logarithmic maps, while encoding more geometric information than purely Euclidean methods.* If so, we will clarify below what we mean, adding the requested information on computational costs:
>
> - RG-VFM in its extrinsic form (RG-VFM-$\mathbb{R}^n$) maintains the same computational complexity as VFM, both in training and sampling. The term “linear” refers to the type of interpolation performed to sample $x_t$, which is the same as for VFM. In this setting, the only difference between VFM and our method is that the former uses standard $L^2$ distance in the endpoint computation, while we use geodesic distance (see Algorithms 2 and 4). Since we assume the geodesic distance to be available explicitly in closed form, it doesn’t add any computational overhead with respect to VFM.

---

> ### Author Response · Authors · 2025-11-21
> **3/4**
>
> (Continuation of **A3.1**)
> - RG-VFM in its intrinsic form (RG-VFM-$\mathcal{M}$ ) maintains the same computational complexity as RFM, with the difference that the velocity computation occurs during sampling rather than training (see Algorithms 1 and 5). This means we perform one fewer operation during training and one additional operation during sampling, which compensates overall. Regarding the losses, the same explanation as above holds: while RFM uses squared distance on the tangent space, we use geodesic distance, still assuming it has an explicit closed form, thus adding no additional cost.
>
> > **Q3.2.** Could the curvature-dependent term be approximated or regularized for arbitrary manifolds, enabling broader applicability?
>
> **A3.2.** In general, estimating the curvature correction term is not tractable. It involves the full Riemann curvature tensor and its covariant derivative, both of which depend on high-order derivatives of the metric. Without strong geometric symmetries, evaluating these quantities along sampled geodesics becomes computationally prohibitive. One possible simplification is to assume an isotropic manifold with constant sectional curvature, where the curvature tensor has a closed-form expression. However, isotropic manifolds are homogeneous by definition, so this information would already be encoded in the RG-VFM loss.
> So, while the idea is very interesting, it appears intractable in practice for non-homogeneous manifolds.
>
> > **Q4.** In Table 3, could the authors provide variance/error bars for the protein design metrics to quantify statistical significance?
>
> **A4.** We appreciate this suggestion and totally agree that reporting statistical significance would strengthen our results. We did not include error bars initially because we follow the established practice in this domain: the original works we compare against (MOFFlow [1], ReQFlow [2]) do not report statistical significance either, and this constitutes standard practice for MOF and protein generation benchmarks.
>
> While we agree that statistical significance would be valuable, the computational constraints make it infeasible to generate these results within the rebuttal period, unfortunately: MOF and protein-backbone generation experiments are computationally expensive and time-intensive, requiring resources beyond what we deploy in the available time. We will make sure to follow this practice in future extensions of the paper.
>
> To ensure validity despite the absence of error bars, we already took the following precautions (so improvement cannot be attributed to seed selection):
> - For MOF experiments, we use the same random seeds as MOFFlow [1] for all reproduced baselines, for fair comparison. In the case of MOFFlow, we initially tried 3 different seeds for a reduced set of experiments, and the standard deviation for our model across these seeds for sample #1 and stol=0.5 is $1.161$ for MR and $0.0053$ for RMSE, which supports the significance of our results, at least for these settings.
> - For protein-backbone generation, we follow ReQFlow’s [2] evaluation protocol when selecting the reported results, and we adopt the same evaluation metrics used in [2]. We argue that the RMSD reported in the table - computed using ESMFold and ProteinMPNN - is not fully deterministic (due to ProteinMPNN) and is additionally influenced by model-specific biases. Nevertheless, the consistent improvement in designability (92.4% vs. 96.8% for VQFlow, with only the loss being variationalized) suggests that adopting our framework is beneficial. We acknowledge that the current metrics used for mini-length proteins (50–300 residues) are somewhat saturated. In future work, we plan to investigate the training dynamics for longer proteins under Riemannian variational flow matching.
>
> **References**
>
> [1] Kim, Nayoung, et al. "Mofflow: Flow matching for structure prediction of metal-organic frameworks." (2024).
>
> [2] Yue, Angxiao, Zichong Wang, and Hongteng Xu. "ReQFlow: rectified quaternion flow for efficient and high-quality protein backbone generation." (2025).

---

> ### Author Response · Authors · 2025-11-21
> **4/4**
>
> > **Q5.** Is there any insight on the trade-off between extrinsic (RG-VFM-R³) and intrinsic (RG-VFM-M) variants beyond the empirical performance?
>
> [We report here part of the answer **A2** for reviewer CUa3, who asked a similar question together with other related concerns.]
>
> **A5.** The choice between the intrinsic and extrinsic variants is not really a trade-off, but rather reflects two different perspectives, where one may fit the problem at hand better than the other. For example, when simulating weather changes on Earth, it may be useful to maintain the same spherical support for all distributions, which would favor the intrinsic approach.
>
> However, the methods do come with distinct benefits and limitations that may influence which one is ideal for a given problem. The extrinsic version requires an ambient space $\mathbb{R}^n$ of sufficiently large dimension to embed the manifold without degeneracy. When applicable though, it offers concrete practical advantages, for example it is significantly simpler to implement than RG-VFM-$\mathcal{M}$ as it requires no exponential/logarithmic maps or manifold-aware ODE solvers. Moreover, some manifolds admit closed-form geodesic distances but lack analytic exponential/logarithmic maps. In such cases, RG-VFM-$\mathbb{R}^n$ remains viable while RG-VFM-$\mathcal{M}$ would require costly numerical approximations at every training and sampling step.

---

> ### Author Response · Authors · 2025-11-21
> **Discussion**
>
> We thank the reviewer for their valuable feedback in strengthening this work. We welcome any further questions and are available for immediate discussion. If our responses have resolved the concerns raised, we would appreciate a reassessment of the score to reflect these improvements.

---

### Official Review · Reviewer_CUa3 · 2025-10-31

**Soundness:** 3
**Presentation:** 4
**Contribution:** 2
**Rating:** 6
**Confidence:** 3

**Summary:**

This paper focuses on extending Variational Flow Matching (VFM) to Riemannian manifolds through the proposed Riemannian Gaussian Variational Flow Matching (RG-VFM). The authors argue that endpoint prediction provides a stronger learning signal by directly minimizing geodesic distances. The method formulates a variational flow matching objective based on Riemannian Gaussian distributions, applicable to manifolds with closed-form geodesics. They further demonstrate with experiments on spherical, hyperbolic, and real-world datasets that RG-VFM can better captures manifold geometry.

**Strengths:**

- Very well-written paper with excellent explanations and visuals.
- Variational extension of FM is well motivated; and this paper lifts this idea to Riemannian manifolds.
- The derivations and design choices are well-justified and appear natural.
- Good empirical results in two different real-world domains and on one synthetic dataset.

**Weaknesses:**

- The work appears somewhat incremental, as extending Riemannian flow matching to a variational formulation is relatively straightforward. Formally establishing a connection between RG-VFM and RFM through Proposition 4.1 is useful, but not very surprising.
- The discussion of intrinsic versus extrinsic viewpoints is interesting, yet it remains unclear how the extrinsic perspective offers a clear advantage. While it provides additional flexibility, it essentially represents a trade‑off rather than an optimal method. Similar to the difference between VFM and FM, the primary benefit seems to arise mainly from supervision at the endpoints.
- Although the ability to choose an arbitrary variational distribution is often presented as a main strength of variational flow matching, this flexibility is rarely reflected in practice, since most experimental setups still rely on simple distributional choices. It is not easy for the reader to establish a direct connection between this generic motivation and the experimental setups (esp. for readers less familiar with these benchmarks). It would be much nicer if the authors could elaborate how this type of generality is used in practical setup.

**Questions:**

Please refer to weaknesses.

---

> ### Author Response · Authors · 2025-11-21
> **1/3**
>
> We thank the reviewer for the positive assessment of our work, and for taking the time to evaluate our manuscript and express their concerns. We address each of the reviewers' concerns separately below, and hope to continue the discussion if any of them are not clarified properly.
>
>
> > **Q1.** The work appears somewhat incremental, as extending Riemannian flow matching to a variational formulation is relatively straightforward. Formally establishing a connection between RG-VFM and RFM through Proposition 4.1 is useful, but not very surprising.
>
>
> **A1.** Thank you for raising this point, as it gives us an opportunity to clarify the development process and theoretical contributions of our work.
>
> We appreciate that our exposition makes the method appear straightforward, as this clarity was intentional. However, the apparent simplicity belies substantial theoretical and practical contributions. First, we emphasize a crucial distinction: we are not extending Riemannian Flow Matching (RFM) to a variational formulation. Rather, we generalize the variational flow matching (VFM) framework itself to incorporate geometric information. This generalization required developing new mathematical results using nontrivial geometric tools never before applied in machine learning contexts. Beyond the method itself, we establish a novel theoretical connection to RFM and provide thorough experimental analyses assessing the practical impact of various design choices within the flow matching paradigm.
>
> In Euclidean space, conditional Flow Matching and Variational Flow Matching share a natural equivalence up to normalization terms [1,2]. No such closed-form connection exists between RFM and geometric variational models. We therefore built both the method and its mathematical foundations from scratch, without recourse to prior work in either geometric flow matching or variational flow matching. Establishing the relationship between our model and RFM required completely novel mathematical results involving Jacobi vector fields corresponding to a specific family of geodesics, and these results bring mathematical novelty beyond merely connecting RFM and VFM intuitively.
>
> Our experiments also tackle a gap in the literature that hasn't been explicitly recognized before. Existing diffusion and flow-based models for structured data like proteins and materials already made use of variational loss for their Euclidean components, but not for their geometric components - likely because no method was available and the extension wasn't straightforward. Beyond introducing a novel method based on Jacobi fields, which haven't been used in this ML context before (to our knowledge), we identify and resolve this methodological limitation, demonstrating practical benefits in settings where geometric variational approaches were previously impossible.
>
> **Appendix to A1**
>
> As an appendix, we report additional detailed information regarding the development steps and the connection between RFM and RG-VFM, in case it can be useful for clarifying further doubts.
>
> - Regarding the development process, we'd like to clarify that RG-VFM does not arise from simply "variationalizing" Riemannian flow matching. Rather, it stems from two key observations: (1) the variational flow matching framework makes the posterior distribution $q_t^\theta(x_1 | x)$ explicit, and (2) incorporating information about the distribution's support into this posterior could significantly benefit geometric methods.
> Developing these ideas into a working method required numerous non-trivial design choices, each with substantial mathematical justification. For instance, once we recognized that the variational objective could incorporate support information, we faced the question of which distribution family to use for geometric domains. Even restricting ourselves to normal distributions, there exist multiple definitions of "normal" on manifolds - each with different levels of generality - and we specifically chose the Riemannian Gaussian of Pennec [3] for principled reasons. Simply inserting this choice into the objective yields an extremely complex expression with intricate normalization terms. At this stage, we had to decide whether to work with the most general objective, introduce approximations, or impose manifold restrictions that would enable simplification (and prove why such simplifications are valid). We chose the latter approach. While the final objective appears intuitive, this clarity emerged from careful mathematical choices and analyses rather than straightforward generalization.

---

> ### Author Response · Authors · 2025-11-21
> **2/3**
>
> (Continuation of **Appendix to A1**)
> - On the connection between RFM and RG-VFM: we appreciate that Proposition 4.1 may appear natural in retrospect. However, establishing this connection required overcoming a significant theoretical challenge. In Euclidean space, the CFM and VFM objectives are equivalent up to a normalization constant [1,2]. In the Riemannian setting, this equivalence breaks down due to manifold curvature, as there is no closed-form connection analogous to the Euclidean case (this absence is related to the triangle comparison theorem). Our relation includes higher-order error terms that reflect genuine geometric differences. We explored several approaches for characterizing these differences and ultimately found that encoding both objectives as specific values of a Jacobi vector field along a tailored family of geodesics provided the most elegant and geometrically natural explanation.
>
>
> > **Q2.** The discussion of intrinsic versus extrinsic viewpoints is interesting, yet it remains unclear how the extrinsic perspective offers a clear advantage. While it provides additional flexibility, it essentially represents a trade‑off rather than an optimal method. Similar to the difference between VFM and FM, the primary benefit seems to arise mainly from supervision at the endpoints.
>
> **A2.** Thank you for this insightful question, as it highlights an important distinction we should clarify more explicitly. We have now added a dedicated discussion at the end of Section 3.1 to address this point. To facilitate comparison, we have also included detailed algorithms for both variants (Algorithms 1 and 2) as well as for related methods (Algorithms 3-6).
>
> First, we want to emphasize that both RG-VFM extrinsic and intrinsic variants perform endpoint supervision through our variational objective (following the VFM framework [2]), so the advantage of the extrinsic approach does not stem from this aspect. Rather, the key distinction lies in **where** the flow operates and what geometric information it encodes.
>
> More in detail,  the intrinsic  RG-VFM-$\mathcal{M}$ framework  learns flows entirely on the manifold, requiring geodesic interpolation and manifold-aware ODE integration.  The extrinsic RG-VFM-$\mathbb{R}^n$ instead learns linear flows in ambient Euclidean space, incorporating geometric information only through the geodesic distance in the loss. Nevertheless, this is not simply a Euclidean method, as the geometry-aware loss ensures the learned flow respects the manifold's structure, even though the flow itself remains computationally simple.
>
> The choice between the intrinsic and extrinsic variants is not really a trade-off, but rather reflects two different perspectives, where one may fit the problem at hand better than the other. For example, when simulating weather changes on Earth, it may be useful to maintain the same spherical support for all distributions, which would favor the intrinsic approach.
>
> However, the methods do come with distinct benefits and limitations that may influence which one is ideal for a given problem. The extrinsic version requires an ambient space $\mathbb{R}^n$ of sufficiently large dimension to embed the manifold without degeneracy. When applicable though, it offers concrete practical advantages, for example it is significantly simpler to implement than RG-VFM-$\mathcal{M}$ as it requires no exponential/logarithmic maps or manifold-aware ODE solvers. Moreover, some manifolds admit closed-form geodesic distances but lack analytic exponential/logarithmic maps. In such cases, RG-VFM-$\mathbb{R}^n$ remains viable while RG-VFM-$\mathcal{M}$ would require costly numerical approximations at every training and sampling step.

---

> ### Author Response · Authors · 2025-11-21
> **3/3**
>
> > Q3. Although the ability to choose an arbitrary variational distribution is often presented as a main strength of variational flow matching, this flexibility is rarely reflected in practice, since most experimental setups still rely on simple distributional choices. It is not easy for the reader to establish a direct connection between this generic motivation and the experimental setups (esp. for readers less familiar with these benchmarks). It would be much nicer if the authors could elaborate how this type of generality is used in practical setup.
>
> **A3.** Thank you for this valuable feedback. You raise an important point about connecting the theoretical flexibility of variational flow matching to practical implementation choices. We address this from two perspectives:
>
> - Flexibility in VFM more broadly: while our work focuses on Riemannian Gaussian posteriors, the broader VFM literature demonstrates diverse distributional choices tailored to specific domains. For instance, CatFlow [2] uses categorical posteriors for discrete data generation, while recent work has applied VFM to mixed modalities [4], molecular generation [5,6], and constrained problems like sea-ice forecasting where physical bounds (e.g., non-negative thickness) are enforced through the variational loss [7]. These examples illustrate how problem-specific constraints can be incorporated directly into the variational objective, which is a key advantage of the framework.
>
> - How we exploit this flexibility: our choice of a Riemannian Gaussian posterior is itself an instance of this flexibility, generalizing the standard Euclidean Gaussian, which results in replacing Euclidean with geodesic distances in the loss function. This geometric adaptation is central to our contribution. Beyond this specific choice, our framework naturally extends to a broader Riemannian exponential family of distributions. The key insight is that different distance metrics in the loss correspond to different posterior distributions: while squared geodesic distance ($L^2$) yields a Riemannian Gaussian, using $L^1$ geodesic distance corresponds to a Riemannian Laplace distribution. We start to investigate this in our synthetic experiments (Section 5.1), comparing the two variants empirically. The results show that the Riemannian Laplace posterior ($L^1$ distance) achieves better performance than the Riemannian Gaussian ($L^2$ distance) in some cases, demonstrating that the choice of posterior distribution can be adapted based on the characteristics of the problem. While these results are promising, developing a complete mathematical framework for the entire Riemannian exponential family is beyond the scope of this paper, which focuses primarily on the Riemannian Gaussian distribution. For this reason, we provide a detailed treatment of the $L^1$ distance variant in Appendix E.4, while the main text concentrates on the $L^2$ formulation.
>
> For the real-world experiments (MOFs and protein backbones), we deliberately use the intrinsic Riemannian Gaussian to enable direct comparison with RFM, isolating the effect of our loss while maintaining the same distributional support. This controlled comparison demonstrates the value of geometric awareness in practical settings. However, you are correct that further variations, such as switching to the extrinsic perspective or exploring alternative posteriors, represent promising directions we plan to investigate in future work.
>
>
> **References**
>
> [1] Lipman, Yaron, et al. "Flow matching guide and code." (2024).
>
> [2] Eijkelboom, Floor, et al. "Variational flow matching for graph generation." (2024).
>
> [3] Pennec, Xavier. "Intrinsic statistics on Riemannian manifolds: Basic tools for geometric measurements." (2006).
>
> [4] Guzmán-Cordero, Andrés, Floor Eijkelboom, and Jan-Willem van de Meent. "Exponential Family Variational Flow Matching for
> Tabular Data Generation." (2025).
>
> [5] Eijkelboom, Floor, et al. "Controlled Generation with Equivariant Variational Flow Matching." (2025).
>
> [6] Sakalyan, Kristiyan, et al. "Modeling microenvironment trajectories on spatial transcriptomics with nicheflow." (2025).
>
> [7] Finn, Tobias Sebastian, et al. "Generative AI models enable efficient and physically consistent sea-ice simulations." (2025).

---

> ### Author Response · Authors · 2025-11-21
> **Discussion**
>
> We sincerely appreciate the reviewer's time and effort in helping us refine this manuscript. We welcome any remaining questions and are available for immediate clarification on any points. If the provided responses have successfully resolved all prior concerns, we would be grateful if the reviewer would reassess and consider an increased evaluation score.

---

### Author Response · Authors · 2025-11-21
**Manuscript revised**

We sincerely thank all reviewers for their thorough evaluation of our work and their positive remarks regarding the proposed method. We appreciate the time invested in analyzing our contribution and raising thoughtful concerns and questions.

In response to the feedback received, we have revised the manuscript to incorporate additional clarifications and algorithmic descriptions that address the points raised. We believe these revisions strengthen the paper and make our contributions more accessible. We momentarily keep the **additions in red color** to clarify which parts were added with respect to the previous version.

We address each reviewer's specific comments in detail below.

---

> ### Author Response · Authors · 2025-11-26
> **Further manuscript revision**
>
> Based on Reviewer 423B's additional feedback, we have revised the abstract and introduction to improve clarity on the motivations and contributions, highlighting the modified sentences in red.

---

### Author Response · Authors · 2025-12-02
**General response: summary of discussion and main questions**

We thank the reviewers for their evaluation and constructive feedback. We are pleased to know that the paper was found to be “very well-written” with “clear mathematical notation,” “rigorous derivations,” and an “elegant” theoretical connection between RG-VFM and RFM via Jacobi fields. We appreciate the positive assessment of the experiments as “extensive and convincing”.

All reviewers recommended for acceptance, with one reviewer (423B) explicitly indicating their will to increase their score after our reply, while responding was still possible on OpenReview. The other reviewers did not have the opportunity to reply, but they had already indicated weak acceptance, and we similarly addressed all their questions.

We value the reviewers' feedback on motivation, novelty, empirical results, and geometric assumptions, and have fully addressed these points for improvement (indicated in red in the updated PDF), summarized them below.

> ### **1: Contributions and motivations.**

A key point for attention raised by some reviewers (CUa3, Kr3K) involves clarifying our motivation and whether our contributions are incremental or novel. However, other reviewers (fzbJ, 423B) clearly recognized the non-incremental, significant contributions already. We took the opportunity to further clarify the novelty and impact of our work, which we summarize as follows.

- Our work is motivated by two observations: (1) VFM's flexibility provides significant benefits across applications, and (2) it naturally allows the geometric structure of the data support to be built into the objective.

- Our main contributions are: (1) proposing RG-VFM, as an extension of VFM to Riemannian manifolds using Riemannian Gaussian posteriors (Proposition 3.1); (2) deriving **formal and new theoretical results** that highlight curvature-dependent terms that do not appear in RFM (Proposition 4.3),  linking RG-VFM to RFM via Jacobi fields; (3) presenting empirical results on synthetic manifolds and real-world datasets (MOFs and proteins) that suggest **improved generation quality** compared to RFM, without increasing computational cost.

> ### **2: Requirement of homogeneous manifolds with closed-form geodesics.**

Some reviewers (Kr3K, fzbJ) commented that the assumptions of homogeneous manifolds with closed-form geodesics may be restrictive, and asked how our method would generalize beyond these constraints. We note that:
- Nearly all RFM applications involve manifolds with readily available closed-form geodesics, e.g. $SO(3)$ for material and protein design, and $SE(3)$ or $\mathbb{T}^n$ for robot planning.
- Homogeneity is essential for the well-posedness of our loss definition (Proposition 3.1), while the closed-form geodesic assumption could be relaxed using numerical methods such as geodesic simulation, as done in RFM. However, a rigorous extension to general manifolds is beyond the scope of this work.

> ### **3: Intrinsic vs extrinsic frameworks.**

Reviewers (Kr3K, CUa3) questioned the meaning of the terms "intrinsic" and "extrinsic" viewpoint—which are key features of our new approach—and the trade-offs between these variants in practice.

We clarified our use of "intrinsic" and "extrinsic" terminology and emphasized that the choice between these variants reflects complementary perspectives suited to different applications rather than a performance trade-off. For clarity, we added the pseudo-code of the intrinsic and extrinsic RG-VFM (Algorithms 1,2).

> ### **4: High-order curvature term and computational complexity.**

Some reviewers (fzbJ, 423B) wondered about the computational and experimental impact of the high-order curvature terms involved in Proposition 4.3, and more broadly about the computational costs of our method:
- The high-order curvature terms are implicitly encoded in our loss function and become explicit only through the Taylor expansion of Equation 13 (provided for theoretical comparison with RFM). These terms impact real-world applications, as they are non-zero even for manifolds with constant curvature.
- The computational complexity of extrinsic RG-VFM is identical to that of VFM/CFM, while the intrinsic version matches that of RFM. To facilitate comparison, we added all pseudo-codes in the appendix (Algorithms 1-8).

> ### **5: Significance of empirical results.**

While two reviewers found the experimental section extensive and convincing, others suggested improving discussion on statistical significance (423B) and adding deeper ablations (fzbJ).

We agree that reporting statistical significance would strengthen our results. However, we follow established practice in MOF and protein-backbone generation, where error bars are not typically reported. While computational constraints prevent us from generating these results during the rebuttal period, we commit to including them in future work. Regarding ablations, these were already partially addressed in Appendix E.4 and have now been further clarified in the main text.

---

### Meta-Review · Area_Chair_qMyg · 2026-01-05

**Summary:**

1. Some reviewers (CUa3, Kr3K) concerns about the contribution being incremental

2. Some reviewers (Kr3K, fzbJ) commented that the assumptions of homogeneous manifolds with closed-form geodesics may be restrictive, and asked how the method would generalize beyond these constraints.

3. Reviewers (Kr3K, CUa3) questioned the meaning of the terms "intrinsic" and "extrinsic" viewpoint

4. Some reviewers (fzbJ, 423B) wondered about the computational and experimental impact of the high-order curvature terms involved in Proposition 4.3, and more broadly about the computational costs of the method.

5. Reviewer fzbJ commented that the experiments lack deeper ablations (e.g., sensitivity to curvature magnitude, loss variants, or effect of variance parameter σ in the Riemannian Gaussian). Reviewer 423B commented that the significance of the empirical results is not entirely convincing.

**Reviewer Concerns:**

After the rebuttal, for the above 5 concerns,

1. The authors clarify their contributions. I have carefully read the paper including the revised version. I think although the novelty and contribution does not standout significantly (top 25%), but is significant enough for at least a poster presentation.

2. The authors clarify their assumptions, and says that these assumptions are not restrictive in practice, which I agrees.

3. The authors clarify that they use the terms "intrinsic" and "extrinsic" to distinguish whether points lie on the manifold or in the ambient space, rather than to describe coordinate choices. This one is properly addressed.

4. This one is addressed.

5. For the ablation, the authors say that the experiments in the paper did contains certain ablations (K=1 and -1, loss variants). For the statistical significance, the authors explains that computational constraints make it infeasible to generate full statistical significance results within the rebuttal period. For this concern, I think the authors could have strengthened more, for example, adding more and deeper ablations.

Overall I think most of the major concerns are addressed. The concern 5 could be strengthened more. The authors are encouraged to add more deeper ablations for the camera-ready paper.

**Reviewer Scores:**

I think CUa3 will remain their score of 6 due to novelty evaluations; fzbJ will remain score of 6 since the ablation experiments could have been more and deeper; Kr3K will remain score of 6 due to novelty evaluations; 423B will remain score of 6 due to that the significance of empirical results.

---

### Decision · Program_Chairs · 2026-01-26

Accept (Poster)